# Characterizing the Training Dynamics of Private Fine-tuning with Langevin diffusion

**Shuqi Ke**                                                            *shuqik@andrew.cmu.edu*
*Carnegie Mellon University*

**Charlie Hou**                                                       *charlieh@andrew.cmu.edu*
*Carnegie Mellon University*

**Sewoong Oh**                                                *sewoong@cs.washington.edu*
*University of Washington*

**Giulia Fanti**                                                        *gfanti@andrew.cmu.edu*
*Carnegie Mellon University*

**Reviewed on OpenReview:** *https://openreview.net/forum?id=LwT8aDv502*

## Abstract

We show that **d**ifferentially **p**rivate **f**ull **f**ine-**t**uning (DP-FFT) can distort pre-trained backbone features based on both theoretical and empirical results. We identify the cause of the distortion as the misalignment between the pre-trained backbone and the randomly initialized linear head. We prove that a sequential fine-tuning strategy can mitigate the feature distortion: first-linear-probing-then-fine-tuning (DP-LP-FFT). A new approximation scheme allows us to derive approximate upper and lower bounds on the training loss of DP-LP and DP-FFT, in a simple but canonical setting of 2-layer neural networks with ReLU activation. Experiments on real-world datasets and architectures are consistent with our theoretical insights. We also derive new upper bounds for 2-layer linear networks without the approximation. Moreover, our theory suggests a trade-off of privacy budget allocation in multi-phase fine-tuning methods like DP-LP-FFT.

## 1 Introduction

Today, many differentially-private (DP) machine learning pipelines proceed in two phases: (1) A model is pre-trained (non-privately) on a public dataset. (2) The model is then fine-tuned on private data, using DP optimization techniques such as DP stochastic gradient descent (DP-SGD) and its variants (Hoory et al., 2021; De et al., 2022; Tang et al., 2023; Zhang et al., 2024b). Pre-training a backbone model on public data enables differentially private fine-tuning to achieve improved performance across various downstream tasks (Yu et al., 2022) and is proven to be necessary in some cases (Ganesh et al., 2023a).

Despite these advances, the effect of DP on fine-tuning training dynamics remains poorly understood. Several key questions are yet to be answered: (1) how does randomness (both of initialization and DP optimization) impact the pre-trained representations? (2) What are the convergence rates of common fine-tuning methods, such as DP **f**ull **f**ine-**t**uning (DP-FFT) and DP **l**inear **p**robing (DP-LP, where feature representations are frozen, and only the linear head is fine-tuned)? (3) Prior work suggests that combining an early stage of DP-LP with a later stage of DP-FFT yields better privacy-utility tradeoffs

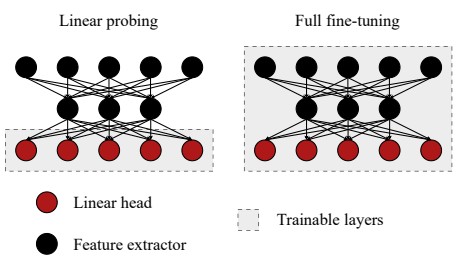

Figure 1: Linear probing (LP) freezes the lower layers and optimizes the last linear layer while full fine-tuning (FFT) optimizes the whole network.

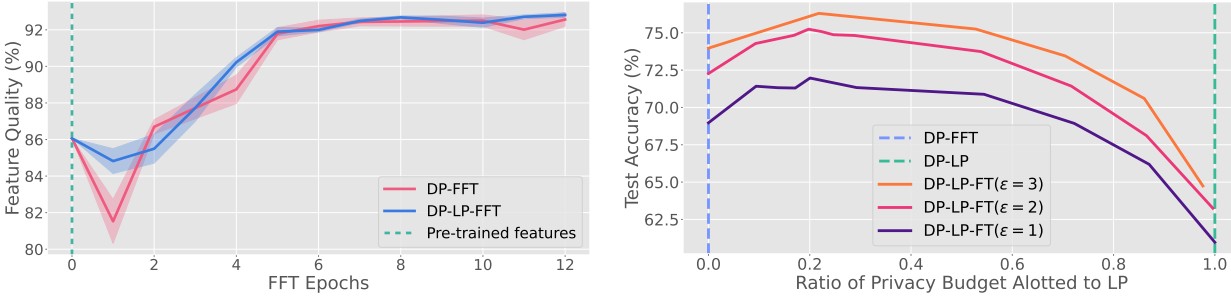

Figure 2: **Left:** Backbone feature quality evaluated by top-1 kNN accuracy on the downstream task, for ResNet-50, through public pre-training on ImageNet-1K and differentially private fine-tuning on STL-10. **Right:** Privacy budget trade-off in DP-LP-FFT, predicted in our theory, for WideResNet-16-4 on CIFAR-10 (Tang et al., 2023). For a detailed explanation, refer to

(Tang et al., 2023), yet there is no theoretical understanding
of this phenomenon, nor is it clear how to optimally combine
these fine-tuning methods.

Answering these questions theoretically requires an analysis that can capture the fine-grained optimization dynamics of DP fine-tuning. We seek a model of DP finetuning that satisfies 2 properties.

1. **Architecture-sensitivity:** The convergence dynamics must differentiate between representation learning in the backbone and learning in the linear head. The analyses of Bassily et al. (2014),Wang et al. (2022),Fang et al. (2023),Ganesh et al. (2023b) focus only on the network's dimension, failing to capture this distinction.

2. **Ability to model nonlinearities:** The model should account for the nonlinearities introduced by multi neural layers, unlike existing methods that simplify analysis by linearizing neural networks (Ye et al., 2023a; Wang et al., 2024).

We propose a novel approximation of DP-SGD training dynamics based on linearizing Langevin diffusion around *the noise term*. This approach offers new insights into DP fine-tuning and significantly simplifies analysis by converting stochastic differential equations into ordinary differential equations (ODEs). We validate our theoretical predictions with real experiments.

**Main contributions.** In summary, our key contributions are:

1. **New approximation technique:** In Section 2, we derive a first-order ODE via an asymptotic expansion of the stochastic noise in Langevin diffusion. Unlike previous methods, which linearize neural network parameters, our technique preserves the multi-layer structure of deep learning models while simplifying the analysis. This approach, commonly used in physics and control theory (Skorokhod et al., 2002), is novel in the context of private machine learning and bridges the gap between non-private neural network theory and the private regime.

2. **Understanding of feature distortion:** In Section 3, we provide a theoretical understanding of how DP fine-tuning affects feature representations. Using our approximation, we prove that, in 2-layer ReLU networks, randomly initialized linear heads distort pre-trained backbone features in the early stages of DP-FFT. Empirically Figure 2 demonstrates that feature quality evaluated on private data initially degrades during DP-FFT but later improves and surpasses pre-fine-tuning quality. Our theory also predicts that running a single epoch of DP-LP before transitioning to DP-FFT can mitigate this initial feature distortion, as shown empirically in the DP-LP-FFT curve of Figure 2 (left). This insight extends the findings of Kumar et al. (2022), who showed that LP-FFT reduces feature distortion in non-private, OOD scenarios, to in-distribution settings for both DP and non-DP cases.

3. **Theoretical convergence bounds:** In Section 4, we present new upper and lower bounds on the training loss of DP-LP and DP-FFT for 2-layer ReLU networks using our approximation technique.

We also prove upper bounds for 2-layer linear networks without the approximation. To the best of our knowledge, this is the first convergence analysis of DP-SGD on non-linear neural network architectures.

4. **Mitigating feature distortion by combining fine-tuning methods:** Prior work by Tang et al. (2023) empirically showed that combining DP-LP and DP-FFT (DP-LP-FFT) can achieve better test accuracy than either method alone. In Figure 2b, we demonstrate that allocating approximately 20% of the privacy budget to DP-LP yields optimal test accuracy. In Section 5, we provide a partial theoretical explanation for this phenomenon. Specifically, our bounds suggest that DP-FFT may underperform relative to DP-LP at lower privacy budgets, while DP-LP-FFT can outperform both methods under moderate privacy budgets. These predictions are empirically verified across various architectures and benchmarks in Section 5.3.

## 1.1 Related Work

Similar empirical phenomena have been explored in non-private, out-of-distribution (OOD) contexts by Aghajanyan et al. (2021), Kumar et al. (2022), Trivedi et al. (2023), and Chen et al. (2024). Kumar et al. (2022) demonstrated that non-DP fine-tuning distorts pre-trained features, leading to degraded OOD performance. But their theory relies on the assumption that OOD test data exists in an orthogonal subspace to the fine-tuning training data, leaving their results unable to explain why, in many transfer learning tasks, linear-probe fine-tuning (LP-FFT) still outperforms both LP and full fine-tuning (FFT) in in-distribution (ID) settings. Our work seeks to fill this research gap.

Wang et al. (2024) examined how pre-trained representations enhance DP fine-tuning within the neural collapse framework, though their analysis was restricted to the final layer. Meanwhile, Tang et al. (2023) empirically observed the privacy budget trade-off for WideResNet models pre-trained on synthetic data, but without accompanying theoretical insights.

Analyses by Wang et al. (2019), Chen et al. (2020a), Ganesh et al. (2023b), and Fang et al. (2023) rely on standard convexity/non-convexity and smoothness assumptions, which abstract away the simultaneous dynamics between the backbone and linear head. Other works (Ye et al., 2023b; Wang et al., 2024) focus on linearized models, limiting their ability to capture the nuanced interactions between these components. Our explanation of representation alignment builds on the theoretical foundation of Min et al. (2024), which we extend to a DP context using novel approximation tools.

## 2 Continuous modeling of differentially private fine-tuning

**Notation.** We use $\partial$ to denote both the deterministic and stochastic differential operators. The dot product between vectors $x, y$ is $x^\top y$, the Euclidean norm of vector $x$ is $\|x\|_2$, and the infinity norm is $\|x\|_\infty$. The trace of a matrix is denoted by tr, and the ReLU activation is $\phi$. For any twice differentiable function $f(x)$, its gradient is denoted $\nabla_x f$ and its Hessian as $H_x f$. $\sqcup$ denotes the disjoint union. $[i] := \{1, \ldots, i\}$. The cosine similarity between two vectors $u, v$ is defined as $\cos(u, v) = \frac{u^\top v}{\|u\|_2 \|v\|_2}$. We denote the privacy cost estimated by Rényi divergence as $r$.

**DP-SGD Dynamics.** Differential privacy (DP) is a widely used framework for evaluating privacy leakage in a dataset accessed through queries (Dwork & Roth, 2014). In machine learning, DP ensures that an adversary cannot confidently determine whether specific training samples are part of the dataset. **D**ifferentially **P**rivate **S**tochastic **G**radient **D**escent (DP-SGD), introduced by Abadi et al. (2016), is the standard algorithm for training deep neural networks while maintaining privacy.

Our fine-tuning theory is built on an analysis of DP-SGD dynamics. Although real-world algorithms are discrete, continuous approximations—such as stochastic differential equations (SDE) like Langevin diffusion—are often used to study these dynamics (Chourasia et al., 2021; Ye et al., 2023b). In a similar vein, Kumar et al. (2022) use gradient flow, a continuous approximation of SGD, to study fine-tuning in a non-private context.

**Definition 2.1** (Langevin diffusion (Ganesh et al., 2023b))**.** Langevin diffusion is an SDE that models the dynamics of a system influenced by both deterministic and random forces (Lemons & Gythiel, 1997). For

DP-SGD, we define a $p$-dimensional Langevin diffusion as follows:

$$\partial\theta = -\nabla_\theta \mathcal{L}(\theta|f)\partial t + \sqrt{2\sigma^2}\partial Q_t, \tag{1}$$

where $\theta \in \mathbb{R}^p$ represents the neural network parameters, $f$ is the network architecture, $\mathcal{L}(\cdot|f) : \mathbb{R}^p \to \mathbb{R}$ is the training loss, and $\sigma > 0$ is the noise multiplier (Abadi et al., 2016). $\{Q_t\}_{t\geq0}$ is the standard Brownian motion in $\mathbb{R}^m$ modeling the Gaussian noise mechanism.

By Itô's lemma (Ito, 1951), the Langevin diffusion of the training loss is given by

$$\partial\mathcal{L} = \left[-\|\nabla_\theta\mathcal{L}(\theta|f)\|_2^2 + \sigma^2\mathrm{tr}(\nabla_\theta^2\mathcal{L})\right]\partial t + \sqrt{2\sigma^2}(\nabla_\theta\mathcal{L}(\theta|f))^\top\partial Q_t. \tag{2}$$

Ye et al. (2023b) study how random initialization affects DP-SGD performance in linearized neural networks via Langevin diffusion. To facilitate theoretical analysis, they linearize the entire neural network using $1^{\text{st}}$-order Taylor expansions at the initial parameter $\theta_0$.

$$f(x) \approx f_{\text{lin}}(x) := f(x)\Big|_{\theta=\theta_0} + \frac{\partial f(x)}{\partial\theta}\Big|_{\theta=\theta_0} \cdot (\theta - \theta_0). \tag{3}$$

Recently, this linearization technique has gained popularity for explaining key deep learning phenomena (Ortiz-Jimenez et al., 2021). However, fully linearizing the model removes critical multi-layer interactions, making this approach unsuitable for our analysis.

To address this, we view the optimization trajectory of neural networks as a dynamical system, with noise in gradient updates treated as random perturbations. We first rewrite a Langevin diffusion like Equation (1) in the following form

$$\partial\theta = F(\theta)\partial t + \sigma G(\theta)\partial Q_t \tag{4}$$

where $F$ is the drift coefficient and $G$ is the diffusion coefficient. We then introduce a small–noise (regular) perturbation expansion of the Langevin dynamics in the spirit of Freidlin–Wentzell (Freidlin et al., 2012). In particular, we decompose a Langevin diffusion (e.g. Equation (1)) to a power series of the perturbation scale $\sigma$

$$\theta = \theta^{(0)} + \sigma\theta^{(1)} + \sigma^2\theta^{(2)} + \cdots, \tag{5}$$

where we define each $\theta^{(i)}$ as

$$\theta^{(i)} = \sum_{r=1}^{i}\frac{1}{r!}\sum_{i_1+\cdots+i_r=i,i_j\geq1}\nabla^r F[\theta^{(i_1)},\ldots,\theta^{(i_r)}]\partial t + \sum_{r=1}^{i-1}\frac{1}{r!}\sum_{i_1+\cdots+i_r=i,i_j\geq1}\nabla^r G[\theta^{(i_1)},\ldots,\theta^{(i_r)}]\partial Q_t. \tag{6}$$

Like Taylor's expansion, we can approximate $\theta$ with the partial sum $\sum_{i=0}^{N}\sigma^i\theta^{(i)}$ and the remainder $\theta - \sum_{i=0}^{N}\sigma^i\theta^{(i)}$ is infinitely small compared with $\sigma^N$, uniformly on any finite interval $[0,T]$. The approximation order $N$ gives us various accuracies for the deviations caused by the random perturbations.

Applying the zeroth-order asymptotic expansion ($N=0$) for the parameter dynamics $\theta$ (Equation (1)) and the loss dynamics $\mathcal{L}$ (Equation (2)), we approximate:

$$\partial\theta \approx \partial\tilde{\theta} = -\nabla\mathcal{L}\left(\tilde{\theta}\big|f\right)\partial t. \tag{7}$$

In the zeroth-order expansion, we ignore the noise term $\partial Q_t$ and only keep the noise effect term $\sigma^2\mathrm{tr}(\nabla_\theta^2\mathcal{L})$ in the loss dynamics. This zeroth-order expansion helps circumvent the complex analysis of stochastic, non-linear equations. By substituting the approximate parameter $\tilde{\theta}$ into Equation (2), our modeling partially preserves the noisy behavior characteristic of DP-SGD. We further explore this property in the next section.

## 2.1 Zeroth order approximation

The noise multiplier $\sigma$ remains explicitly in our convergence bounds. We retain the key noise effects for the loss dynamics by keeping the second-order term from Ito's lemma in Equation (2) and preserving the second-order terms associated with Brownian motion.

This approach allows us to capture the essential stochastic characteristics of DP-SGD without modeling the full noise term directly on the parameters. In essence, this approximation enables us to analyze the expected behavior of parameter updates while preserving the noise-sensitive behavior of the loss itself. By isolating these core elements, we provide insights into the overall training dynamics under differential privacy without losing the major noise effects that influence convergence properties and feature alignment.

To support our claim that this approximation does not introduce too much error, we have proved an error approximation guarantee, which shows that our approximated model does not differ too much from the original Langevin diffusion model. We present the theorem based on Langevin diffusion with gradient clipping. We use the subscript $t$ in $\theta_t$ to denote the parameter $\theta$ at training step $t$.

$$\text{Clipped Langevin diffusion: } \partial\theta_t = -\sum_{i\in[n]} \text{clip}_C(\nabla\ell_i(\theta_t|f))\partial t + \sqrt{2\sigma^2}\partial Q_t,$$

$$\text{Zeroth order approximation: } \partial\tilde{\theta}_t = -\sum_{i\in[n]} \text{clip}_C\left(\nabla\ell_i\left(\tilde{\theta}_t|f\right)\right)\partial t, \tag{8}$$

$$\text{where } \text{clip}_C(u) := \min\left(1, \frac{C}{\|u\|_2}\right)u.$$

**Theorem 2.2** (Zeroth order approximation error). *Denote the model parameter vector in original Langevin diffusion as $\theta_t$, and its zeroth-order approximated version as $\tilde{\theta}$. For any training time $t > 0$ and clipping threshold $C > 0$,*

$$\mathbb{E}\left[\left\|\theta_t - \tilde{\theta}_t\right\|^2\right] \leq \left(\sigma(2p)^{\frac{1}{2}}t^{\frac{1}{2}} + 2nCt\right)^2 \tag{9}$$

Note that this approximation error significantly improves upon the $O(\exp(T))$ error found under standard regularity assumptions (Freidlin et al., 2012, Theorem 1.2, Chapter 2.1). The approximation does not remove the effect of noise, nor is the resulting model equivalent to gradient flow. We defer the proof to Appendix F.

The the best of our knowledge, this is the first analysis of clipped Langevin diffusion as a continuous model of DP-SGD. We present more technical details in Appendix F.

## 3 Representation Alignment

In this section, we introduce the concept of representation alignment, present our theoretical findings, and validate them with experiments. Representation alignment refers to the process by which the classification head aligns itself with the pre-trained backbone features. During the DP-FFT process, this alignment creates a characteristic trend in feature quality: initially, the randomly initialized linear head distorts the pre-trained features, but as it better aligns with the backbone, the distortion diminishes, and the overall quality of the backbone features improves over time.

### 3.1 Theory

Our goal is to understand (1) how does DP fine-tuning distort the pre-trained features in the backbone, and (2) under what conditions this distortion can be mitigated. We consider the simple binary classification setup from Min et al. (2024), which provides a clear and intuitive understanding of representation alignment. The results generalize to our experiments in Section 3.2. Specifically, we use a 2-layer fully-connected neural network with $h$ hidden nodes and ReLU activation $\phi$,

$$f(x) = v^\top g(x) = v^\top \phi(W^\top x) = \sum_{j=1}^{h} v_j \phi(w_j^\top x). \tag{10}$$

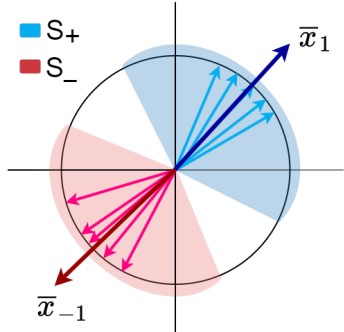

Figure 3: Visualization of Assumption 3.1.

fine-tuning on a dataset $\mathcal{D} := \{(x_i, y_i)\}_{i=1}^n$ with $n$ inputs $x_i \in \mathbb{R}^{d_x}$, and binary labels $y_i \in \{-1, 1\}$. The objective is to minimize the training

loss $\mathcal{L}(\tilde{\theta}|f) := \sum_{i=1}^{n} \ell(y_i, f(x_i))$, using the exponential loss $\ell(y, \hat{y}) := \exp(-y\hat{y})$. Similar results hold for logistic loss (Min et al., 2024).

Our use of a two-layer surrogate and a zeroth-order ODE is a local approximation around the pre-trained weights. In the short horizon that governs the distortion phase, it has been previously shown that deep networks behave approximately like their linearization (Jacot et al., 2018; Lee et al., 2019; Kumar et al., 2022); the dominant term is the interaction between the head's random initialization and the backbone's Jacobian under DP-SGD updates. This is precisely what our surrogate captures.

For simplicity, we make the two assumptions.

**Assumption 3.1** (Data correlation (Min et al., 2024))**.** For any pair of data $(x_i, y_i), (x_j, y_j)$, the inputs are positively/negatively correlated if the labels are the same/different.

$$\inf_{i,j\in[n]} \left[ (y_i y_j) \cdot \frac{x_i^\top x_j}{\|x_i\|_2 \|x_j\|_2} \right] := \mu > 0. \tag{11}$$

We define two cones in $\mathbb{R}^{d_x}$ that separate subspaces spanned by data points in the positive and negative classes, respectively: $S_+ = \{z \in \mathbb{R}^{d_x} : \forall i \in [n], \mathbb{I}_{x_i^\top z > 0} = \mathbb{I}_{y_i=1}\}, S_- = \{z \in \mathbb{R}^{d_x} : \forall i \in [n], \mathbb{I}_{x_i^\top z > 0} = \mathbb{I}_{y_i=-1}\}$. Min et al. (2024) prove that $S_+ \cap S_- = \emptyset$, and $x_i \in S_{+/-}$ if $y_i = 1/-1$ (see Figure 3). We define the mean data directions of class $c \in \{-1, 1\}$ by $\bar{x}_c := \sum_{i\in[n]} x_i \cdot \mathbb{I}_{y_i=c}$.

We assume that a "clustering" behavior emerges in the pre-trained features, which allows the features to work well in transfer learning (Galanti et al., 2022). This phenomenon is well-documented in the neural collapse literature (Kothapalli, 2023), suggests that pre-trained features $w_j$ tend to converge around the mean direction for data in class $c(j)$.

**Assumption 3.2** (Collapsed neural features)**.** For each $w_j$ in Equation (10) where $j \in [h]$ (with $h$ denoting the dimension of the linear head), it holds that $w_j \in S_+$ or $w_j \in S_-$. We define $c(j) = 1$ if $w_j \in S_+$, and $c(j) = -1$ if $w_j \in S_-$. Thus, there is a partition $[h] = F_+ \sqcup F_-$ over the index set $[h]$, such that for each $w_j$,

$$\begin{cases} j \in F_+ \text{ if } w_j \in S_+, \\ j \in F_- \text{ if } w_j \in S_-. \end{cases} \tag{12}$$

**Feature quality.** Assumption 3.2 says that data with positive label (resp. negative) only activates the $j$-th neuron if $j \in F_+$ (resp. $j \in F_-$). As a result, any positive data pair, $(x, y)$ and $(x, y')$ with $y = y'$, activate the same set of neurons. From a contrastive learning viewpoint, this assumption makes the representations of them semantically similar (Saunshi et al., 2019). Namely, when the features $w_j$ and data inputs $x_i$ are normalized unit vectors, the difference between representations of a positive data pair is bounded by:

$$\|g(x) - g(x')\|_\infty \le \max_{y_i = c(j) = y} \cos(w_j, x_i), \tag{13}$$

which represents the maximum cosine similarity between the features $w_j$ and the data points.

Note that our assumptions are local/early-phase and serve to make the distortion mechanism transparent. We further discuss the relaxation of the assumptions in Appendix B.1.

However, FFT or DP-FFT with random initialization may reduce the feature quality.

**Theorem 3.3** (Random initialization causes feature distortion)**.** *If Assumption 3.1 and Assumption 3.2 hold, and the linear head is randomly initialized by $v_0 \sim \mathcal{N}(0, \beta I_{h\times h})$, then with probability $1 - 2^{-h}$, $\forall \beta > 0, \exists j \in [h], \Delta t > 0$ such that during the time interval $(0, \Delta t)$, DP-FFT distorts $w_j$ reducing its alignment with the data cluster. The cosine similarity between $w_j$ and the data cluster mean $\bar{x}_{c(j)}$ decreases monotonically:*

$$\frac{\partial}{\partial t} \cos\left(w_j, \bar{x}_{c(j)}\right)\Big|_t < 0, \quad \forall t \in (0, \Delta t) \tag{14}$$

For a pre-trained $w_j$ that aligns with $c(j)$-labeled data, DP-FFT (as modeled by Equation (7)) makes it deviate from $\bar{x}_{c(j)}$, the mean direction of those data. $w_j$ is optimal when $\cos(w_j, \bar{x}_{c(j)}) = 1$. This result holds

for both DP and non-DP settings and explains the potential feature distortion observed in in-distribution and non-private settings, such as those studied by Kumar et al. (2022)). The stochastic analysis of non-smooth loss, activation, cosine similarity functions is challenging without our approximation.

Next, we show that running (DP-)LP before (DP-)FFT could mitigate feature distortion.

**Theorem 3.4** (DP-LP first mitigates feature distortion). *Suppose Assumption 3.1 and Assumption 3.2 hold, and the linear head is randomly initialized by $v_0 \sim \mathcal{N}(0, \beta I_{h \times h})$ for any $\beta > 0$. There exists $\Delta t > 0$ such that after running DP-LP for time $\Delta t$, switching to full fine-tuning ensures that DP-FFT does not distort the pre-trained features. Specifically, $\cos(w_j, \bar{x}_{c(j)})$ is non-decreasing for all $j \in [h]$:*

$$\frac{\partial}{\partial t} \cos\left(w_j, \bar{x}_{c(j)}\right)\Big|_t \geq 0, \quad \forall t \in (\Delta t, +\infty) \tag{15}$$

See complete proofs of Theorem 3.3 and Theorem 3.4 in Appendix C.1.

**Corollary 3.5** (Non-DP feature distortion). *The results in Theorem 3.3 and Theorem 3.4 still hold in non-DP case ($\sigma = 0$). In particular, if Assumption 3.1 and Assumption 3.2 hold and the linear head is randomly initialized by $v_0 \sim \mathcal{N}(0, \beta I_{h \times h})$:*

1. *Then with probability $1 - 2^{-h}$, $\forall \beta > 0, \exists j \in [h], \Delta t > 0$ such that during the time interval $(0, \Delta t)$, FFT distorts $w_j$:*

$$\frac{\partial}{\partial t} \cos\left(w_j, \bar{x}_{c(j)}\right)\Big|_t < 0, \quad \forall t \in (0, \Delta t). \tag{16}$$

2. *There exists $\Delta t$ such that after running LP for time $\Delta t$, FFT does not distort the pre-trained features. Specifically, $\cos(w_j, \bar{x}_{c(j)})$ is non-decreasing for all $j \in [h]$:*

$$\frac{\partial}{\partial t} \cos\left(w_j, \bar{x}_{c(j)}\right)\Big|_t \geq 0, \quad \forall t \in (\Delta t, +\infty). \tag{17}$$

### 3.2 Experiments on Representation Alignment

In this section, we show empirical evidence supporting Theorems 3.3 and 3.4.

**Pre-training and Model.** We pre-train Vision Transformers (ViT) and ResNet-50 backbones on ImageNet-1K using Self-Supervised Learning methods, including BYOL (Grill et al., 2020) and MoCo v2 (Chen et al., 2020b), as well as distillation methods (Touvron et al., 2021). Then we fine-tune the backbone with a linear classification head on CIFAR-10 and STL-10 using DP-SGD.

**Experiment protocols.** We conduct public pre-training for 100 epochs with a batch size of 256. Following this, we implement DP-SGD using the pre-trained weights and a randomly initialized linear head for 30 epochs. Each DP fine-tuning process is repeated with 5 random seeds and a batch size of 1000. We evaluate the backbone features on both the pre-training and fine-tuning datasets, measuring feature quality through top-1 kNN accuracy (Chen et al., 2023).

**Private fine-tuning initially distorts features.** Figure 4 qualitatively visualizes the effect of DP-FFT on feature quality with respect to the private test data. We pre-train (BYOL) a ResNet-50 backbone on ImageNet-1K and DP fine-tune (DP-SGD, $\epsilon = 1$) it on STL-10. We qualitatively assess the features of the private test data within the ResNet-50 backbone by visualizing the backbone mappings (outputs from the penultimate layer) of data points using UMAP (McInnes et al., 2020). For simplicity, we only plot 3 classes in CIFAR-10.

Figure 4 indicates that during the initial phases of DP-FFT, the randomly initialized linear head interferes with the pre-trained features in the backbone network, leading to a degradation in feature quality on both the pre-training and fine-tuning datasets. This observation validates Theorem 3.3. Concurrently, the linear head begins adapting to these pre-trained features, a process we refer to as "**representation alignment**." As this alignment progresses, the backbone starts to regain a portion of its original feature quality, which had been degraded by DP noise and shifts in data distribution.

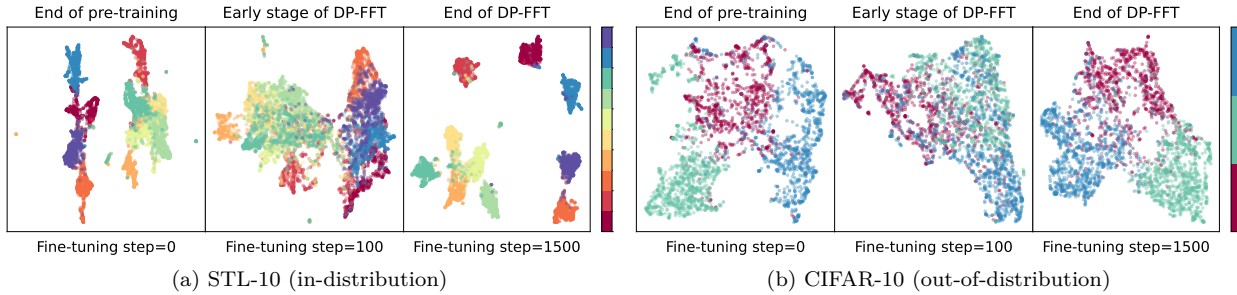

(a) STL-10 (in-distribution)                    (b) CIFAR-10 (out-of-distribution)

Figure 4: We pre-train (BYOL) a ResNet-50 backbone on ImageNet-1K and DP fine-tune (DP-SGD, $\epsilon = 1$) it on STL-10. We qualitatively evaluate the features in the ResNet-50 backbone by visualizing the backbone mappings (penultimate layer outputs) of data points via UMAP (McInnes et al., 2020). These results suggest that DP-FFT distorts feature quality before improving it, as predicted by Theorem 3.3.

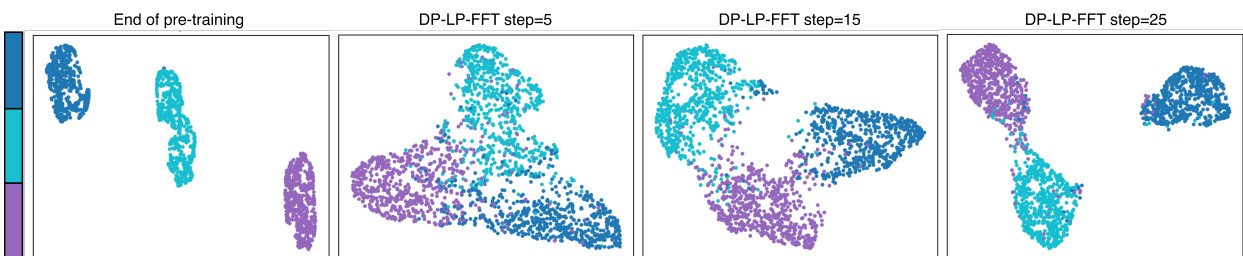

Figure 5: UMAP of penultimate-layer features on a subset of MNIST (labels $\{0,3,7\}$). We visualize the features at the end of non-private pretraining and the end of DP fine-tuning. As we increase the number $q$ of DP-LP steps ($q \in \{0, 10, 20\}$) before DP-FFT (5 steps), we observe that the the severity of the feature distortion reduces.

**Linear probing mitigates feature distortion.** To illustrate the benefits of linear probing, we first run DP-LP for 1 epoch before transitioning to DP-FFT for the remaining epochs. In the initial steps of DP-FFT, the feature distortion is significantly weaker (Figure 2a if we first run DP-LP. This supports the claim of Theorem 3.4. Similarly, we evaluate features on the pre-training domain (see Figure 7).

We also visualize with UMAP the penultimate-layer features on MNIST (labels 0,3,7) in taken at two checkpoints of the training pipeline: non-private pretrain and final DP-FFT (after some early DP-LP steps). In Figure 5, the pretrain panel (left-most) shows three compact, well-separated clusters. We switch to DP-LP after the pre-training stage. We consider three settings with different DP-LP steps. In the second, third, and fourth plots (left→right in Figure 5), we run DP-LP for $0, 10, 20$ epochs respectively, and then run DP-FFT for 5 epochs after the DP-LP phase. We fix the noise multiplier to $\sigma = 1$ for DP-LP and $\sigma = 5$ for DP-FFT.

As our theory predicted, private updates in DP-FFT induce the expected feature distortion: class prototypes drift from their pretrain locations, clusters elongate and partially mix along a shared manifold, and the inter-class margin narrows relative to the increase in intra-class spread. This behavior is consistent with our theory that, at the onset of DP-FFT, gradients are misaligned due to (i) the random or poorly aligned classification head and (ii) DP noise injected into per-sample gradients; as a result, the backbone momentarily adapts in directions that do not preserve the pretrain geometry. When we increase the number of DP-LP, we effectively mitigate the feature distortion: the clusters are better aligned and separated (though not identical to the pretrain configuration).

# 4 DP Fine-tuning Convergence Rates

Section 3 showed that DP-LP-FFT can mitigate feature distortion. A natural question is, for a fixed privacy budget, how do DP-LP and DP-FFT affect the convergence of fine-tuning loss function? We study this question under two models: (1) our zeroth-order approximation of Langevin diffusion (Section 4.1), and (2) a two-layer neural network without our zeroth-order approximation (Section 4.1.1). The second result will be used to study the budget allocation of DP-LP-FFT in Section 5. To our knowledge, these are the first convergence guarantees (approximate or not) for DP fine-tuning on explicit nonlinear neural network architectures.

**Privacy guarantees**  We begin by establishing the privacy guarantees of Langevin diffusion by bounding the Rényi divergence of its trajectory distributions on neighboring datasets (Mironov, 2017). Both Ganesh et al. (2023b) and Ye et al. (2023b) show that the Rényi divergence increases linearly over time. We use this guarantee for all fine-tuning variants.

**Theorem 4.1** (Rényi privacy guarantee (Ganesh et al., 2023b))**.** *Suppose we initialize a pair of neural network parameters* $\theta, \theta'$ *by some i.i.d. distributions* $\Theta_0, \Theta_0'$*. We fine-tune* $\theta, \theta'$ *respectively on neighboring datasets* $\mathcal{D}, \mathcal{D}'$ *via Langevin diffusion. Denote the distribution of the trajectory of* $\theta$ *by* $\Theta_{[0,T]}$ *over* $[0,T]$*. Similarly, denote the trajectory distribution of* $\theta'$ *by* $\Theta_{[0,T]}'$*. Then for any* $\alpha \geq 1$*, the Rényi divergence* $R_\alpha$ *is bounded linearly in time,*

$$r := R_\alpha(\Theta_{[0,T]} \| \Theta_{[0,T]}') = O\left(\frac{\alpha \Delta_g T}{\sigma^2}\right) \tag{18}$$

*where* $\sigma$ *is the noise multiplier, and* $\Delta_g \geq \|\nabla \mathcal{L}(\theta; \mathcal{D}) - \nabla \mathcal{L}(\theta; \mathcal{D}')\|$ *is the upper bound of gradient difference between neighboring datasets. Thus, for any* $\delta \in (0,1)$*, the Langevin diffusion satisfies*

$$\left(\frac{\alpha \Delta_g T}{4\sigma^2} + \frac{\log(1/\delta)}{\alpha - 1}, \delta\right) - differential\ privacy. \tag{19}$$

## 4.1 Convergence Rates under the Zeroth-order Approximation

We follow the approximation scheme outlined in Equation (7)to derive convergence results for two-layer ReLU neural networks. These results are derived from our zeroth-order approximation; recall that we bound the error of this approximation relative to the Langevin dynamics model in Theorem 2.2. To support these findings, we also include a separate convergence proof *without the zeroth-order approximation* for a two-layer linear neural network in Section 4.1.1.

**Theorem 4.2** (Approximate DP-LP loss convergence)**.** *If Assumption 3.1 and Assumption 3.2 hold at* $t = 0$*, we can bound the loss after running DP-LP for* $t = T$*:*

$$\frac{1}{\frac{1}{\mathcal{L}_c(0)} e^{-B_1 T} + \frac{A_1}{B_1}(1 - e^{-B_1 T})} \leq \mathcal{L}_c(T) \leq \frac{1}{\frac{1}{\mathcal{L}_c(0)} e^{-B_2 T} + \frac{A_2}{B_2}(1 - e^{-B_2 T})} \tag{20}$$

*where* $\mathcal{L}_c(t)$ *denotes the training loss of data points labeled* $c \in \{-1, 1\}$*,* $\mathcal{L} = \mathcal{L}_1 + \mathcal{L}_2$*, and*

$$\begin{cases} A_1 = \sum_{w_j \in S_c} \left[\max_{y_i=c} w_j^\top x_i\right]^2 \\ B_1 = \frac{1}{2}\sigma^2 \left\{\sum_{y_i=c} \|\text{relu}(W^\top x_i)\|_2^{-2}\right\}^{-1} \\ A_2 = \sum_{w_j \in S_c} \left[\min_{y_i=c} w_j^\top x_i\right]^2 \\ B_2 = \frac{1}{2}\sigma^2 \left\{\sum_{y_i=c} \|\text{relu}(W^\top x_i)\|_2^4\right\}^{1/2} \end{cases} \tag{21}$$

*are constants for DP-LP.*

When we set $n = h = 2, y_1 = -y_2, w_1 = x_1 = -w_2 = -x_2$, the upper and lower bounds are equal and we achieve a tight bound on the DP-LP loss.

**Theorem 4.3** (Approximate DP-FFT loss convergence). *For simplicity, we assume that $\|x_i\|_2 = R$ for all $i \in [n]$. If Assumption 3.1 and Assumption 3.2 hold, and we consider a balanced initialization $\|W\|_F^2 = \|v_0\|_2^2$ (Min et al., 2023a) at $t = 0$, then*

*(i) we lower bound the loss after running DP-FFT for $T > 0$:*

$$\mathcal{L}_c(T) \geq \frac{1}{\frac{1}{\mathcal{L}_c(0)} e^{(1-\exp(\lambda_c T))A_l C_l/\lambda_c} + \frac{B_l}{C_l} \left[1 - e^{(1-\exp(\lambda_c T))A_l C_l/\lambda_c}\right]} \tag{22}$$

*where we define $A_l = \|W_0\|_F^2, B_l = 2R^2, C_l = \frac{R^2 \sigma^2 (1+\mu^2)}{2}$ and $\lambda_c = 2R\mathcal{L}_c(0)$.*

*(ii) we upper bound the loss after running DP-FFT for $T > 0$:*

$$\mathcal{L}_c(T) \leq \frac{1}{\frac{B_u}{C_u}(1 - e^{-A_c C_u T}) + \frac{1}{\mathcal{L}_c(0)} e^{-A_c C_u T}} \tag{23}$$

*where we define $A_c = \sum_{w_j \in S_c} \left[v_{j,t=0}^2 + \|w_j\|_2^2\right], B_u = R^2 \mu^2$ and $C_u = \frac{1}{2} R^2 \sigma^2$.*

### 4.1.1 Theory without the zeroth-order approximation (2-layer linear network)

We complement the results in Section 4.1 by removing the zeroth-order approximation in a simpler setup: 2-layer linear networks for a regression task. We define a linear network by replacing the ReLU activation $\phi$ with an identity function in Equation (10). We collect the data inputs in a matrix $X \in \mathbb{R}^{n \times d_x}$ and put the labels in a vector $Y \in \mathbb{R}^n$. For simplicity, we assume that $n \geq d$ and $X^T X = I_{d_x \times d_x}$. We consider the MSE training loss $\mathcal{L}(v, W) := \frac{1}{2} \sum_{i \in [n]} (v^\top W^\top x_i - y_i)^2 = \frac{1}{2} \|XWv - Y\|_2^2$.

Note that the loss function is nonconvex in the parameters being fine-tuned, so the gradient descent training becomes a nonlinear dynamical system. This significantly complicates theoretical analysis. Prior works have dealt with the challenging analysis by using heavy approximations (Bu et al., 2023; Ye et al., 2023b). We overcome these theoretical difficulties by using conservation laws and geometric properties of Langevin dynamics (see Appendix for more detail).

**Pretrained features.** We evaluate a backbone $W$ by the least square error:

$$\gamma(W) := \inf_{u \in \mathbb{R}^h} \mathcal{L}(u, W) = Y^T (I_{n \times n} - XW(XW)^\dagger)Y. \tag{24}$$

where $(\cdot)^\dagger$ denotes the pseudo inverse of a matrix. This metric measures the optimal loss for LP when fixing the current features. $\gamma = \gamma(W_0)$ denotes the initial least square error. We suppose $W_0$ has orthonormal columns, following prior works (Tripuraneni et al., 2020; Kumar et al., 2022).

**Theorem 4.4** (DP-LP loss convergence). *If we randomly initialize the linear head $v_0 \sim \mathcal{N}(0, \beta I_{h \times h})$ and we run linear probing for time $T$, then*

$$\mathbb{E}[\mathcal{L}(T)] \leq \frac{1}{2}(h\beta + \|Y\|^2)e^{-T} + (\gamma + h\sigma^2)(1 - e^{-T}) \tag{25}$$

In this theorem, the first term describes that the loss tends to exponentially decrease, while the second term describes the limiting behavior induced by linear probing and the added noise.

**Theorem 4.5** (DP-FFT loss convergence). *If $v_0 \sim \mathcal{N}(0, \beta I_{h \times h})$ and Assumption E.7 holds, and we run fine-tuning (Equation (127)) for time $T$, then the loss converges:*

$$\mathbb{E}[\mathcal{L}(T)] \leq \frac{1}{2}(h\beta + \|Y\|_2^2)e^{-AT} + L^\square(1 - e^{-AT}) \tag{26}$$

*where* $\begin{cases} A = h\beta - 1 - \sqrt{2}\sigma^2(1+d_x) > 0 \\ L^\square = \sigma^2 \frac{(1+d_x)\|X^T Y\|_2 + d_x}{A} \end{cases}$ .

This upper bound has a similar form to Equation (25) while the factor $A$ of the exponential terms depends on the initialization and the noise. When we take limit $\sigma \to 0$ in Theorem 4.4 and 4.5, the Langevin diffusion degenerates to a gradient flow and the loss converges exponentially to zero as $T \to \infty$. This recovers known results from the non-private optimization literature (Min et al., 2023a).

The bounds in Section 4.1 and Section 4.1.1 exhibit different dependencies on the hidden dimension $h$ and the data dimension $d_x$ due to the differing curvature properties of the loss functions in each setup. The underlying reason is that the noise term introduced by Itô's formula (Equation (2)) is influenced by the curvature of the loss function. While the square function has constant curvature, the exponential function does not, leading to varying noise impacts.

## 5   Budget Allocation between DP-LP and DP-FFT

Finally, we consider the DP-LP-FFT fine-tuning strategy, which first applies DP-LP for some portion $r$ of the privacy budget (i.e. for some number of training iterations), then uses the remaining privacy budget for DP-FFT. In this section, we ask: given a fixed privacy budget, how should we allocate it across DP-LP and DP-FFT? Our results, both theoretical and empirical, suggest that at low total privacy budget, one should allocate more of the total privacy budget to DP-LP.

### 5.1   Results under Zeroth-order Approximation

We first show how to allocate privacy budget to avoid the feature distortion analyzed in Section 3, using the zeroth-order approximation.

**Theorem 5.1** (Estimated privacy budget allocated to DP-LP). *If Assumption 3.1 and Assumption 3.2 hold at $t = 0$, then for any $\rho \in (0,1)$, with probability $(1-\rho)^h$, we can avoid feature distortion by spending*

$$r \propto \sigma^4 \sqrt{\ln(2/\rho)} \tag{27}$$

*amount $r$ of privacy budget on DP-LP, where $\sigma$ is the noise multiplier. That is, we ensure that $\forall j \in [h]$, and any $t > 0$ after DP-LP,*

$$\frac{\partial}{\partial t} \cos\left(w_j, \bar{x}_{c(j)}\right)\bigg|_t \geq 0 \tag{28}$$

According to Theorem 5.1, a greater proportion of the privacy budget should be allocated to DP-LP when the total privacy budget is smaller.

### 5.2   Results without approximation (2-layer linear network)

Complementing the result of Section 5.1, we use the 2-layer linear model of Section 4.1.1 to show that DP-LP-FFT may work better in some settings than linear probing or full fine-tuning alone. Linear probing first can accelerate fine-tuning by aligning the linear head. The following result provides a convergence bound for DP-LP-FFT when we linear-probe for time $t_{\mathrm{lp}}$, and then fully fine-tune for time $t$.

**Proposition 5.2** (Convergence of DP-LP-FFT). *Suppose we randomly initialize the linear head $v_0 \sim \mathcal{N}(0, \beta I_{h \times h})$ and Assumption E.7 hold. We run linear probing for time $t_{\mathrm{lp}}$ and then fine-tuning (Equation equation 127) for time $t$, then the loss is upper bounded by:*

$$\mathbb{E}[\mathcal{L}(t)] \leq \mathbb{E}[\mathcal{L}_{\mathrm{lp}}]e^{-At} + L^{\square}(1 - e^{-At}) \tag{29}$$

*where $\mathcal{L}_{\mathrm{lp}}$ is the expected loss after linear probing, $A = h\beta - 1 - \sqrt{2}\sigma^2(1 + d_x)$, and $L^{\square} = \sigma^2 \frac{(1+d_x)\|X^T Y\|_2 + d_x}{A}$. The coefficient $A = \mathbb{E}[\lambda_{\max}(D)] > 0$ increases as $t_{\mathrm{lp}}$ increases when we run linear probing in a finite time interval $t_{\mathrm{lp}} < \ln\left[3 + \frac{h(\sigma^2 - \beta)}{\|W_0^\top X^T Y\|_2^2}\right]$.*

**Corollary 5.3.** *Suppose we randomly initialize the linear head $v_0 \sim \mathcal{N}(0, \beta I_{h \times h})$ and Assumption E.7 hold. Then the two-phase method, first-linear-probing-then-finetuning (LP-FFT), could achieve a tighter loss upper bound than linear probing or fine-tuning in expectation if we first run linear probing for $t_{\mathrm{lp}} < \ln\left[3 + \frac{h(\sigma^2 - \beta)}{\|W_0^\top X^T Y\|_2^2}\right]$.*

Corollary 5.3 suggests that when we fix other hyperparameters (e.g. the total training time $T$), the performance of LP-FFT depends on the noise scale $\sigma$. If $\sigma$ is large enough such that $T < \ln\left[3 + \frac{k(\sigma^2 - \beta)}{\|B_0 X^T Y\|_2^2}\right]$, then LP may be the best; if $\sigma$ is small enough such that $\ln\left[3 + \frac{k(\sigma^2 - \beta)}{\|B_0 X^T Y\|_2^2}\right] \leq 0$, then FT may be the best; LP-FT could achieve the best performance when the noise scale is in a proper interval $\sigma^2 \in \left(\beta - 2\frac{\|B_0 X^T Y\|_2^2}{k}, \beta + (e^T - 3)\frac{\|B_0 X^T Y\|_2^2}{k}\right)$.

In our theory without approximation, these predictions are based only on upper bounds, so we cannot conclusively say that any fine-tuning approach outperforms another. Nonetheless, our theoretical results in two approaches suggest that the smaller the total budget, the more privacy budget should be allotted to DP-LP.

## 5.3 Experiments

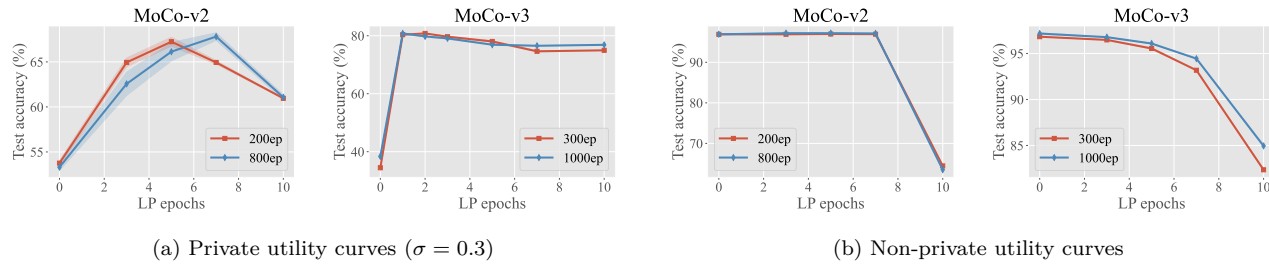

(a) Private utility curves ($\sigma = 0.3$)  (b) Non-private utility curves

Figure 6: Utility curves for pretraining on ImageNet-1K and fine-tuning on CIFAR-10 over ResNet-50, with pretrained features from MoCo-v2 and MoCo-v3 (Chen et al., 2020b; Chen* et al., 2021). We compare the performance from pre-trained weights of different pre-training epochs (200/800 epochs for MoCo-v2, 300/1k epochs for MoCo-v3). The x-axis sweeps the number of LP epochs from 0 to 10; the remaining epochs (out of 10) use FFT.

To illustrate the privacy budget trade-off, we empirically evaluate the benefits of DP-LP-FFT on real data and architectures. For experiments in Table 1 and Table 2, we use clipping thresholds C=0.1 and C=1, use batch size 1000 and sweep over learning rates {9, 5, 1, 0.5, 0.2, 0.15, 0.1, 0.05, 0.025}. These values are based on established empirical studies that explore optimal clipping thresholds for DP-SGD. In particular, Appendix B.1 of De et al. (2022) provides an in-depth analysis of clipping norms, concluding with the choice of C=1 for their primary experiments. Our experimental settings also draw from the methodologies outlined in Tang et al. (2023).

**DP-LP-FFT outperforms other fine-tuning methods: Pre-training on synthetic data.** We follow the setup in Tang et al. (2023) and generate utility curves for $\epsilon = 1, 2, 3$ (Figure 2b). We pre-train WideResNet with synthetic images generated from StyleGAN-oriented (Baradad et al., 2021) , and fine-tune it with DP-SGD on CIFAR-10. The x-axis sweeps the fraction of privacy budget allocated to DP-LP, and the remaining budget is used for DP-FFT. We find that at various privacy levels, DP-LP-FFT gives a clear advantage over either DP-FFT or DP-LP alone.

Figure 2b presents a different trend from our theoretical prediction, where we expect the optimal budget ratio for DP-LP to increase as the privacy noise grows. A possible intuitive explanation is that, in the Figure 2b experiments, the pre-training data is synthetic, making it 'distant' from the CIFAR-10 fine-tuning data distribution. This divergence may violate our assumption that the pre-trained weights $w_j$ are well-aligned with the fine-tuning data $x_i$.

**DP-LP-FFT outperforms other fine-tuning methods: Pre-training on ImageNet-1K.** Figure 6 illustrates the utility curves on ResNet-50 for $\sigma = 0, 0.3$. Here we fix $\sigma$ and vary $e_{LP}$ to trace the full utility

curve predicted by Corollary 5.3; Table 1 instead varies $\sigma$ (hence $\epsilon$) at a fixed $e_{LP} = 5$. [1]. To demonstrate utility curves for DP-LP-FFT, we vary the number of epochs of linear probing from $e_{LP} = 0$ to $e_{LP} = 10$; all remaining epochs (out of 10 total) are allocated to full fine-tuning, i.e., $e_{FFT} = 10 - e_{LP}$. Note that full fine-tuning corresponds to $e_{LP} = 0$ (the leftmost point of our subplots), and linear probing corresponds to $e_{LP} = 10$. We observe that for non-private optimization (Figure 6b), full fine-tuning achieves the highest test accuracy. However, for DP-SGD (Figure 6a), linear probing outperforms full fine-tuning, and DP-LP-FFT outperforms both DP-LP and DP-FFT.

| **Model** | ResNet$_{18}$ | | | MobileNet$_{v3}$ | | | Transformer$_{DeiT}$ | | |
|---|---|---|---|---|---|---|---|---|---|
| $\epsilon$ | $\infty$ | 1.29 | 0.57 | $\infty$ | 1.29 | 0.57 | $\infty$ | 1.29 | 0.26 |
| LP | $68.54_{0.02}$ | $67.90_{0.12}$ | $66.60_{0.04}$ | $71.12_{0.31}$ | $69.54_{0.08}$ | $67.32_{0.03}$ | $95.74_{0.04}$ | $93.61_{0.08}$ | $94.21_{0.08}$ |
| LP-FFT | $72.66_{0.12}$ | $68.65_{0.08}$ | $59.79_{1.03}$ | $71.30_{0.11}$ | $71.18_{0.06}$ | $66.94_{0.08}$ | $96.82_{0.08}$ | $93.66_{0.15}$ | $93.62_{0.05}$ |
| FFT | $73.69_{0.03}$ | $59.79_{1.03}$ | $53.82_{0.37}$ | $77.02_{0.31}$ | $63.06_{0.05}$ | $45.12_{0.07}$ | $96.17_{0.08}$ | $90.31_{0.53}$ | $84.19_{0.82}$ |

Table 1: Test accuracies of DP-LP, DP-LP-FFT, and DP-FFT on various architectures.

**Comparing DP fine-tuning methods.** As suggested by Theorem 5.1 and Corollary 5.3, as the noise scale $\sigma$ increases, the best fine-tuning strategy changes from DP-FFT (small $\sigma$, low privacy regime) to DP-LP-FFT, to DP-LP (large $\sigma$, high privacy regime). To qualitatively test this prediction, we sweep over different noise scales $\sigma$ and fix other hyperparameters in each benchmark and model architecture. We sort the rows by the number of parameters of each model and the noise scale in an ascending order. For each experiment setting, we report average test accuracies with standard errors. As expected, among the three fine-tuning methods (Table 1), DP-FFT almost always does the best under small noise scales (including the non-private setting where $\sigma = 0$), DP-LP-FFT does the best under moderate noise scales, and DP-LP does the best under large noise scales. The close non-DP ($\epsilon$) performance of FFT and LP-FFT on transformer architectures is consistent with previous observations in Kumar et al. (2022, Table 1).

| Transformer$_{DeiT}$ | | | | | |
|---|---|---|---|---|---|
| $\epsilon$ | $\infty$ | 12.28 | 1.29 | 0.57 | 0.26 |
| LP | $95.81_{0.05}$ | $95.55_{0.05}$ | $94.80_{0.06}$ | $94.21_{0.08}$ | $92.48_{0.27}$ |
| LP-LoRA | $96.2_{0.05}$ | $95.90_{0.03}$ | $94.81_{0.08}$ | $94.18_{0.05}$ | $91.99_{0.19}$ |
| LoRA | $96.26_{0.05}$ | $95.50_{0.06}$ | $94.76_{0.08}$ | $93.05_{0.09}$ | $91.28_{0.43}$ |

Table 2: Test accuracies of LP, LP-LoRA, LoRA on Transformer$_{DeiT}$.

**More experiments on parameter-efficient fine-tuning (PEFT) methods.** We conduct experiments with another fine-tuning trick: differentially private LoRA (Hu et al., 2022a). We run experiments on the Mini-DeiT-Ti architecture, where we use LoRA instead of full fine-tuning. In these experiments (Table 2), our batch size is 1000, and our LoRA rank is set to 8. We observe the same trend as what we saw for full fine-tuning; namely, as we increase the noise scale (i.e., as we reduce epsilon, giving a stronger privacy guarantee), it becomes more beneficial to use LP-LoRA or even just LP.

## 6 Conclusion and Discussion

We characterize the training dynamics of DP fine-tuning under a simplified theoretic setup (2-layer neural networks, separable datasets with -1/1 labels) using a Langevin diffusion-based approximation of DP-SGD, with an asymptotic expansion of random perturbations in dynamical systems as an approximation for Langevin diffusion. Our theory identifies and explains the phenomenon of representation distortion and alignment during DP fine-tuning, which we confirm empirically. Our work takes a step towards understanding how different private fine-tuning strategies can be mixed to improve performance, which could be useful

---

[1]The model performance is compromised because we replace the BatchNorm (Ioffe & Szegedy, 2015) in the pre-trained weights with GroupNorm (Wu & He, 2018). BatchNorm relies on batch statistics, which conflicts with the principles of differential privacy.

for designing or mixing other strategies, such as memory-efficient zeroth-order optimization with differential privacy (Zhang et al., 2024a).

**Limitations and open questions** There are several open questions we cannot cover in this work, such as generalizing our results to multi-layer neural networks with our approximation technique, the effect of other loss functions on the fine-tuning dynamics, and loss lower bounds for DP-LP/FFT without the zeroth-order approximation. Moreover, it is unclear how to apply our theory to other fine-tuning methods like LoRA (Hu et al., 2022b), as well as generative models for which neural collapse does not happen. Understanding whether the zeroth-order approximation can facilitate analysis in these settings is an interesting and important question for future work.

**Reproducibility Statement.** We have included full proofs for all theoretical results and sufficient experimental details in appendices to reproduce our results. We will also release our code under a permissive open-source license upon acceptance.

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

## A Additional experiment results

In this section, we provide more experiment results and detailed configurations.

**Evaluations back in the pre-training distribution (Figure 7).** We also evaluate the feature quality on ImageNet1-K, the pre-training dataset. The representation alignment for the pre-training domain is different: once a proper alignment is achieved, the backbone gradually recovers a portion of its original feature quality, which had been compromised due to DP noise and distribution-shift.

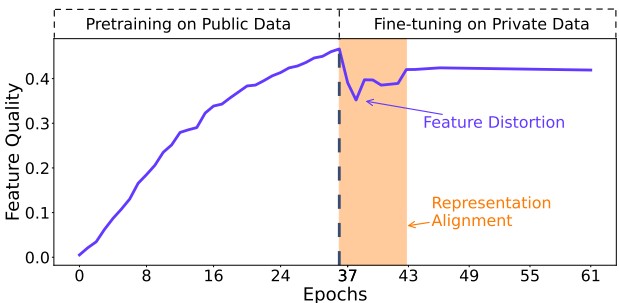

Figure 7: Backbone feature quality evaluated by average top-1 kNN accuracy on the pre-training dataset, for ResNet-50, through public pre-training on ImageNet-1K and differentially private fine-tuning on STL-10.

**Experiment setup in Table 1.** We use batch size 1000, clipping thresholds C=0.1 and C=1, and sweep over a range of learning rates $\{9, 5, 1, 0.5, 0.2, 0.15, 0.1, 0.05, 0.025\}$.

**Summary of experiment configurations.** We run experiments on five deep learning models and four transfer learning benchmarks to verify if our theoretical prediction, the existence of concave utility curves, generalizes to deep neural networks and real datasets. Each experimental setting comprises: (1) a model architecture, (2) a (larger) dataset for public pretraining, and (3) a (smaller) dataset as the private data for fine-tuning. The benchmarks we use are:

- ImageNet-1K→CIFAR-10. ImageNet-1K is a large-scale dataset. We initialize pretrained features of ResNet-50 from MoCo-v2 Chen et al. (2020b) and MoCo-v3 Chen* et al. (2021), trained on ImageNet-1K Russakovsky et al. (2015) without privacy. We then privately fine-tune the ResNet-50 on CIFAR-10.

- ImageNet-1K→STL-10. We pretrain a DeiT model on ImageNet then pretrain a Mini-DeiT-Ti model with weight distillation from the DeiT model Touvron et al. (2021); Zhang et al. (2022). After that, we privately fine-tune the Mini-DeiT-Ti model on STL-10 Coates et al. (2011) for 20 epochs.

- CIFAR-10→STL-10. We pretrain the feature extractor on CIFAR-10 Krizhevsky (2009) using stochastic gradient descent without privacy mechanisms. Then we finetune the pretrained features and a randomly initialized linear head on STL-10. This benchmark has been studied in the context of domain adaptation French et al. (2018); Kumar et al. (2022). The training subset of STL-10 only contains 500 images. To align with the small scale fine-tuning data, we run the experiments with smaller and data-efficient models: MobileNet-v3 and ResNet-18.

- RandP→CIFAR-10. To reproduce the results of Tang et al. (2023) and verify the general existence of concave utility curves, we also consider a slightly non-standard pretraining protocol. We pretrain a wide residual network (WRN) Zagoruyko & Komodakis (2016) on synthetic images generated by random diffusion processes. We follow the settings in Tang et al. (2023).

We employ early stopping, and select the optimal learning rate based on the accuracy of the in-distribution validation.

## A.1 Privacy-utility curves

We further plot the privacy-utility curves to aid the information in Table 1.

As expected, accuracy increases with epsilon for every method and backbone, and the results generally (but not always) qualitatively match our theoretical predictions.

For Mini-DeiT-Ti ,the ViT-style backbone is comparatively robust. DP-LP-FFT retains the lead in high epsilon regimes while DP-LP wins for small epsilons, as predicted by our theory.

For MobileNet-v3 and ResNet-18, the cross-over pattern is different from Mini-DeiT: even at moderate epsilon, DP-LP-FFT outperforms DP-LP, and under strong privacy DP-LP is best. And DP-FFT retains the lead over the high epsilon regime. This suggests that small conv-nets are more prone to head-induced distortion, so the front-loading budget into LP pays off sooner.

With a deeper conv-net, the trends predicted by our theory persist: DP-FFT wins at large epsilon, DP-LP-FFT at moderate epsilon, DP-LP at small epsilon. The DP-LP-FFT curve sits close to DP-FFT in the high-epsilon regime (no downside when noise is small) yet clearly exceeds it as epsilon shrinks, which is exactly the "mitigate-then-fine-tune" behavior predicted by Theorem 3.3 and Theorem 3.4.

## A.2 Explanation on side examples

Figure 2b follows Tang et al. (2023) protocol, which introduces EMA smoothing and gradient-averaging across augmentations before clipping. These two ingredients are absent from our theoretical setup, and these modifications dampen the representation-distortion predicted by Theorem 3.3. Our interpretation of Figure 2b is currently heuristic and is an early-stage conjecture rather than a formally proved result.

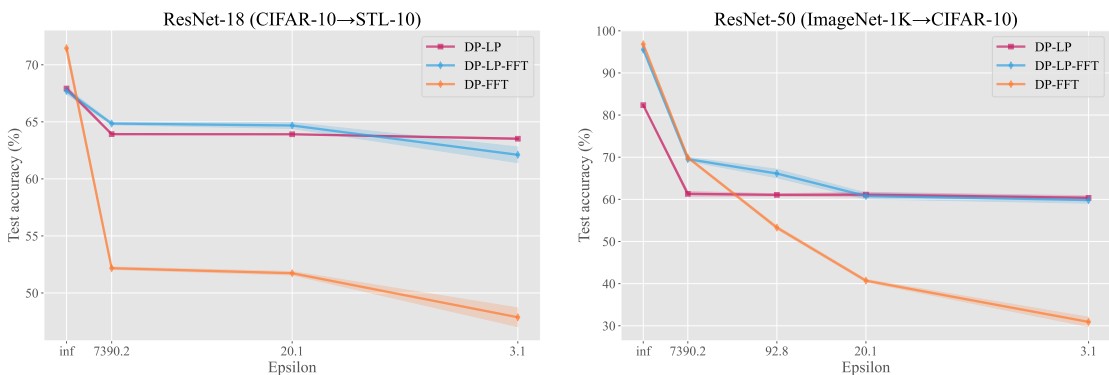

(a) ResNet architectures

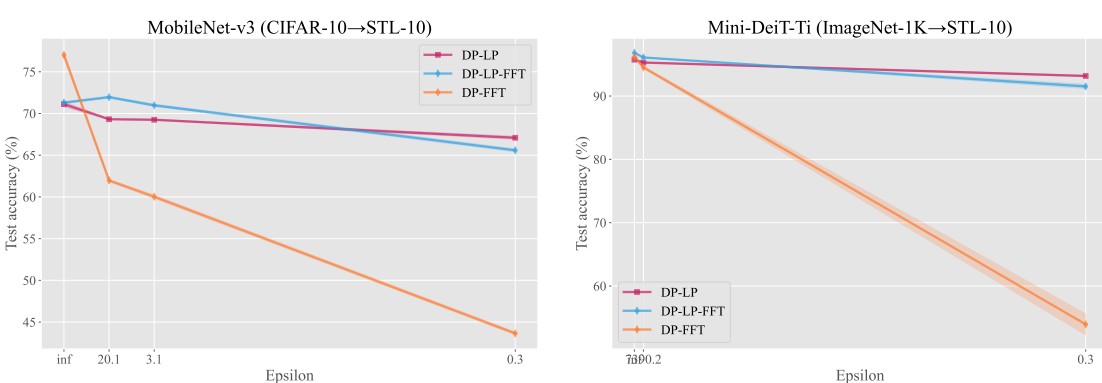

(b) Other architectures

Figure 8

1. EMA: Tang et al. (2023) maintain an EMA copy of the network parameters and report accuracy with that averaged model. EMA acts as a low-pass filter on the parameter trajectory, effectively smoothing out the rapid weight adjustments induced by the large initial head-gradient. This could delay the transient distortion our theory attributes to the first few DP-FFT steps.

2. Gradient averaging over augmentations: Before per-example clipping, Tang et al. (2023) average the gradients of multiple augmentations of the same image. Averaging reduces variance and shrinks the expected norm of each per-example gradient, lowering the probability that the clipping threshold is hit. Consequently, the random-initialisation error injected by the head could have a smaller effective magnitude. This potentially mitigates the early distortion phase.

## B  Technical results

**Lemma B.1** (Holder's inequality for sums). *For a sequence $x = [x_i]_{i=1}^n$ of positive real numbers and $p > 0$, define $\|x\|_p := (\sum_{i=1}^n x_i^p)^{1/p}$. Then for any pair of positive real numbers $p > 0, q > 0$ with $\frac{1}{p} + \frac{1}{q} = 1$, and any pair of sequence of positive real numbers $x$ and $y$,*

$$\|xy\|_1 \leq \|x\|_p \|y\|_q$$

**Lemma B.2** (Reverse Holder's inequality for sums). *For a sequence $x = [x_i]_{i=1}^n$ of positive real numbers and $p > 0$, define $\|x\|_p := (\sum_{i=1}^n x_i^p)^{1/p}$. Then for any pair of positive real numbers $p > 0, q > 0$ with $\frac{1}{p} - \frac{1}{q} = 1$, and any pair of sequence of positive real numbers $x$ and $y$,*

$$\|xy\|_1 \geq \|x\|_p \|y\|_{-q}$$

**Lemma B.3** (Reverse QM-AM inequality for sums). *For a sequence $x = [x_i]_{i=1}^n$ of positive real numbers,*

$$\left(\sum_{i=1}^n x_i\right)^2 \geq \sum_{i=1}^n x_i^2$$

**Lemma B.4** ($\mu$-coherent data conic hull (Min et al., 2024, Lemma 5)). *Define a conic hull $K := \mathcal{CH}(\{y_i x_i : i \in [n]\}) = \{\sum_{i=1}^n a_i y_i x_i : \forall a_i \geq 0, i \in [n]\}$. If Assumption 3.1 holds, i.e. the dataset is separable, then $K$ is $\mu$-coherent:*

$$\forall z_1, z_2 \in K \backslash \{0\}, \quad \cos(z_1, z_2) \geq \mu$$

**Corollary B.5** (Orthogonally separable $\implies$ linearly separable (Min et al., 2024)). *If Assumption 3.1 holds, then $\exists \gamma > 0$ and $z \in \mathbb{S}^{D-1}$ such that*

$$\forall i \in [n], \quad y_i \langle z, x_i \rangle \geq \gamma$$

*Proof of Corollary B.5.* We prove the existence statement by picking a valid pair of $z, \gamma$. Take $z := \frac{y_1 x_1}{\|x_1\|_2}$. Then $\forall i \in [n]$,

$$
\begin{aligned}
y_i \langle z, x_i \rangle =& \|x_i\|_2 \cos(y_1 x_1, y_i x_i) \\
& //\text{by Lemma B.4} \\
\geq& \|x_i\|_2 \mu \\
\geq& \mu \cdot \min_{i \in [n]} \|x_i\|_2
\end{aligned}
$$

Therefore $\gamma = \mu \cdot \min_{i \in [n]} \|x_i\|_2$. □

### B.1  Relaxed assumptions

We relax Assumption 3.1 by allowing non-zero cross-class correlation, controlled by a parameter $\rho[0, 1)$, and we relax Assumption 3.2 by allowing bounded activation leakage of a feature $w_j$ onto the opposite class, also controlled by $\rho$ (setting $\rho = 0$ recovers the original assumptions).

**Assumption B.6** (Relaxed data correlation)**.** Let $\bar{x}_c$ be the class means defined in the paper. There exists $\mu_{\text{in}}$ and $\rho \in [0, 1)$ such that for all $i \neq j$,

$$\text{(within class) } y_i = y_j \implies \frac{\langle x_i, x_j \rangle}{\|x_i\|\|x_j\|} \geq \mu_{\text{in}}, \tag{30}$$

$$\text{(across class) } y_i \neq y_j \implies \frac{\langle x_i, x_j \rangle}{\|x_i\|\|x_j\|} \leq \rho\mu_{\text{in}} \tag{31}$$

Equivalently, the (label-signed) pairwise cosine similarity has a positive gap

$$\inf_{y_i=y_j} \cos(x_i, x_j) - \sup_{y_i \neq y_j} \cos(x_i, x_j) \geq (1-\rho)\mu_{\text{in}} > 0. \tag{32}$$

This weakens Assumption 3.1, which enforced a sign separation, to a gap separation that permits some positive cross-class correlation. The original Assumption 3.1 and its cone construction.

**Assumption B.7** (Relaxed neural feature collapsing)**.** Let $c(j) \in \{+1, -1\}$ be the class index associated with feature $w_j$ (same convention as Assumption 3.2). Define the "activated mass" at t=0 for $w_j$ under the exponential loss weights $\ell_i = \exp(-y_i f(x_i))$:

$$A_j^+ = \sum_{i:y_i=c(j)} \ell_i(0)\mathbf{1}\{w_j(0)^\top x_i > 0\}, \tag{33}$$

$$A_j^- = \sum_{i:y_i \neq c(j)} \ell_i(0)\mathbf{1}\{w_j(0)^\top x_i > 0\} \tag{34}$$

Assume leakage is bounded by the same $\rho$ above and below:

$$\forall j, \ A_j^- \leq \rho A_j^+ \tag{35}$$

And the pre-trained features are not well aligned yet with the downstream data, i.e. we need to fine-tune the features. We describe this by an upper bound upon the alignment

$$\cos(\bar{x}_{c(j)}, w_j) < \mu_{\text{in}}(1 - \rho^2). \tag{36}$$

This is a quantitative relaxation of Assumption 3.2 (the old statement implied $A_j^- = 0$).

Then we show that, based on the relaxed assumptions, we can similarly prove a similar result to Theorem 3.3.

**Theorem B.8** (Random initialization causes feature distortion)**.** *If Assumption B.6 and Assumption B.7 hold at $t = 0$, then for each $j$,*

$$\frac{d}{dt}\cos(w_j, \bar{x}_{c(j)})\Big|_{t=0} = v_j(0)\Gamma_j(0), \tag{37}$$

*with the positive lower bound*

$$\Gamma_j(0) \geq 2\langle w_j, \bar{x}_{c(j)}\rangle A_j^+(\mu_{\text{in}}(1-\rho^2) - \cos(\bar{x}_{c(j)}, w_j)). \tag{38}$$

*In particular, if $v_j(0) < 0$ then $\frac{d}{dt}\cos(w_j, \bar{x}_{c(j)})\Big|_{t=0} < 0$. By continuity of the Langevin dynamics, there exists $\Delta t > 0$ such that*

$$\frac{d}{dt}\cos(w_j, \bar{x}_{c(j)})\Big|_{t=0} > 0, \quad \forall t \in (0, \Delta t). \tag{39}$$

*Since $v_0 \sim \mathcal{N}(0, \beta I_h)$, with probability at least $1 - 2^{-h}$ there exists some $j$ with $v_j(0) < 0$.*

Hence early-stage feature distortion occurs with the same high probability as in Theorem 3.3, now with strength scaled by the factor $(1 - \rho)$. Setting $\rho = 0$ recovers exactly the sign identity used in the proof of Theorem 3.3. The bound is monotone in the leakage: larger $\rho$ weakens but does not flip the sign as long as $\rho < 1$.

*Proof of Theorem B.8.*

1. **Zeroth order DP-FFT dynamics**. For $j$-th head/backbone pair, at $t = 0$ the zeroth order ODE gives

$$\dot{w}_j = \sum_{i \in [n]} y_i \ell_i(0) v_j(0) \mathbf{1}\{w_j(0)^\top x_i > 0\} x_i \tag{40}$$

$$= v_j(0) \sum_{i \in [n]} y_i \ell_i(0) \mathbf{1}\{w_j(0)^\top x_i > 0\} x_i \tag{41}$$

$$=: v_j(0) Z_j \tag{42}$$

where $Z_j$ is defined as the activated, label-signed data aggregate.

2. **Derivative of the cosine**. Using the exact identity for the time derivative of $\cos(w_j, \bar{x}_{c(j)})$,

$$\frac{d}{dt} \cos(w_j, \bar{x}_{c(j)}) \Big|_{t=0} = \frac{2\langle w_j, \bar{x}_{c(j)} \rangle}{\|w_j\|_2^2} \langle S_j, \dot{w}_j \rangle = \frac{2\langle w_j, \bar{x}_{c(j)} \rangle}{\|w_j\|_2^2} v_j(0) \langle S_j, Z_j \rangle, \tag{43}$$

$$S_j := \|w_j\|^2 \bar{x}_{c(j)} - \langle \bar{x}_{c(j)}, w_j \rangle w_j, \tag{44}$$

so that $\langle S_j, w_j \rangle = 0$ and $\langle S_j, \bar{x}_{c(j)} \rangle = \|w_j\|^2 \|\bar{x}_{c(j)}\|^2 - \langle \bar{x}_{c(j)}, w_j \rangle^2 \geq 0$. From a geometric perspective, $S_j$ define the component of $\bar{x}_{c(j)}$ orthogonal to $w_j$.

3. **Lower bound** $\langle S_j, Z_j \rangle$.

$$\langle S_j, Z_j \rangle = \sum_{y_i = c(j)} \ell_i(0) \mathbf{1}\{w_j^\top x_i > 0\} \langle x_i, S_j \rangle - \sum_{y_i \neq c(j)} \ell_i(0) \mathbf{1}\{w_j^\top x_i > 0\} \langle x_i, S_j \rangle \tag{45}$$

$$= \|w_j\|^2 \underbrace{\left( \sum_{y_i = c(j)} \ell_i(0) \mathbf{1}\{w_j^\top x_i > 0\} \langle x_i, \bar{x}_{c(j)} \rangle - \sum_{y_i \neq c(j)} \ell_i(0) \mathbf{1}\{w_j^\top x_i > 0\} \langle x_i, \bar{x}_{c(j)} \rangle \right)}_{T_a} \tag{46}$$

$$- \langle \bar{x}_{c(j)}, w_j \rangle \underbrace{\left( \sum_{y_i = c(j)} \ell_i(0) \mathbf{1}\{w_j^\top x_i > 0\} \langle x_i, w_j \rangle - \sum_{y_i \neq c(j)} \ell_i(0) \mathbf{1}\{w_j^\top x_i > 0\} \langle x_i, w_j \rangle \right)}_{T_b} \tag{47}$$

- Term $T_a$. By Assumption B.6 and Assumption B.7,

$$T_a \geq \mu_{\text{in}}(A_j^+ - \rho A_j^-) \geq \mu_{\text{in}}(1 - \rho^2) A_j^+ \tag{48}$$

- Term $T_b$.

$$T_b \leq \|w_j\| A_j^+ \tag{49}$$

Combine the two bounds to get

$$\langle S_j, Z_j \rangle \geq \|w_j\|^2 \mu_{\text{in}}(1 - \rho^2) A_j^+ - \langle \bar{x}_{c(j)}, w_j \rangle \|w_j\| A_j^+ \tag{50}$$

$$\geq \|w_j\|^2 A_j^+ (\mu_{\text{in}}(1 - \rho^2) - \cos(\bar{x}_{c(j)}, w_j)) \tag{51}$$

$$//\text{by Assumption B.7,} \tag{52}$$

$$> 0. \tag{53}$$

Consequently, $\frac{d}{dt} \cos(w_j, \bar{x}_{c(j)})$ has the same sign as $v_j(0)$.

$\square$

# C  Appendix: Representation alignment

## C.1  Theory

The Langevin diffusion of $w_j$ on a $n$-sized data cluster ($j \in [h]$) is

$$\dot{w}_j = \sum_{i=1}^{n} y_i \exp(-y_i f(x_i; W, v)) v_j \mathrm{relu}'(w_j^\top x_i) x_i + \sigma \partial Q_t, \tag{54}$$

where $Q_t$ is a vector containing $D$ independent 1-dimensional Brownian motion.

The Langevin diffusion of $v$ on a $n$-sized data cluster is

$$\dot{v} = \sum_{i=1}^{n} y_i \exp(-y_i f(x_i; W, v)) \mathrm{relu}(W^\top x_i) + \sigma \partial Q_t,$$

where $Q_t$ is a vector containing $h$ independent 1-dimensional Brownian motion.

We rewrite the Langevin diffusion by asymptotic expansion (Freidlin et al., 2012, Equation 2.1, Chapter 2.2),

$$\begin{cases} v_j \approx v_j^{(0)} + \sigma v_j^{(1)} + \cdots \\ w_j \approx w_j^{(0)} + \sigma w_j^{(1)} + \cdots, \end{cases} \tag{55}$$

i.e. we expand the Langevin diffusion as a linear combination of the original gradient flow and a linear stochastic diffusion.

$$\begin{cases} \dot{v}_j^{(0)} = \sum_{i=1}^{n} y_i \exp(-y_i f(x_i; W^{(0)}, v^{(0)})) \mathrm{relu}((w_j^{(0)})^\top x_i) \\ \dot{w}_j^{(0)} = \sum_{i=1}^{n} y_i \exp(-y_i f(x_i; W^{(0)}, v^{(0)})) v_j^{(0)} \mathrm{relu}'((w_j^{(0)})^\top x_i) x_i. \end{cases} \tag{56}$$

**Lemma C.1** (Zeroth order invariance of locally linearized LD). *If we rewrite the Langevin diffusion by asymptotic expansion (Freidlin et al., 2012, Equation 2.1, Chapter 2.2),*

$$\begin{cases} v_j \approx v_j^{(0)} + \sigma v_j^{(1)} \\ w_j \approx w_j^{(0)} + \sigma w_j^{(1)}. \end{cases}$$

*then the layer invariance still holds for zeroth order approximation*

$$\frac{d}{dt}[(v_j^{(0)})^2 - \|w_j^{(0)}\|_2^2] = 0. \tag{57}$$

*This result is similar to the imbalance matrix in gradient flow (Arora et al., 2018; Du et al., 2018; Min et al., 2023a).*

We are ready to prove Theorem 3.3.

*Proof of Theorem 3.3.* The explicit expression of the cosine value is

$$\cos(w_j, \bar{x}_{c(j)}) = \frac{w_j^\top \bar{x}_{c(j)}}{\|w_j\|_2 \|\bar{x}_{c(j)}\|_2} \tag{58}$$

Without loss of generality, let $\|\bar{x}_{c(j)}\|_2 = 1$. To show that the cosine value decreases with high probability, we only need to prove that the derivative of $\frac{(w_j^\top \bar{x}_{c(j)})^2}{\|w_j\|_2^2}$ is negative at $t = 0$ with high probability. The explicit derivative expression is

$$\frac{\partial}{\partial t} \cos(w_j, \bar{x}_{c(j)}) = \frac{2(w_j^\top \bar{x}_{c(j)})}{\|w_j\|_2^2} \left[ \|w_j\|_2^2 \bar{x}_{c(j)}^\top \frac{\partial w_j}{\partial t} - \bar{x}_{c(j)}^\top w_j w_j^\top \frac{\partial w_j}{\partial t} \right] \tag{59}$$

$$= \frac{2(w_j^\top \bar{x}_{c(j)})}{\|w_j\|_2^2} \left[ \|w_j\|_2^2 \bar{x}_{c(j)} - (\bar{x}_{c(j)}^\top w_j) w_j \right]^\top \frac{\partial w_j}{\partial t} \tag{60}$$

$$// \text{by Assumption 3.2} \tag{61}$$

$$\text{sign}\left( \frac{\partial}{\partial t} \cos(w_j, \bar{x}_{c(j)}) \right) = \text{sign}\left( \left[ \|w_j\|_2^2 \bar{x}_{c(j)} - (\bar{x}_{c(j)}^\top w_j) w_j \right]^\top \frac{\partial w_j}{\partial t} \right) \tag{62}$$

$$= \text{sign}\left( v_j (\|w_j\|_2^2 - (\bar{x}_{c(j)}^\top w_j)^2) \right) \tag{63}$$

$$= \text{sign}(v_j) \tag{64}$$

Since we initialize $v \sim \mathcal{N}(0, \beta I_{h \times h})$, with probability $1 - 2^{-h}$, there exists $j$ such that $v_j < 0$ at $t = 0 \implies \frac{\partial}{\partial t} \cos(w_j, \bar{x}_{c(j)}) < 0$ at $t = 0$. By the continuity of the approximated Langevin diffusion, there exists $\Delta t > 0$ such that for any $t \in (0, \Delta t)$,

$$\frac{\partial}{\partial t} \cos(w_j, \bar{x}_{c(j)}) < 0. \tag{65}$$

$\square$

*Proof of Theorem 3.4.* In the proof of Theorem 3.3, we show that for $w_j \in S_c, c \in \{-1, 1\}$,

$$\text{sign}\left( \frac{\partial}{\partial t} \cos(w_j, \bar{x}_{c(j)}) \right) = \text{sign}(v_j) \cdot \text{sign}(c) \tag{66}$$

To mitigate the feature distortion after some time index $\Delta t$, we only need $c \cdot v_j > 0$. For DP-LP, every $\frac{\partial}{\partial t} v_j$ increases/decreases if $c = 1/-1$. Therefore, for any initialization, there exists $\Delta t$ such that $\text{sign}(v_j) = \text{sign}(c)$ after time index $\Delta t$. If we switch to DP-FFT after $\Delta t$, $\frac{\partial}{\partial t} \cos(w_j, \bar{x}_{c(j)}) > 0$ for any $j \in [h]$. Thus $\cos(w_j, \bar{x}_{c(j)})$ is non-decreasing in DP-FFT. $\square$

## D  Approximate convergence of DP-LP and DP-FFT

### D.1  Approximate DP-LP convergence

We add some extra notations for the following proofs:

- Positive data subset $\mathcal{I}_+ := \{i \in [n] : y_i > 0\}$

- Negative data subset $\mathcal{I}_- := \{i \in [n] : y_i < 0\}$

- Positive head cluster $\mathcal{V}_+(t) := \{j \in [h] : \text{sign}(v_j(t)) > 0\}$

- Negative head cluster $\mathcal{V}_-(t) := \{j \in [h] : \text{sign}(v_j(t)) < 0\}$

- Index function $\mathscr{I} : \mathbb{R}^D \to \{\mathcal{I}_+, \mathcal{I}_-\}$ maps feature vector to its cluster

$$\mathscr{I}(w) = \begin{cases} \mathcal{I}_+ & w \in S_+ \\ \mathcal{I}_- & w \in S_- \\ \emptyset & \text{otherwise} \end{cases}$$

We first derive the upper bound for approximate DP-LP.

*Upper bound proof of Theorem 4.2.* We construct a lower bound of the drift terms in the zeroth order approximation

$$\|\nabla_v \mathcal{L}^{(0)}\|_2^2 = \sum_{j=1}^h \left( \sum_{i=1}^n y_i \exp(-y_i f(x_i; W^{(0)}, v^{(0)})) \text{relu}((w_j^{(0)})^\top x_i) \right)^2 \tag{67}$$

$$= \sum_{j=1}^{h} \left( \sum_{i \in \mathscr{I}(w_j^{(0)})} y_i \exp(-y_i f(x_i; W^{(0)}, v^{(0)})) \text{relu}((w_j^{(0)})^\top x_i) \right)^2 \tag{68}$$

$$\geq \sum_{j=1}^{h} \left[ \min_{i \in \mathscr{I}(w_j^{(0)})} \text{relu}((w_j^{(0)})^\top x_i) \right]^2 \left( \sum_{i \in \mathscr{I}(w_j^{(0)})} y_i \exp(-y_i f(x_i; W^{(0)}, v^{(0)}))) \right)^2 \tag{69}$$

$$= \sum_{j=1}^{h} \left[ \min_{i \in \mathscr{I}(w_j^{(0)})} \text{relu}((w_j^{(0)})^\top x_i) \right]^2 \left( \sum_{i \in \mathscr{I}(w_j^{(0)})} \exp(-y_i f(x_i; W^{(0)}, v^{(0)}))) \right)^2 \tag{70}$$

$$= \sum_{j \in \mathcal{V}_+} \left[ \min_{i \in \mathcal{I}_+} \text{relu}((w_j^{(0)})^\top x_i) \right]^2 (\mathcal{L}_+^{(0)})^2 + \sum_{j \in \mathcal{V}_-} \left[ \min_{i \in \mathcal{I}_-} \text{relu}((w_j^{(0)})^\top x_i) \right]^2 (\mathcal{L}_+^{(0)})^2 \tag{71}$$

$$\geq \min \left\{ \sum_{j \in \mathcal{V}_+} \left[ \min_{i \in \mathcal{I}_+} \text{relu}((w_j^{(0)})^\top x_i) \right]^2, \sum_{j \in \mathcal{V}_-} \left[ \min_{i \in \mathcal{I}_-} \text{relu}((w_j^{(0)})^\top x_i) \right]^2 \right\} \left[ (\mathcal{L}_+^{(0)})^2 + (\mathcal{L}_-^{(0)})^2 \right] \tag{72}$$

$$\geq \frac{1}{2} \min \left\{ \sum_{j \in \mathcal{V}_+} \left[ \min_{i \in \mathcal{I}_+} \text{relu}((w_j^{(0)})^\top x_i) \right]^2, \sum_{j \in \mathcal{V}_-} \left[ \min_{i \in \mathcal{I}_-} \text{relu}((w_j^{(0)})^\top x_i) \right]^2 \right\} \left[ \mathcal{L}_+^{(0)} + \mathcal{L}_-^{(0)} \right]^2 \tag{73}$$

$$= \frac{1}{2} \min \left\{ \sum_{j \in \mathcal{V}_+} \left[ \min_{i \in \mathcal{I}_+} \text{relu}((w_j^{(0)})^\top x_i) \right]^2, \sum_{j \in \mathcal{V}_-} \left[ \min_{i \in \mathcal{I}_-} \text{relu}((w_j^{(0)})^\top x_i) \right]^2 \right\} (\mathcal{L}^{(0)})^2 \tag{74}$$

We construct an upper bound of the diffusion terms in the zeroth order approximation

$$\frac{1}{2} \sigma^2 \sum_{i=1}^{n} \ell(y_i, f(x_i; W^{(0)}, v^{(0)})) \|\text{relu}((W^{(0)})^\top x_i)\|_2^2$$

$$= \frac{1}{2} \sigma^2 \sum_{i=1}^{n} \left\{ \ell(y_i, f(x_i; W^{(0)}, v^{(0)})) \right\} \cdot \left\{ \|\text{relu}((W^{(0)})^\top x_i)\|_2^2 \right\}$$

//by Lemma B.1

$$\leq \frac{1}{2} \sigma^2 \left\{ \sum_{i=1}^{n} \ell^2(y_i, f(x_i; W^{(0)}, v^{(0)})) \right\}^{1/2} \cdot \left\{ \sum_{i=1}^{n} \|\text{relu}((W^{(0)})^\top x_i)\|_2^4 \right\}^{1/2}$$

//by Lemma B.3

$$\leq \frac{1}{2} \sigma^2 \left\{ \sum_{i=1}^{n} \ell(y_i, f(x_i; W^{(0)}, v^{(0)})) \right\} \cdot \left\{ \sum_{i=1}^{n} \|\text{relu}((W^{(0)})^\top x_i)\|_2^4 \right\}^{1/2}$$

$$= \frac{1}{2} \sigma^2 \mathcal{L}^{(0)} \cdot \left\{ \sum_{i=1}^{n} \|\text{relu}((W^{(0)})^\top x_i)\|_2^4 \right\}^{1/2}$$

Then we have an upper bound

$$\mathcal{L}^{(0)}(T) \leq \frac{1}{\frac{1}{\mathcal{L}^{(0)}(0)} e^{-BT} + \frac{A}{B}(1 - e^{-BT})}$$

where constants $A, B$ are defined as

$$\begin{cases} A = \frac{1}{2} \min \left\{ \sum_{j \in \mathcal{V}_+} \left[ \min_{i \in \mathcal{I}_+} \text{relu}((w_j^{(0)})^\top x_i) \right]^2, \sum_{j \in \mathcal{V}_-} \left[ \min_{i \in \mathcal{I}_-} \text{relu}((w_j^{(0)})^\top x_i) \right]^2 \right\} \\ B = \frac{1}{2} \sigma^2 \left\{ \sum_{i=1}^{n} \|\text{relu}((W^{(0)})^\top x_i)\|_2^4 \right\}^{1/2} \end{cases}$$

$\square$

**We give the lower bound of approxiamte DP-LP below. We first give a loose lower bound as a warm-up. Then we improve the techniques and provide a tight lower bound.**

*Loose lower bound proof of Theorem 4.2.* We rewrite the Langevin diffusion by asymptotic expansion (Freidlin et al., 2012, Equation 2.1, Chapter 2.2)

$$\dot{\mathcal{L}}^{(0)} = -\|\nabla_v \mathcal{L}^{(0)}\|_2^2 + \frac{1}{2}\sigma^2 \sum_{i=1}^n y_i^2 \ell(y_i, f(x_i; W^{(0)}, v^{(0)})) \|\text{relu}((W^{(0)})^\top x_i)\|_2^2$$

$$= -\|\nabla_v \mathcal{L}^{(0)}\|_2^2 + \frac{1}{2}\sigma^2 \sum_{i=1}^n \ell(y_i, f(x_i; W^{(0)}, v^{(0)})) \|\text{relu}((W^{(0)})^\top x_i)\|_2^2$$

$$\geq -\|\nabla_v \mathcal{L}^{(0)}\|_2^2 + \left(\min_{i \in \mathcal{V}_+^{(0)}} \|\text{relu}((W^{(0)})^\top x_i)\|_2^2\right) \cdot \frac{1}{2}\sigma^2 \sum_{i \in \mathcal{V}_+^{(0)}} \ell(y_i, f(x_i; W^{(0)}, v^{(0)}))$$

$$\quad + \left(\min_{i \in \mathcal{V}_-^{(0)}} \|\text{relu}((W^{(0)})^\top x_i)\|_2^2\right) \cdot \frac{1}{2}\sigma^2 \sum_{i \in \mathcal{V}_-^{(0)}} \ell(y_i, f(x_i; W^{(0)}, v^{(0)}))$$

$$= -\|\nabla_v \mathcal{L}^{(0)}\|_2^2 + \left(\min_{i \in [n]} \|\text{relu}((W^{(0)})^\top x_i)\|_2^2\right) \cdot \frac{1}{2}\sigma^2 \sum_{i \in [n]} \ell(y_i, f(x_i; W^{(0)}, v^{(0)}))$$

$$= -\|\nabla_v \mathcal{L}^{(0)}\|_2^2 + \left(\min_{i \in [n]} \|\text{relu}((W^{(0)})^\top x_i)\|_2^2\right) \cdot \frac{1}{2}\sigma^2 \mathcal{L}^{(0)}$$

$$= -\sum_{j=1}^h \left(\sum_{i=1}^n y_i \exp(-y_i f(x_i; W^{(0)}, v^{(0)}))\text{relu}((w_j^{(0)})^\top x_i)\right)^2 + \left(\min_{i \in [n]} \|\text{relu}((W^{(0)})^\top x_i)\|_2^2\right) \cdot \frac{1}{2}\sigma^2 \mathcal{L}^{(0)}$$

//by trapping

$$= -\sum_{j \in \mathcal{V}_+^{(0)}} \left(\sum_{i \in \mathcal{I}_+} \exp(-f(x_i; W^{(0)}, v^{(0)}))\text{relu}((w_j^{(0)})^\top x_i)\right)^2$$

$$\quad - \sum_{j \in \mathcal{V}_-^{(0)}} \left(\sum_{i \in \mathcal{I}_-} \exp(f(x_i; W^{(0)}, v^{(0)}))\text{relu}((w_j^{(0)})^\top x_i)\right)^2$$

$$\quad + \left(\min_{i \in [n]} \|\text{relu}((W^{(0)})^\top x_i)\|_2^2\right) \cdot \frac{1}{2}\sigma^2 \mathcal{L}^{(0)}$$

$$\geq -\left(\max_{j \in [h], i \in [n]} (\text{relu}((w_j^{(0)})^\top x_i))^2\right) \sum_{j \in \mathcal{V}_+^{(0)}} \left(\sum_{i \in \mathcal{I}_+} \exp(-f(x_i; W^{(0)}, v^{(0)}))\right)^2$$

$$\quad - \left(\max_{j \in [h], i \in [n]} (\text{relu}((w_j^{(0)})^\top x_i))^2\right) \sum_{j \in \mathcal{V}_-^{(0)}} \left(\sum_{i \in \mathcal{I}_-} \exp(f(x_i; W^{(0)}, v^{(0)}))\right)^2$$

$$\quad + \left(\min_{i \in [n]} \|\text{relu}((W^{(0)})^\top x_i)\|_2^2\right) \cdot \frac{1}{2}\sigma^2 \mathcal{L}^{(0)}$$

$//a^2 + b^2 \leq (a+b)^2$ when $a > 0, b > 0$

$$\geq -\left(\max_{j \in [h], i \in [n]} (\text{relu}((w_j^{(0)})^\top x_i))^2\right) \sum_{j \in [h]} \left(\sum_{i \in [n]} \exp(-f(x_i; W^{(0)}, v^{(0)}))\right)^2$$

$$+ \left( \min_{i\in[n]} \|\text{relu}((W^{(0)})^\top x_i)\|_2^2 \right) \cdot \frac{1}{2}\sigma^2 \mathcal{L}^{(0)}$$

$$\geq -h \left( \max_{j\in[h],i\in[n]} (\text{relu}((w_j^{(0)})^\top x_i))^2 \right) \left( \sum_{i\in[n]} \exp(-f(x_i; W^{(0)}, v^{(0)})) \right)^2 + \left( \min_{i\in[n]} \|\text{relu}((W^{(0)})^\top x_i)\|_2^2 \right) \cdot \frac{1}{2}\sigma^2 \mathcal{L}^{(0)}$$

$$\geq -h \left( \max_{j\in[h],i\in[n]} (\text{relu}((w_j^{(0)})^\top x_i))^2 \right) (\mathcal{L}^{(0)})^2 + \left( \min_{i\in[n]} \|\text{relu}((W^{(0)})^\top x_i)\|_2^2 \right) \cdot \frac{1}{2}\sigma^2 \mathcal{L}^{(0)}$$

In linear probing, the coefficients $h\left(\max_{j\in[h],i\in[n]}(\text{relu}((w_j^{(0)})^\top x_i))^2\right)$ and $\frac{1}{2}\sigma^2\left(\min_{i\in[n]}\|\text{relu}((W^{(0)})^\top x_i)\|_2^2\right)$ are constants. We replace them with dummy notation $A$ and $B$. We solve the first-order nonlinear ODE by turning it into a first-order linear ODE.

$$\dot{\mathcal{L}}^{(0)} \geq -A(\mathcal{L}^{(0)})^2 + B\mathcal{L}^{(0)}$$

$$\frac{1}{(\mathcal{L}^{(0)})^2}\dot{\mathcal{L}}^{(0)} \geq -A + B\frac{1}{\mathcal{L}^{(0)}}$$

$$-\frac{d}{dt}\left(\frac{1}{\mathcal{L}^{(0)}}\right) \geq -A + B\frac{1}{\mathcal{L}^{(0)}}$$

$$\mathcal{L}^{(0)}(T) \geq \frac{1}{\frac{1}{\mathcal{L}^{(0)}(0)}e^{-BT} + \frac{A}{B}(1 - e^{-BT})}$$

$$\square$$

*Remark* D.1 (On the qualitative properties of loose DP-LP lower bound). If we take the limit to initial point, then the lower bound degenerate to the initial loss value.

$$\lim_{t\to 0} \frac{1}{\frac{1}{\mathcal{L}^{(0)}(0)}e^{-BT} + \frac{A}{B}(1 - e^{-BT})} = \mathcal{L}^{(0)}(t=0) = \mathcal{L}(t=0) \tag{75}$$

If we take the limit to infinite time,

$$\lim_{t\to\infty} \frac{1}{\frac{1}{\mathcal{L}^{(0)}(0)}e^{-BT} + \frac{A}{B}(1 - e^{-BT})} = \frac{B}{A} = \frac{\frac{1}{2}\sigma^2\left(\min_{i\in[n]}\|\text{relu}((W^{(0)})^\top x_i)\|_2^2\right)}{h\left(\max_{j\in[h],i\in[n]}(\text{relu}((w_j^{(0)})^\top x_i))^2\right)} \tag{76}$$

the following interpretation holds:

1. For larger noise $\sigma \uparrow$, the lower bound is higher, i.e. worse performance.

2. For bad alignment between pretrained features $W^{(0)}$ and data points, both the denominator and the numerator could shrink. It is not obvious how the lower bound changes.

In the following result, we modify the proof, replace the $\min(\cdot)$, and provide a tighter bound.

*Tight lower bound proof of Theorem 4.2.* This is an alternative construction of a lower bound for drift terms in the zeroth order approximation

$$\|\nabla_v \mathcal{L}^{(0)}\|_2^2 = \sum_{j=1}^{h} \left( \sum_{i=1}^{n} y_i \exp(-y_i f(x_i; W^{(0)}, v^{(0)}))\text{relu}((w_j^{(0)})^\top x_i) \right)^2$$

$$= \sum_{j\in\mathcal{V}_+^{(0)}} \left( \sum_{i\in\mathcal{I}_+} \exp(-f(x_i; W^{(0)}, v^{(0)}))\text{relu}((w_j^{(0)})^\top x_i) \right)^2$$

$$+ \sum_{j \in \mathcal{V}_-^{(0)}} \left( \sum_{i \in \mathcal{I}_-} \exp(f(x_i; W^{(0)}, v^{(0)})) \mathrm{relu}((w_j^{(0)})^\top x_i) \right)^2$$

//by Lemma B.3

$$\leq \left( \sum_{j \in \mathcal{V}_+^{(0)}} \sum_{i \in \mathcal{I}_+} \exp(-f(x_i; W^{(0)}, v^{(0)})) \mathrm{relu}((w_j^{(0)})^\top x_i) \right)^2$$

$$+ \left( \sum_{j \in \mathcal{V}_-^{(0)}} \sum_{i \in \mathcal{I}_-} \exp(f(x_i; W^{(0)}, v^{(0)})) \mathrm{relu}((w_j^{(0)})^\top x_i) \right)^2$$

$$\leq \left( \sum_{j \in [h]} \sum_{i \in [n]} \exp(-f(x_i; W^{(0)}, v^{(0)})) \mathrm{relu}((w_j^{(0)})^\top x_i) \right)^2$$

$$= \left( \sum_{i \in [n]} \sum_{j \in [h]} \exp(-f(x_i; W^{(0)}, v^{(0)})) \mathrm{relu}((w_j^{(0)})^\top x_i) \right)^2$$

$$\leq \left( \sum_{i \in [n]} \left[ \max_{j \in [h]} \mathrm{relu}((w_j^{(0)})^\top x_i) \right] \exp(-f(x_i; W^{(0)}, v^{(0)})) \right)^2$$

//by Lemma B.1

$$\leq \left( \sum_{i \in [n]} \left[ \max_{j \in [h]} \mathrm{relu}((w_j^{(0)})^\top x_i) \right]^2 \right) \left( \sum_{i \in [n]} \exp(-f(x_i; W^{(0)}, v^{(0)}))^2 \right)$$

//by Lemma B.3

$$\leq \left( \sum_{i \in [n]} \left[ \max_{j \in [h]} \mathrm{relu}((w_j^{(0)})^\top x_i) \right]^2 \right) \left( \sum_{i \in [n]} \exp(-f(x_i; W^{(0)}, v^{(0)})) \right)^2$$

$$\leq \left( \sum_{i \in [n]} \left[ \max_{j \in [h]} \mathrm{relu}((w_j^{(0)})^\top x_i) \right]^2 \right) (\mathcal{L}^{(0)})^2$$

We replace the $A$ constant by $\sum_{i \in [n]} \left[ \max_{j \in [h]} \mathrm{relu}((w_j^{(0)})^\top x_i) \right]^2$. This is an alternative construction of a lower bound for diffusion-resulted terms in the zeroth order approximation

$$\frac{1}{2} \sigma^2 \sum_{i=1}^n \ell(y_i, f(x_i; W^{(0)}, v^{(0)})) \| \mathrm{relu}((W^{(0)})^\top x_i) \|_2^2$$

$$= \frac{1}{2} \sigma^2 \sum_{i=1}^n \left\{ \ell(y_i, f(x_i; W^{(0)}, v^{(0)})) \right\} \cdot \left\{ \| \mathrm{relu}((W^{(0)})^\top x_i) \|_2^2 \right\}$$

//by Lemma B.2

$$\geq \frac{1}{2} \sigma^2 \left\{ \sum_{i=1}^n \ell^{1/2}(y_i, f(x_i; W^{(0)}, v^{(0)})) \right\}^2 \cdot \left\{ \sum_{i=1}^n \| \mathrm{relu}((W^{(0)})^\top x_i) \|_2^{-2} \right\}^{-1}$$

//by Lemma B.3

$$\geq \frac{1}{2} \sigma^2 \left\{ \sum_{i=1}^n \ell(y_i, f(x_i; W^{(0)}, v^{(0)})) \right\} \cdot \left\{ \sum_{i=1}^n \| \mathrm{relu}((W^{(0)})^\top x_i) \|_2^{-2} \right\}^{-1}$$

$$\geq \frac{1}{2}\sigma^2 \mathcal{L}^{(0)} \cdot \left\{ \sum_{i=1}^{n} \|\text{relu}((W^{(0)})^\top x_i)\|_2^{-2} \right\}^{-1}$$

We replace the $B$ constant by $\left\{ \sum_{i=1}^{n} \|\text{relu}((W^{(0)})^\top x_i)\|_2^{-2} \right\}^{-1}$ in the previous proof of loose lower bound of Theorem 4.2. Similarly,

$$\mathcal{L}^{(0)}(T) \geq \frac{1}{\frac{1}{\mathcal{L}^{(0)}(0)} e^{-BT} + \frac{A}{B}(1 - e^{-BT})}$$

where $A = \sum_{i \in [n]} \left[ \max_{j \in [h]} \text{relu}((w_j^{(0)})^\top x_i) \right]^2, B = \frac{1}{2}\sigma^2 \left\{ \sum_{i=1}^{n} \|\text{relu}((W^{(0)})^\top x_i)\|_2^{-2} \right\}^{-1}$. The limit of this lower bound is

$$\lim_{t \to \infty} \frac{1}{\frac{1}{\mathcal{L}^{(0)}(0)} e^{-BT} + \frac{A}{B}(1 - e^{-BT})} = \frac{B}{A} = \frac{1}{2}\sigma^2 \left\{ \sum_{i=1}^{n} \|\text{relu}((W^{(0)})^\top x_i)\|_2^{-2} \right\}^{-1} \left\{ \sum_{i \in [n]} \left[ \max_{j \in [h]} \text{relu}((w_j^{(0)})^\top x_i) \right]^2 \right\}^{-1}$$

$\square$

**Example D.2** (On the downstream alignment of pretrained features (Theorem 4.2)). Here we provide an example on how the pretrained feature space affects the linear probing lower bound in Theorem 4.2 in the **overparametrized** regime. Consider one data point $x_+$ and two pretrained features $w_{+,1}, w_{+,2}$ with $\|x_+\|_2 = \|w_{+,1}\|_2 = \|w_{+,2}\|_2 = 1, \cos(x_+, w_{+,2}) = \frac{1}{3}\pi$.

1. If we get lucky such that $w_{+,1} = x_+$, then the limit is $\frac{B}{A} = \frac{15}{24}\sigma^2$.

2. If the $w_{+,1}$ is not so good for the downstream task such that $\cos(x_+, w_{+,1}) = \frac{1}{6}\pi$, then the limit becomes $\frac{B}{A} = \frac{16}{24}\sigma^2$.

Since $\frac{16}{24} > \frac{15}{24}$, we can tell that when the pretrained features do not align well with the downstream task, the lower bound gets higher, i.e. worse performance.

## D.2 Approximate DP-FT convergence

**Analysis of DP-FFT loss diffusion.** In the following $0^{\text{th}}$-order approximation of loss Langevin diffusion, denote the drift term by $W$-gradient as $T_1$, the drift term by $v$-gradient as $T_2$, the diffusion term by $W$-hessian as $T_3$, the diffusion term by $v$-hessian as $T_4$.

$$\dot{\mathcal{L}}^{(0)} = -\underbrace{\left\|\nabla_W \mathcal{L}^{(0)}\right\|_F^2}_{T_1} - \underbrace{\left\|\nabla_v \mathcal{L}^{(0)}\right\|_2^2}_{T_2} \tag{77}$$

$$+ \frac{1}{2}\sigma^2 \sum_{i=1}^{n} y_i^2 \ell(y_i, f(x_i; W^{(0)}, v^{(0)})) \left( \|\text{relu}((W^{(0)})^\top x_i)\|_2^2 + \sum_{j=1}^{h} (v_j^{(0)})^2 [\text{relu}'((w_j^{(0)})^\top x_i)]^2 \|x_i\|_2^2 \right) \tag{78}$$

$$= -\sum_{j=1}^{h} \left( \sum_{i=1}^{n} y_i \exp(-y_i f(x_i; W^{(0)}, v^{(0)})) \text{relu}((w_j^{(0)})^\top x_i) \right)^2 \tag{79}$$

$$-\sum_{j=1}^{h} \left\| \sum_{i=1}^{n} y_i \exp(-y_i f(x_i; W^{(0)}, v^{(0)})) v_j^{(0)} \mathbb{1}_{(w_j^{(0)})^\top x_i > 0} x_i \right\|_2^2 \tag{80}$$

$$+ \frac{1}{2}\sigma^2 \sum_{i=1}^{n} y_i^2 \ell(y_i, f(x_i; W^{(0)}, v^{(0)})) \left( \|\text{relu}((W^{(0)})^\top x_i)\|_2^2 + \sum_{j=1}^{h} (v_j^{(0)})^2 \mathbb{1}_{(w_j^{(0)})^\top x_i > 0}^2 \|x_i\|_2^2 \right) \tag{81}$$

$$= -\underbrace{\sum_{j=1}^{h} \left( \sum_{i=1}^{n} y_i \exp(-y_i f(x_i; W^{(0)}, v^{(0)})) \text{relu}((w_j^{(0)})^\top x_i) \right)^2}_{T_2} \tag{82}$$

$$-\sum_{j=1}^{h}\underbrace{\left\|\sum_{i=1}^{n}y_i\exp(-y_i f(x_i;W^{(0)},v^{(0)}))v_j^{(0)}\mathbb{1}_{(w_j^{(0)})^\top x_i>0}x_i\right\|_2^2}_{T_1} \tag{83}$$

$$+\underbrace{\frac{1}{2}\sigma^2\sum_{i=1}^{n}y_i^2\ell(y_i,f(x_i;W^{(0)},v^{(0)}))\|\mathrm{relu}((W^{(0)})^\top x_i)\|_2^2}_{T_4} \tag{84}$$

$$+\underbrace{\frac{1}{2}\sigma^2\sum_{i=1}^{n}y_i^2\ell(y_i,f(x_i;W^{(0)},v^{(0)}))\sum_{j=1}^{h}(v_j^{(0)})^2\mathbb{1}^2_{(w_j^{(0)})^\top x_i>0}\|x_i\|_2^2}_{T_3} \tag{85}$$

*Upper bound proof of Theorem 4.3.* **1. Upper bounds for** $T_1, T_3$. For $T_1$, the key idea is $\|x\|_2^2 \geq \langle x, z \rangle^2$ for any unit vector $z$.

$$T_1 = -\sum_{j=1}^{h}\left\|\sum_{i=1}^{n}y_i\exp(-y_i f(x_i;W^{(0)},v^{(0)}))v_j^{(0)}\mathbb{1}_{(w_j^{(0)})^\top x_i>0}x_i\right\|_2^2$$

$$//\text{since }\forall x\in\mathbb{R}^D, z\in\mathbb{S}^{D-1}, \|x\|_2^2\geq\langle x,z\rangle^2$$

$$\leq -\sum_{j=1}^{h}\left\langle\sum_{i=1}^{n}y_i\exp(-y_i f(x_i;W^{(0)},v^{(0)}))v_j^{(0)}\mathbb{1}_{(w_j^{(0)})^\top x_i>0}x_i, z\right\rangle^2$$

$$= -\sum_{j=1}^{h}\left(\sum_{i=1}^{n}y_i\exp(-y_i f(x_i;W^{(0)},v^{(0)}))v_j^{(0)}\mathbb{1}_{(w_j^{(0)})^\top x_i>0}\langle x_i,z\rangle\right)^2$$

$$= -\sum_{j=1}^{h}(v_j^{(0)})^2\left(\sum_{i=1}^{n}y_i\exp(-y_i f(x_i;W^{(0)},v^{(0)}))\mathbb{1}_{(w_j^{(0)})^\top x_i>0}\langle x_i,z\rangle\right)^2$$

$$//\text{pick }z=\frac{y_1 x_1}{\|x_1\|_2}, \text{ by Corollary B.5}$$

$$\leq -\gamma^2\sum_{j=1}^{h}(v_j^{(0)})^2\left(\sum_{i=1}^{n}\exp(-y_i f(x_i;W^{(0)},v^{(0)}))\mathbb{1}_{(w_j^{(0)})^\top x_i>0}\right)^2$$

$$= -\gamma^2\sum_{j=1}^{h}(v_j^{(0)})^2\left(\sum_{i\in\mathscr{I}(w_j^{(0)})}\exp(-y_i f(x_i;W^{(0)},v^{(0)}))\right)^2$$

$$= -\gamma^2\sum_{j=1}^{h}(v_j^{(0)})^2\left(\sum_{i\in\mathscr{I}(w_j^{(0)})}\ell(y_i,f(x_i;W^{(0)},v^{(0)}))\right)^2$$

For $T_3$, we align its form with $T_1$.

$$T_3 = \frac{1}{2}\sigma^2\sum_{i=1}^{n}y_i^2\ell(y_i,f(x_i;W^{(0)},v^{(0)}))\sum_{j=1}^{h}(v_j^{(0)})^2\mathbb{1}^2_{(w_j^{(0)})^\top x_i>0}\|x_i\|_2^2$$

$$//\text{since }\forall i\in[n], |y_i|=1$$

$$= \frac{1}{2}\sigma^2\sum_{i=1}^{n}\ell(y_i,f(x_i;W^{(0)},v^{(0)}))\sum_{j=1}^{h}(v_j^{(0)})^2\mathbb{1}_{(w_j^{(0)})^\top x_i>0}\|x_i\|_2^2$$

$$= \frac{1}{2}\sigma^2\sum_{j=1}^{h}(v_j^{(0)})^2\sum_{i=1}^{n}\|x_i\|_2^2\mathbb{1}_{(w_j^{(0)})^\top x_i>0}\ell(y_i,f(x_i;W^{(0)},v^{(0)}))$$

$$\leq \frac{1}{2}\sigma^2 \left(\max_{i\in[n]} \|x_i\|_2^2\right) \sum_{j=1}^{h}(v_j^{(0)})^2 \sum_{i=1}^{n} \mathbb{1}_{(w_j^{(0)})^\top x_i > 0}\ell(y_i, f(x_i; W^{(0)}, v^{(0)}))$$

$$= \frac{1}{2}\sigma^2 \left(\max_{i\in[n]} \|x_i\|_2^2\right) \sum_{j=1}^{h}(v_j^{(0)})^2 \sum_{i\in\mathscr{I}(w_j^{(0)})} \ell(y_i, f(x_i; W^{(0)}, v^{(0)}))$$

**2. Upper bounds of $T_2, T_4$.** For $T_2$, we use linear separability.

$$T_2 = -\sum_{j=1}^{h}\left(\sum_{i=1}^{n} y_i \exp(-y_i f(x_i; W^{(0)}, v^{(0)}))\mathrm{relu}((w_j^{(0)})^\top x_i)\right)^2$$

$$\text{//by Corollary B.5}$$

$$\leq -\sum_{j=1}^{h}\left(\sum_{i\in[n]} \exp(-y_i f(x_i; W^{(0)}, v^{(0)}))\mathbb{1}_{(w_j^{(0)})^\top x_i > 0}\gamma\|w_j^{(0)}\|_2\right)^2$$

$$= -\gamma^2\sum_{j=1}^{h}\|w_j^{(0)}\|_2^2\left(\sum_{i\in\mathscr{I}(w_j^{(0)})} \exp(-y_i f(x_i; W^{(0)}, v^{(0)}))\right)^2$$

$$= -\gamma^2\sum_{j=1}^{h}\|w_j^{(0)}\|_2^2\left(\sum_{i\in\mathscr{I}(w_j^{(0)})} \ell(y_i, f(x_i; W^{(0)}, v^{(0)}))\right)^2$$

For $T_4$, we align its form with $T_3$.

$$T_4 = \frac{1}{2}\sigma^2\sum_{i=1}^{n} y_i^2 \ell(y_i, f(x_i; W^{(0)}, v^{(0)}))\|\mathrm{relu}((W^{(0)})^\top x_i)\|_2^2$$

$$\text{//since } \forall i \in [n], |y_i| = 1$$

$$= \frac{1}{2}\sigma^2\sum_{i=1}^{n} \ell(y_i, f(x_i; W^{(0)}, v^{(0)}))\|\mathrm{relu}((W^{(0)})^\top x_i)\|_2^2$$

$$= \frac{1}{2}\sigma^2\sum_{i=1}^{n} \ell(y_i, f(x_i; W^{(0)}, v^{(0)}))\sum_{j\in[h]} \mathbb{1}_{(w_j^{(0)})^\top x_i > 0}\langle w_j^{(0)}, x_i\rangle^2$$

$$\leq \frac{1}{2}\sigma^2\sum_{i=1}^{n} \ell(y_i, f(x_i; W^{(0)}, v^{(0)}))\sum_{j\in[h]} \mathbb{1}_{(w_j^{(0)})^\top x_i > 0}\|w_j^{(0)}\|_2^2\|x_i\|_2^2$$

$$\leq \frac{1}{2}\sigma^2 \left(\max_{i\in[n]} \|x_i\|_2^2\right)\sum_{j=1}^{h}\|w_j^{(0)}\|_2^2\sum_{i\in[n]} \mathbb{1}_{(w_j^{(0)})^\top x_i > 0}\ell(y_i, f(x_i; W^{(0)}, v^{(0)}))$$

$$= \frac{1}{2}\sigma^2 \left(\max_{i\in[n]} \|x_i\|_2^2\right)\sum_{j=1}^{h}\|w_j^{(0)}\|_2^2\sum_{i\in\mathscr{I}(w_j^{(0)})} \ell(y_i, f(x_i; W^{(0)}, v^{(0)}))$$

**3. Combine upper bounds of $T_1, T_2, T_3, T_4$.**

$$\dot{\mathcal{L}}^{(0)} = T_1 + T_2 + T_3 + T_4$$

$$\leq -\gamma^2\sum_{j=1}^{h}\left[(v_j^{(0)})^2 + \|w_j^{(0)}\|_2^2\right]\left(\sum_{i\in\mathscr{I}(w_j^{(0)})} \ell(y_i, f(x_i; W^{(0)}, v^{(0)}))\right)^2$$

$$+ \frac{1}{2}\sigma^2 \left(\max_{i\in[n]} \|x_i\|_2^2\right) \sum_{j=1}^{h} \left[(v_j^{(0)})^2 + \|w_j^{(0)}\|_2^2\right] \sum_{i\in\mathcal{I}(w_j^{(0)})} \ell(y_i, f(x_i; W^{(0)}, v^{(0)}))$$

//abbr. $\ell_i := \ell(y_i, f(x_i; W^{(0)}, v^{(0)}))$

$$= -\gamma^2 \sum_{j=1}^{h} \left[(v_j^{(0)})^2 + \|w_j^{(0)}\|_2^2\right] \left(\sum_{i\in\mathcal{I}(w_j^{(0)})} \ell_i\right)^2$$

$$+ \frac{1}{2}\sigma^2 \left(\max_{i\in[n]} \|x_i\|_2^2\right) \sum_{j=1}^{h} \left[(v_j^{(0)})^2 + \|w_j^{(0)}\|_2^2\right] \sum_{i\in\mathcal{I}(w_j^{(0)})} \ell_i$$

$$= \sum_{j=1}^{h} \left[(v_j^{(0)})^2 + \|w_j^{(0)}\|_2^2\right] \left\{-\gamma^2 \left(\sum_{i\in\mathcal{I}(w_j^{(0)})} \ell_i\right)^2 + \frac{1}{2}\sigma^2 \left(\max_{i\in[n]} \|x_i\|_2^2\right) \left(\sum_{i\in\mathcal{I}(w_j^{(0)})} \ell_i\right)\right\}$$

$\because (v_j^{(0)})^2 + \|w_j^{(0)}\|_2^2 \geq (v_{j,t=0}^{(0)})^2 + \|w_{j,t=0}^{(0)}\|_2^2$

$\therefore$ When the drift term (negative) still dominates the dynamics, we take $t = 0$ for $(v_j^{(0)})^2 + \|w_j^{(0)}\|_2^2$.

$$\dot{\mathcal{L}}^{(0)} \leq \sum_{j=1}^{h} \left[(v_{j,t=0}^{(0)})^2 + \|w_{j,t=0}^{(0)}\|_2^2\right] \left\{-\gamma^2 \left(\sum_{i\in\mathcal{I}(w_j^{(0)})} \ell_i\right)^2 + \frac{1}{2}\sigma^2 \left(\max_{i\in[n]} \|x_i\|_2^2\right) \left(\sum_{i\in\mathcal{I}(w_j^{(0)})} \ell_i\right)\right\}$$

**4. Decompose loss by trapping.** If the trapping condition holds, we can decompose the loss $\mathcal{L}^{(0)} = \mathcal{L}_+^{(0)} + \mathcal{L}_-^{(0)}$, where $\mathcal{L}_*^{(0)}$ is only controlled by $w_j$ if $w_j^{(0)} \in \mathcal{S}_*$ ($* \in \{+, -\}$).

$$\dot{\mathcal{L}}_*^{(0)} \leq \sum_{j\in[h], w_j^{(0)}\in\mathcal{S}_*} \left[(v_{j,t=0}^{(0)})^2 + \|w_{j,t=0}^{(0)}\|_2^2\right] \left\{-\gamma^2 \left(\sum_{i\in\mathcal{I}(w_j^{(0)})} \ell_i\right)^2 + \frac{1}{2}\sigma^2 \left(\max_{i\in[n]} \|x_i\|_2^2\right) \left(\sum_{i\in\mathcal{I}(w_j^{(0)})} \ell_i\right)\right\}$$

$$\leq \sum_{j\in[h], w_j^{(0)}\in\mathcal{S}_*} \left[(v_{j,t=0}^{(0)})^2 + \|w_{j,t=0}^{(0)}\|_2^2\right] \left\{-\gamma^2 \left(\mathcal{L}_*^{(0)}\right)^2 + \frac{1}{2}\sigma^2 \left(\max_{i\in[n]} \|x_i\|_2^2\right) \mathcal{L}_*^{(0)}\right\}$$

Let $u = 1/\mathcal{L}_*^{(0)}, A = \sum_{j\in[h], w_j^{(0)}\in\mathcal{S}_*} \left[(v_{j,t=0}^{(0)})^2 + \|w_{j,t=0}^{(0)}\|_2^2\right], B = \gamma^2, C = \frac{1}{2}\sigma^2 \left(\max_{i\in[n]} \|x_i\|_2^2\right)$. Then

$$-\frac{du}{dt} \leq -AB + ACu$$

$$AB\exp(ACt) \leq \frac{d}{dt}(ue^{ACt})$$

$$\frac{B}{C}(\exp(ACt) - 1) \leq ue^{ACt} - u_0$$

$$\frac{B}{C}(\exp(ACt) - 1) + u_0 \leq ue^{ACt}$$

$$\frac{B}{C}(1 - \exp(-ACt)) + u_0 e^{-ACt} \leq u$$

$$\mathcal{L}_*^{(0)} \leq \frac{1}{\frac{B}{C}(1 - e^{-ACt}) + \frac{1}{\mathcal{L}_{t=0,*}^{(0)}} e^{-ACt}}$$

The time limit of the upper bound is

$$\lim_{t\to\infty} \mathcal{L}_*^{(0)} \leq \frac{C}{B} = \frac{\sigma^2}{2\gamma^2} \left(\max_{i\in[n]} \|x_i\|_2^2\right) = \frac{1}{2} \frac{\max_{i\in[n]} \|x_i\|_2^2}{\min_{i\in[n]} \|x_i\|_2^2} \sigma^2 \frac{1}{\mu^2}$$

**5. Combine clustered losses.**

$$\mathcal{L}^{(0)} = \mathcal{L}_-^{(0)} + \mathcal{L}_+^{(0)}$$

$$\leq \frac{1}{\frac{B}{C}(1 - e^{-A_+ Ct}) + \frac{1}{\mathcal{L}_{t=0,+}^{(0)}} e^{-A_+ Ct}} + \frac{1}{\frac{B}{C}(1 - e^{-A_- Ct}) + \frac{1}{\mathcal{L}_{t=0,-}^{(0)}} e^{-A_- Ct}}$$

$\square$

*Lower bound (type I) proof of Theorem 4.3.* **1. Upper bounds for** $T_1, T_3$. For $T_1$, the key idea is $\|x\|_2^2 \geq \langle x, z \rangle^2$ for any unit vector $z$.

$$T_1 = -\sum_{j=1}^h \left\| \sum_{i=1}^n y_i \exp(-y_i f(x_i; W^{(0)}, v^{(0)})) v_j^{(0)} \mathbb{1}_{(w_j^{(0)})^\top x_i > 0} x_i \right\|_2^2$$

$$//\text{since } \forall x \in \mathbb{R}^D, z \in \mathbb{S}^{D-1}, \|x\|_2^2 \geq \langle x, z \rangle^2$$

$$\leq -\sum_{j=1}^h \left\langle \sum_{i=1}^n y_i \exp(-y_i f(x_i; W^{(0)}, v^{(0)})) v_j^{(0)} \mathbb{1}_{(w_j^{(0)})^\top x_i > 0} x_i, z \right\rangle^2$$

$$= -\sum_{j=1}^h \left( \sum_{i=1}^n y_i \exp(-y_i f(x_i; W^{(0)}, v^{(0)})) v_j^{(0)} \mathbb{1}_{(w_j^{(0)})^\top x_i > 0} \langle x_i, z \rangle \right)^2$$

$$= -\sum_{j=1}^h (v_j^{(0)})^2 \left( \sum_{i=1}^n y_i \exp(-y_i f(x_i; W^{(0)}, v^{(0)})) \mathbb{1}_{(w_j^{(0)})^\top x_i > 0} \langle x_i, z \rangle \right)^2$$

$$//\text{pick } z = \frac{y_1 x_1}{\|x_1\|_2}, \text{ by Corollary B.5}$$

$$\leq -\gamma^2 \sum_{j=1}^h (v_j^{(0)})^2 \left( \sum_{i=1}^n \exp(-y_i f(x_i; W^{(0)}, v^{(0)})) \mathbb{1}_{(w_j^{(0)})^\top x_i > 0} \right)^2$$

$$= -\gamma^2 \sum_{j=1}^h (v_j^{(0)})^2 \left( \sum_{i \in \mathscr{I}(w_j^{(0)})} \exp(-y_i f(x_i; W^{(0)}, v^{(0)})) \right)^2$$

$$= -\gamma^2 \sum_{j=1}^h (v_j^{(0)})^2 \left( \sum_{i \in \mathscr{I}(w_j^{(0)})} \ell(y_i, f(x_i; W^{(0)}, v^{(0)})) \right)^2$$

For $T_3$, we align its form with $T_1$.

$$T_3 = \frac{1}{2}\sigma^2 \sum_{i=1}^n y_i^2 \ell(y_i, f(x_i; W^{(0)}, v^{(0)})) \sum_{j=1}^h (v_j^{(0)})^2 \mathbb{1}_{(w_j^{(0)})^\top x_i > 0}^2 \|x_i\|_2^2$$

$$//\text{since } \forall i \in [n], |y_i| = 1$$

$$= \frac{1}{2}\sigma^2 \sum_{i=1}^n \ell(y_i, f(x_i; W^{(0)}, v^{(0)})) \sum_{j=1}^h (v_j^{(0)})^2 \mathbb{1}_{(w_j^{(0)})^\top x_i > 0} \|x_i\|_2^2$$

$$= \frac{1}{2}\sigma^2 \sum_{j=1}^h (v_j^{(0)})^2 \sum_{i=1}^n \|x_i\|_2^2 \mathbb{1}_{(w_j^{(0)})^\top x_i > 0} \ell(y_i, f(x_i; W^{(0)}, v^{(0)}))$$

$$\leq \frac{1}{2}\sigma^2 \left( \max_{i \in [n]} \|x_i\|_2^2 \right) \sum_{j=1}^h (v_j^{(0)})^2 \sum_{i=1}^n \mathbb{1}_{(w_j^{(0)})^\top x_i > 0} \ell(y_i, f(x_i; W^{(0)}, v^{(0)}))$$

$$= \frac{1}{2}\sigma^2 \left( \max_{i \in [n]} \|x_i\|_2^2 \right) \sum_{j=1}^{h} (v_j^{(0)})^2 \sum_{i \in \mathcal{I}(w_j^{(0)})} \ell(y_i, f(x_i; W^{(0)}, v^{(0)}))$$

**2. Upper bounds of $T_2, T_4$.** For $T_2$, we use linear separability.

$$T_2 = -\sum_{j=1}^{h} \left( \sum_{i=1}^{n} y_i \exp(-y_i f(x_i; W^{(0)}, v^{(0)})) \mathrm{relu}((w_j^{(0)})^\top x_i) \right)^2$$

//by Corollary B.5

$$\leq -\sum_{j=1}^{h} \left( \sum_{i \in [n]} \exp(-y_i f(x_i; W^{(0)}, v^{(0)})) \mathbb{1}_{(w_j^{(0)})^\top x_i > 0} \gamma \|w_j^{(0)}\|_2 \right)^2$$

$$= -\gamma^2 \sum_{j=1}^{h} \|w_j^{(0)}\|_2^2 \left( \sum_{i \in \mathcal{I}(w_j^{(0)})} \exp(-y_i f(x_i; W^{(0)}, v^{(0)})) \right)^2$$

$$= -\gamma^2 \sum_{j=1}^{h} \|w_j^{(0)}\|_2^2 \left( \sum_{i \in \mathcal{I}(w_j^{(0)})} \ell(y_i, f(x_i; W^{(0)}, v^{(0)})) \right)^2$$

For $T_4$, we align its form with $T_3$.

$$T_4 = \frac{1}{2}\sigma^2 \sum_{i=1}^{n} y_i^2 \ell(y_i, f(x_i; W^{(0)}, v^{(0)})) \|\mathrm{relu}((W^{(0)})^\top x_i)\|_2^2$$

//since $\forall i \in [n], |y_i| = 1$

$$= \frac{1}{2}\sigma^2 \sum_{i=1}^{n} \ell(y_i, f(x_i; W^{(0)}, v^{(0)})) \|\mathrm{relu}((W^{(0)})^\top x_i)\|_2^2$$

$$= \frac{1}{2}\sigma^2 \sum_{i=1}^{n} \ell(y_i, f(x_i; W^{(0)}, v^{(0)})) \sum_{j \in [h]} \mathbb{1}_{(w_j^{(0)})^\top x_i > 0} \langle w_j^{(0)}, x_i \rangle^2$$

$$\leq \frac{1}{2}\sigma^2 \sum_{i=1}^{n} \ell(y_i, f(x_i; W^{(0)}, v^{(0)})) \sum_{j \in [h]} \mathbb{1}_{(w_j^{(0)})^\top x_i > 0} \|w_j^{(0)}\|_2^2 \|x_i\|_2^2$$

$$\leq \frac{1}{2}\sigma^2 \left( \max_{i \in [n]} \|x_i\|_2^2 \right) \sum_{j=1}^{h} \|w_j^{(0)}\|_2^2 \sum_{i \in [n]} \mathbb{1}_{(w_j^{(0)})^\top x_i > 0} \ell(y_i, f(x_i; W^{(0)}, v^{(0)}))$$

$$= \frac{1}{2}\sigma^2 \left( \max_{i \in [n]} \|x_i\|_2^2 \right) \sum_{j=1}^{h} \|w_j^{(0)}\|_2^2 \sum_{i \in \mathcal{I}(w_j^{(0)})} \ell(y_i, f(x_i; W^{(0)}, v^{(0)}))$$

**3. Combine upper bounds of $T_1, T_2, T_3, T_4$.**

$$\dot{\mathcal{L}}^{(0)} = T_1 + T_2 + T_3 + T_4$$

$$\leq -\gamma^2 \sum_{j=1}^{h} \left[ (v_j^{(0)})^2 + \|w_j^{(0)}\|_2^2 \right] \left( \sum_{i \in \mathcal{I}(w_j^{(0)})} \ell(y_i, f(x_i; W^{(0)}, v^{(0)})) \right)^2$$

$$+ \frac{1}{2}\sigma^2 \left( \max_{i \in [n]} \|x_i\|_2^2 \right) \sum_{j=1}^{h} \left[ (v_j^{(0)})^2 + \|w_j^{(0)}\|_2^2 \right] \sum_{i \in \mathcal{I}(w_j^{(0)})} \ell(y_i, f(x_i; W^{(0)}, v^{(0)}))$$

$$//\text{abbr. } \ell_i := \ell(y_i, f(x_i; W^{(0)}, v^{(0)}))$$

$$= -\gamma^2 \sum_{j=1}^{h} \left[(v_j^{(0)})^2 + \|w_j^{(0)}\|_2^2\right] \left(\sum_{i \in \mathscr{I}(w_j^{(0)})} \ell_i\right)^2$$

$$+ \frac{1}{2}\sigma^2 \left(\max_{i \in [n]} \|x_i\|_2^2\right) \sum_{j=1}^{h} \left[(v_j^{(0)})^2 + \|w_j^{(0)}\|_2^2\right] \sum_{i \in \mathscr{I}(w_j^{(0)})} \ell_i$$

$$= \sum_{j=1}^{h} \left[(v_j^{(0)})^2 + \|w_j^{(0)}\|_2^2\right] \left\{-\gamma^2 \left(\sum_{i \in \mathscr{I}(w_j^{(0)})} \ell_i\right)^2 + \frac{1}{2}\sigma^2 \left(\max_{i \in [n]} \|x_i\|_2^2\right) \left(\sum_{i \in \mathscr{I}(w_j^{(0)})} \ell_i\right)\right\}$$

$$\because (v_j^{(0)})^2 + \|w_j^{(0)}\|_2^2 \geq (v_{j,t=0}^{(0)})^2 + \|w_{j,t=0}^{(0)}\|_2^2$$

$\therefore$ When the drift term (negative) still dominates the dynamics, we take $t = 0$ for $(v_j^{(0)})^2 + \|w_j^{(0)}\|_2^2$.

$$\dot{\mathcal{L}}^{(0)} \leq \sum_{j=1}^{h} \left[(v_{j,t=0}^{(0)})^2 + \|w_{j,t=0}^{(0)}\|_2^2\right] \left\{-\gamma^2 \left(\sum_{i \in \mathscr{I}(w_j^{(0)})} \ell_i\right)^2 + \frac{1}{2}\sigma^2 \left(\max_{i \in [n]} \|x_i\|_2^2\right) \left(\sum_{i \in \mathscr{I}(w_j^{(0)})} \ell_i\right)\right\}$$

**4. Decompose loss by trapping.** If the trapping condition holds, we can decompose the loss $\mathcal{L}^{(0)} = \mathcal{L}_+^{(0)} + \mathcal{L}_-^{(0)}$, where $\mathcal{L}_*^{(0)}$ is only controlled by $w_j$ if $w_j^{(0)} \in \mathcal{S}_*$ ($* \in \{+, -\}$).

$$\dot{\mathcal{L}}_*^{(0)} \leq \sum_{j \in [h], w_j^{(0)} \in \mathcal{S}_*} \left[(v_{j,t=0}^{(0)})^2 + \|w_{j,t=0}^{(0)}\|_2^2\right] \left\{-\gamma^2 \left(\sum_{i \in \mathscr{I}(w_j^{(0)})} \ell_i\right)^2 + \frac{1}{2}\sigma^2 \left(\max_{i \in [n]} \|x_i\|_2^2\right) \left(\sum_{i \in \mathscr{I}(w_j^{(0)})} \ell_i\right)\right\}$$

$$\leq \sum_{j \in [h], w_j^{(0)} \in \mathcal{S}_*} \left[(v_{j,t=0}^{(0)})^2 + \|w_{j,t=0}^{(0)}\|_2^2\right] \left\{-\gamma^2 \left(\mathcal{L}_*^{(0)}\right)^2 + \frac{1}{2}\sigma^2 \left(\max_{i \in [n]} \|x_i\|_2^2\right) \mathcal{L}_*^{(0)}\right\}$$

Let $u = 1/\mathcal{L}_*^{(0)}, A = \sum_{j \in [h], w_j^{(0)} \in \mathcal{S}_*} \left[(v_{j,t=0}^{(0)})^2 + \|w_{j,t=0}^{(0)}\|_2^2\right], B = \gamma^2, C = \frac{1}{2}\sigma^2 \left(\max_{i \in [n]} \|x_i\|_2^2\right)$. Then

$$-\frac{du}{dt} \leq -AB + ACu$$

$$AB \exp(ACt) \leq \frac{d}{dt}(ue^{ACt})$$

$$\frac{B}{C}(\exp(ACt) - 1) \leq ue^{ACt} - u_0$$

$$\frac{B}{C}(\exp(ACt) - 1) + u_0 \leq ue^{ACt}$$

$$\frac{B}{C}(1 - \exp(-ACt)) + u_0 e^{-ACt} \leq u$$

$$\mathcal{L}_*^{(0)} \leq \frac{1}{\frac{B}{C}(1 - e^{-ACt}) + \frac{1}{\mathcal{L}_{t=0,*}^{(0)}}e^{-ACt}}$$

The time limit of the upper bound is

$$\lim_{t \to \infty} \mathcal{L}_*^{(0)} \leq \frac{C}{B} = \frac{\sigma^2}{2\gamma^2}\left(\max_{i \in [n]} \|x_i\|_2^2\right) = \frac{1}{2}\frac{\max_{i \in [n]} \|x_i\|_2^2}{\min_{i \in [n]} \|x_i\|_2^2}\sigma^2 \frac{1}{\mu^2}$$

**5. Combine clustered losses.**

$$\mathcal{L}^{(0)} = \mathcal{L}_-^{(0)} + \mathcal{L}_+^{(0)}$$

$$\leq \frac{1}{\frac{B}{C}(1 - e^{-A_+ Ct}) + \frac{1}{\mathcal{L}_{t=0,+}^{(0)}} e^{-A_+ Ct}} + \frac{1}{\frac{B}{C}(1 - e^{-A_- Ct}) + \frac{1}{\mathcal{L}_{t=0,-}^{(0)}} e^{-A_- Ct}}$$

$\square$

*Lower bound (type III) proof of Theorem 4.3.* **1.** **Lower bounds for** $T_1, T_3$. For $T_1$, we use $\left(\max_{k \in [n]} \|x_k\|_2^2\right)$.

$$T_1 = -\sum_{j=1}^{h} \left\| \sum_{i=1}^{n} y_i \exp(-y_i f(x_i; W^{(0)}, v^{(0)})) v_j^{(0)} \mathbb{1}_{(w_j^{(0)})^\top x_i > 0} x_i \right\|_2^2$$

$$//\text{abbr. } \ell_i := \exp(-y_i f(x_i; W^{(0)}, v^{(0)}))$$

$$= -\sum_{j=1}^{h} \left\| \sum_{i \in \mathscr{I}(w_j^{(0)})} y_i \ell_i v_j^{(0)} x_i \right\|_2^2$$

$$= -\sum_{j=1}^{h} \left\| \sum_{i \in \mathscr{I}(w_j^{(0)})} \ell_i v_j^{(0)} x_i \right\|_2^2$$

$$= -\sum_{j \in [h]} (v_j^{(0)})^2 \left\| \sum_{i \in \mathscr{I}(w_j^{(0)})} \ell_i x_i \right\|_2^2$$

$$\geq -\sum_{j \in [h]} (v_j^{(0)})^2 \left( \sum_{i \in \mathscr{I}(w_j^{(0)})} \ell_i \|x_i\|_2 \right)^2$$

$$\geq -\left( \max_{k \in [n]} \|x_k\|_2^2 \right) \sum_{j \in [h]} (v_j^{(0)})^2 \left( \sum_{i \in \mathscr{I}(w_j^{(0)})} \ell_i \right)^2$$

For $T_3$, we align its form with $T_1$.

$$T_3 = \frac{1}{2} \sigma^2 \sum_{i=1}^{n} y_i^2 \ell(y_i, f(x_i; W^{(0)}, v^{(0)})) \sum_{j=1}^{h} (v_j^{(0)})^2 \mathbb{1}_{(w_j^{(0)})^\top x_i > 0}^2 \|x_i\|_2^2$$

$$= \frac{1}{2} \sigma^2 \sum_{i=1}^{n} \ell_i \sum_{j=1}^{h} (v_j^{(0)})^2 \mathbb{1}_{(w_j^{(0)})^\top x_i > 0} \|x_i\|_2^2$$

$$= \frac{1}{2} \sigma^2 \sum_{j \in [h]} (v_j^{(0)})^2 \sum_{i \in \mathscr{I}(w_j^{(0)})} \ell_i \|x_i\|_2^2$$

$$\geq \frac{1}{2} \sigma^2 \left( \min_{k \in [n]} \|x_k\|_2^2 \right) \sum_{j \in [h]} (v_j^{(0)})^2 \left( \sum_{i \in \mathscr{I}(w_j^{(0)})} \ell_i \right)$$

**2. Lower bounds for** $T_2, T_4$. For $T_2$, we use $\langle x, y \rangle \leq \|x\|_2 \|y\|_2$.

$$T_2 = -\sum_{j=1}^{h} \left( \sum_{i=1}^{n} y_i \exp(-y_i f(x_i; W^{(0)}, v^{(0)})) \text{relu}((w_j^{(0)})^\top x_i) \right)^2$$

$$= -\sum_{j=1}^{h} \left( \sum_{i \in \mathscr{I}(w_j^{(0)})} y_i \exp(-y_i f(x_i; W^{(0)}, v^{(0)}))(w_j^{(0)})^\top x_i \right)^2$$

$$= -\sum_{j \in [h]} \left( \sum_{i \in \mathscr{I}(w_j^{(0)})} \ell_i \langle w_j^{(0)}, x_i \rangle \right)^2$$

$$\geq -\sum_{j \in [h]} \left( \sum_{i \in \mathscr{I}(w_j^{(0)})} \ell_i \|w_j^{(0)}\|_2 \|x_i\|_2 \right)^2$$

$$\geq -\left( \max_{k \in [n]} \|x_k\|_2^2 \right) \sum_{j \in [h]} \|w_j^{(0)}\|_2^2 \left( \sum_{i \in \mathscr{I}(w_j^{(0)})} \ell_i \right)^2$$

For $T_4$, we align its form with $T_2$.

$$T_4 = \frac{1}{2}\sigma^2 \sum_{i=1}^{n} y_i^2 \ell(y_i, f(x_i; W^{(0)}, v^{(0)})) \|\text{relu}((W^{(0)})^\top x_i)\|_2^2$$

$$//\text{since } \forall i \in [n], |y_i| = 1$$

$$= \frac{1}{2}\sigma^2 \sum_{i=1}^{n} \ell(y_i, f(x_i; W^{(0)}, v^{(0)})) \|\text{relu}((W^{(0)})^\top x_i)\|_2^2$$

$$= \frac{1}{2}\sigma^2 \sum_{i=1}^{n} \ell(y_i, f(x_i; W^{(0)}, v^{(0)})) \sum_{j \in [h]} \mathbb{1}_{(w_j^{(0)})^\top x_i > 0} \langle w_j^{(0)}, x_i \rangle^2$$

$$= \frac{1}{2}\sigma^2 \sum_{j \in [h]} \sum_{i \in \mathscr{I}(w_j^{(0)})} \ell_i \langle w_j^{(0)}, x_i \rangle^2$$

$$//\text{by Lemma B.4}$$

$$\geq \frac{1}{2}\sigma^2 \sum_{j \in [h]} \sum_{i \in \mathscr{I}(w_j^{(0)})} \ell_i \mu^2 \|w_j^{(0)}\|_2^2 \|x_i\|_2^2$$

$$= \frac{1}{2}\sigma^2 \mu^2 \sum_{j \in [h]} \|w_j^{(0)}\|_2^2 \sum_{i \in \mathscr{I}(w_j^{(0)})} \ell_i \|x_i\|_2^2$$

$$\geq \frac{1}{2}\sigma^2 \mu^2 \left( \min_{k \in [n]} \|x_k\|_2^2 \right) \sum_{j \in [h]} \|w_j^{(0)}\|_2^2 \left( \sum_{i \in \mathscr{I}(w_j^{(0)})} \ell_i \right)$$

**3. Combine lower bounds of $T_1, T_2, T_3, T_4$.**

$$\dot{\mathcal{L}}^{(0)} = T_1 + T_2 + T_3 + T_4$$

$$\geq -\left( \max_{k \in [n]} \|x_k\|_2^2 \right) \sum_{j \in [h]} \left[ (v_j^{(0)})^2 + \|w_j^{(0)}\|_2^2 \right] \left( \sum_{i \in \mathscr{I}(w_j^{(0)})} \ell_i \right)^2$$

$$+ \frac{1}{2}\sigma^2 \left( \min_{k \in [n]} \|x_k\|_2^2 \right) \sum_{j \in [h]} \left[ (v_j^{(0)})^2 + \mu^2 \|w_j^{(0)}\|_2^2 \right] \left( \sum_{i \in \mathscr{I}(w_j^{(0)})} \ell_i \right)$$

$$//\text{by balancedness, } \|w_j^{(0)}\|_2^2 = (v_j^{(0)})^2$$

$$\geq -2 \left( \max_{k \in [n]} \|x_k\|_2^2 \right) \sum_{j \in [h]} \|w_j^{(0)}\|_2^2 \left( \sum_{i \in \mathscr{I}(w_j^{(0)})} \ell_i \right)^2 + \frac{\sigma^2(1+\mu^2)}{2} \left( \min_{k \in [n]} \|x_k\|_2^2 \right) \sum_{j \in [h]} \|w_j^{(0)}\|_2^2 \left( \sum_{i \in \mathscr{I}(w_j^{(0)})} \ell_i \right)$$

**4. Decompose loss by trapping.** If the trapping condition holds, we can decompose the loss $\mathcal{L}^{(0)} = \mathcal{L}_+^{(0)} + \mathcal{L}_-^{(0)}$, where $\mathcal{L}_*^{(0)}$ is only controlled by $w_j$ if $w_j^{(0)} \in \mathcal{S}_*$ ($* \in \{+,-\}$).

$$\dot{\mathcal{L}}_*^{(0)} \geq -2 \left( \max_{k \in [n]} \|x_k\|_2^2 \right) \sum_{j \in [h], w_j^{(0)} \in \mathcal{S}_*} \|w_j^{(0)}\|_2^2 (\mathcal{L}_*^{(0)})^2 + \frac{\sigma^2(1+\mu^2)}{2} \left( \min_{k \in [n]} \|x_k\|_2^2 \right) \sum_{j \in [h], w_j^{(0)} \in \mathcal{S}_*} \|w_j^{(0)}\|_2^2 \mathcal{L}_*^{(0)}$$

$$= \left\{ \sum_{j \in [h], w_j^{(0)} \in \mathcal{S}_*} \|w_j^{(0)}\|_2^2 \right\} \cdot \left\{ -2 \left( \max_{k \in [n]} \|x_k\|_2^2 \right) (\mathcal{L}_*^{(0)})^2 + \frac{\sigma^2(1+\mu^2)}{2} \left( \min_{k \in [n]} \|x_k\|_2^2 \right) \mathcal{L}_*^{(0)} \right\}$$

The time limit of the loss lower bound is

$$\lim_{t \to \infty} \mathcal{L}_*^{(0)} \geq \frac{1}{2} \frac{\min_{k \in [n]} \|x_k\|_2^2}{\max_{k \in [n]} \|x_k\|_2^2} \sigma^2 \frac{1+\mu^2}{2}$$

By the previous lower bound proof,

$$\|W^{(0)}\|_F^2 \leq \|W_0^{(0)}\|_F^2 e^{2(\max_{k \in [n]} \|x_i\|_2)\mathcal{L}_0^{(0)} t}$$

Let $u = \frac{1}{\mathcal{L}_*^{(0)}}, A = \|W_0^{(0)}\|_F^2, \lambda_2 = 2(\max_{k \in [n]} \|x_i\|_2)\mathcal{L}_0^{(0)}, B = 2\max_{k \in [n]} \|x_k\|_2^2, C = \frac{\sigma^2(1+\mu^2)}{2} \min_{k \in [n]} \|x_k\|_2^2$. Then consider integrating factor $\exp(AC/\lambda_2 \exp(\lambda_2 t))$.

$$-\frac{d}{dt} u \geq A e^{\lambda_2 t}(-B + Cu)$$

$$ABe^{\lambda_2 t} \geq ACe^{\lambda_2 t} u + \frac{d}{dt} u$$

$$ABe^{\lambda_2 t} \exp(AC/\lambda_2 \exp(\lambda_2 t)) \geq AC \exp(AC/\lambda_2 \exp(\lambda_2 t))e^{\lambda_2 t} u + \exp(AC/\lambda_2 \exp(\lambda_2 t))\frac{d}{dt} u$$

$$\frac{B}{C}\frac{d}{dt}[\exp(AC/\lambda_2 \exp(\lambda_2 t))] \geq \frac{d}{dt}(u \cdot \exp(AC/\lambda_2 \exp(\lambda_2 t)))$$

$$\frac{B}{C}[\exp(AC/\lambda_2 \exp(\lambda_2 t)) - \exp(AC/\lambda_2)] \geq u \cdot \exp(AC/\lambda_2 \exp(\lambda_2 t)) - u_0 \cdot \exp(AC/\lambda_2)$$

$$\frac{B}{C}[1 - \exp(AC/\lambda_2(1 - \exp(\lambda_2 t)))] \geq u - u_0 \cdot \exp(AC/\lambda_2(1 - \exp(\lambda_2 t)))$$

$$\mathcal{L}_*^{(0)} \geq \frac{1}{\frac{1}{\mathcal{L}_{*,t=0}^{(0)}} e^{AC/\lambda_2(1-\exp(\lambda_2 t))} + \frac{B}{C}\left[1 - e^{AC/\lambda_2(1-\exp(\lambda_2 t))}\right]}$$

**5. Combine clustered losses.**

$$\mathcal{L}^{(0)} = \mathcal{L}_-^{(0)} + \mathcal{L}_+^{(0)}$$

$$\geq \frac{1}{\frac{1}{\mathcal{L}_{+,t=0}^{(0)}} e^{AC/\lambda_2(1-\exp(\lambda_2 t))} + \frac{B}{C}\left[1 - e^{AC/\lambda_2(1-\exp(\lambda_2 t))}\right]} + \frac{1}{\frac{1}{\mathcal{L}_{-,t=0}^{(0)}} e^{AC/\lambda_2(1-\exp(\lambda_2 t))} + \frac{B}{C}\left[1 - e^{AC/\lambda_2(1-\exp(\lambda_2 t))}\right]}$$

$\square$

## D.3 Privacy budget allocation

*Proof of Theorem 5.1.* For any $j \in [h]$, with probability $1 - \rho$, its initial absolute value is bounded by

$$|v_j| \leq \sqrt{2\beta^2 \ln(2/\rho)} \tag{86}$$

Then with probability $(1 - \rho)^h$, the maximum worse initial value is bounded by

$$\max_{j \in [h]} (c_j \cdot v_j) \leq \sqrt{\beta^2 \ln(2/\rho)} \tag{87}$$

where we define $c_j$ by $w_j \in S_{c_j}$. The approximate DP-LP dynamics is

$$\dot{v}_j = \sum_{i=1}^{n} y_i \ell_i \mathrm{relu}(w_j^\top x_i) \tag{88}$$

Say $w_j \in S_c$ for some $c \in \{-1, 1\}$, then during DP-LP, when $\mathrm{sign}(v_j(T)) = \mathrm{sign}(v_j(0))$,

$$|v_j(T) - v_j(0)| = \int_0^T \sum_{y_i=c} \ell_i \mathrm{relu}(w_j^\top x_i) dt \tag{89}$$

$$\geq \min_{y_i=c} |\mathrm{relu}(w_j^\top x_i)| \int_0^T \mathcal{L}_c(t) dt \tag{90}$$

$$//\text{by Theorem 4.2} \tag{91}$$

$$\geq \min_{y_i=c} \mathrm{relu}(w_j^\top x_i) \frac{\frac{1}{2}\sigma^2 \left\{ \sum_{y_i=c} \|\mathrm{relu}(W^\top x_i)\|_2^{-2} \right\}^{-1}}{\sum_{w_j \in S_c} \left[ \max_{y_i=c} w_j^\top x_i \right]^2} \tag{92}$$

$$= \frac{1}{2}\sigma^2 \frac{\min_{y_i=c} \mathrm{relu}(w_j^\top x_i)}{\sum_{w_j \in S_c} \left[ \max_{y_i=c} w_j^\top x_i \right]^2} \left\{ \sum_{y_i=c} \|\mathrm{relu}(W^\top x_i)\|_2^{-2} \right\}^{-1} \tag{93}$$

$$= \frac{1}{2}\sigma^2 Q \tag{94}$$

where we define a constant $Q$ to describe the pre-training quality. If the pre-trained features are better, $Q$ becomes larger. To mitigate the feature distortion, we need $c \cdot v_j > 0$, then the necessary DP-LP run-time is

$$\Delta t \propto \frac{\sigma^2}{Q} \sqrt{\beta^2 \ln(2/\rho)} \propto \frac{\sigma^2}{Q} \sqrt{\ln(2/\rho)} \tag{95}$$

where we ignore $\beta$ as it is typically pre-determined in real implementations (e.g. the Linear layers in PyTorch). $\qquad \square$

## E  Appendix: Theory without approximation

For convenience, we use different notations for the data input dimension $d = d_x$ and the backbone weight matrix $B = W^\top$ in the following proofs.

### E.1  Itô's formula and its consequences

We denote $M_{m,n}(\mathbb{R})$ as the space of m-by-n real matrices.

**Theorem E.1** (Itô's formula). *Let $X_t$ be a $\mathbb{R}^n$-valued Itô process satisfying the stochastic differential equation $\partial X_t = A_1(t, X_t)\partial t + A_2(t, X_t)\partial W_t$ with $A_1(t, X_t)$ being $\mathbb{R}^n$-valued, $A_2(t, X_t)$ being $M_{m,n}(\mathbb{R})$-valued, and $W_t$ being a standard n-dimensional brownian motion. Let $f : [0, \infty) \times \mathbb{R}^n \to \mathbb{R}$ be a function with continuous partial derivatives. Then $Y_t := f(t, X_t)$ is also an Itô process, and its stochastic differential equation is*

$$\partial Y_t = \frac{\partial f(t, X_t)}{\partial t}\partial t + \langle \nabla f(t, X_t), A_1(t, X_t)\partial t + A_2(t, X_t)\partial W_t \rangle + \frac{1}{2}\langle A_2(t, X_t)\partial W_t, H_f A_2(t, X_t)\partial W_t \rangle \tag{96}$$

*where $H_f$ is the Hessian matrix of $f$ over $X_t$ defined as $(H_f)_{ij} = \frac{\partial^2 f}{\partial (X_t)_i \partial (X_t)_j}$ and $(X_t)_i$ denotes the i-th entry of random vector $X_t$.*

**Corollary E.2** (Loss dynamics during linear probing). *During linear probing (Equation equation 121), the stochastic differential equation describing the loss dynamics is*

$$\partial\mathcal{L}_{\text{lp}} = -(B_0^T v - X^T Y)^T B_0^T B_0 (B_0^T v - X^T Y)\partial t + \sqrt{2\sigma^2}(B_0^T v - X^T Y)^T B_0^T \partial W_t + h\sigma^2 \partial t. \tag{97}$$

*Proof of Corollary E.2.* By Itô's formula (Equation equation E.1), the loss dynamics is

$$\partial\mathcal{L}_{\text{lp}} = \partial\frac{1}{2}\|XB_0^T v - Y\|^2 \tag{98}$$

$$= (XB_0^T v - Y)^T XB_0^T \partial v + \frac{1}{2}(\partial v)^T B_0 X^T XB_0^T (\partial v) \tag{99}$$

$$= (XB_0^T v - Y)^T XB_0^T \partial v + \frac{1}{2}(\partial v)^T (\partial v) \tag{100}$$

$$//\text{by Definition E.5} \tag{101}$$

$$= (XB_0^T v - Y)^T XB_0^T[-B_0 X^T (XB_0^T v - Y)\partial t + \sqrt{2\sigma^2}\partial W_t] + h\sigma^2 \partial t \tag{102}$$

$$= (B_0^T v - X^T Y)^T B_0^T[-B_0(B_0^T v - X^T Y)\partial t + \sqrt{2\sigma^2}\partial W_t] + h\sigma^2 \partial t \tag{103}$$

$$= -(B_0^T v - X^T Y)^T B_0^T B_0 (B_0^T v - X^T Y)\partial t + \sqrt{2\sigma^2}(B_0^T v - X^T Y)^T B_0^T \partial W_t + h\sigma^2 \partial t \tag{104}$$

$$\square$$

**Corollary E.3** (Loss dynamics during fine-tuning). *During fine-tuning (Equation equation 122), the stochastic differential equation describing the loss dynamics is*

$$\partial\mathcal{L}_{\text{ft}} = -(B^T v - X^T Y)^T B^T B(B^T v - X^T Y)\partial t + (B^T v - X^T Y)^T B^T \sqrt{2\sigma^2}\partial W_t$$
$$- (B^T v - X^T Y)^T (B^T v - X^T Y)v^T v\partial t + (B^T v - X^T Y)^T (\sqrt{2\sigma^2}\partial W_t')v \tag{105}$$
$$+ \sigma^2\|B\|_F^2 \partial t + \sigma^2 d\|v\|_2^2 \partial t.$$

*where we use $\partial$ as the differential sign and use $d$ as the data input dimension.*

*Proof of Corollary E.3.* Similar to Corollary E.2, we use Itô's formula (Equation E.1), the loss dynamics of fine-tuning is

$$\partial\mathcal{L}_{\text{ft}} = \partial\frac{1}{2}\|XB^T v - Y\|^2 \tag{106}$$

$$= \frac{1}{2}\left\langle\nabla_v\|(XB^T v - Y)\|^2, \partial v\right\rangle + \frac{1}{2}\left\langle\nabla_B\|(XB^T v - Y)\|^2, \text{vec}(\partial B)\right\rangle \tag{107}$$

$$+ \frac{1}{4}(\partial v)^T H_{\|(XB^T v - Y)\|^2}(\partial v) + \frac{1}{4}[\text{vec}(\partial B)]^T H_{\|(XB^T v - Y)\|^2}\text{vec}(\partial B) \tag{108}$$

$$= (XB^T v - Y)^T XB^T \partial v + (XB^T v - Y)^T X(\partial B)^T v \tag{109}$$

$$+ \frac{1}{2}(\partial v)^T BX^T XB^T (\partial v) + \frac{1}{2}[\text{vec}(\partial B)]^T \begin{bmatrix} v_1 \\ 0 \\ \vdots \\ v_h \end{bmatrix} \underbrace{\begin{bmatrix} v_1 & 0 & \cdots & v_h \end{bmatrix}}_{d \times h} \text{vec}(\partial B) \tag{110}$$

$$= -(B^T v - X^T Y)^T B^T B(B^T v - X^T Y)\partial t + (B^T v - X^T Y)^T B^T \sqrt{2\sigma^2}\partial W_t \tag{111}$$

$$- (B^T v - X^T Y)^T (B^T v - X^T Y)v^T v\partial t + (B^T v - X^T Y)^T (\sqrt{2\sigma^2}\partial W_t')v \tag{112}$$

$$+ \sigma^2\text{trace}(BB^T)\partial t + \sigma^2 d\|v\|^2 \partial t \tag{113}$$

$$= -(B^T v - X^T Y)^T B^T B(B^T v - X^T Y)\partial t + (B^T v - X^T Y)^T B^T \sqrt{2\sigma^2}\partial W_t \tag{114}$$

$$- (B^T v - X^T Y)^T (B^T v - X^T Y)v^T v\partial t + (B^T v - X^T Y)^T (\sqrt{2\sigma^2}\partial W_t')v \tag{115}$$

$$+ \sigma^2\|B\|_F^2 \partial t + \sigma^2 d\|v\|_2^2 \partial t \tag{116}$$

$$\square$$

*Remark* E.4 (Noise effects on linear networks). In the loss dynamics of fine-tuning (Corollary E.3), the noise induced deterministic terms

$$\sigma^2(\|B\|_F^2 + d\|v\|_2^2)\partial t$$

does not explicitly depend on the linear head size $h$. We do a sanity check for this result in a discretized setting (so that we skip Itô's lemma and stochastic calculus). Say we inject noise $\Delta B$ to $B$, where $\Delta B$ is a $h \times d$-matrix, and its entries are independent and follow Gaussian distribution $\mathcal{N}(0, \sigma)$. Then the expectation of the perturbed loss is:

$$\mathbb{E}[\mathcal{L}] = \frac{1}{2}\mathbb{E}[\|X(B + \Delta B)^T v - Y\|^2] \tag{117}$$

$$= \frac{1}{2}\|XB^T v - Y\|^2 + \mathbb{E}[(XB^T v - Y)^T X(\Delta B)^T v] + \frac{1}{2}\mathbb{E}[v^T \Delta B(\Delta B)^T v] \tag{118}$$

$$= \frac{1}{2}\|XB^T v - Y\|^2 + \frac{1}{2}\mathbb{E}[v^T \Delta B(\Delta B)^T v] \tag{119}$$

$$= \frac{1}{2}\|XB^T v - Y\|^2 + \frac{1}{2}\sigma^2 \cdot d \cdot \|v\|^2 \tag{120}$$

As a result, we find that, in the discrete updates, the noise induced deterministic terms does not explicitly depend on the linear head size $h$ either. So our findings in the continuous case matches the discrete case.

## E.2   Modified Langevin diffusion

**Definition E.5** (Langevin diffusion for linear probing). Let $Q_t$ be the standard $h$-dimensional Brownian motion. Then the Langevin diffusion for linear probing is defined by the following stochastic differential equation:

$$\partial v = -\nabla_v \mathcal{L}(v, B_0)\partial t + \sqrt{2\sigma^2}\partial Q_t$$
$$= -B_0 X^T(XB_0^T v - Y)\partial t + \sqrt{2\sigma^2}\partial Q_t. \tag{121}$$

Here we use "$\partial$" as the differential notation.

**Definition E.6** (Langevin diffusion for fine-tuning). Let $Q_t$ be the standard $h$-dimensional brownian motion and $Q_t'$ be a matrix whose entries are standard and independent brownian motions. Then we define the Langevin diffusion for fine-tuning a two-layer linear network as

$$\partial v = -\nabla_v \mathcal{L}(v, B)\partial t + \sqrt{2\sigma^2}\partial Q_t$$
$$= -BX^T(XB^T v - Y)\partial t + \sqrt{2\sigma^2}\partial Q_t$$
$$\partial B = -\nabla_B \mathcal{L}(v, B)\partial t + \sqrt{2\sigma^2}\partial Q_t'$$
$$= -v(XB^T v - Y)^T X\partial t + \sqrt{2\sigma^2}\partial Q_t'. \tag{122}$$

Here we introduce an assumption based on random initialization. It describes a common phenomenon in differential privacy deployment: the loss might not converge if the privacy mechanism perturbs the gradients too much (Ponomareva et al., 2023). To ensure that DP-SGD works for full fine-tuning, we assume that the noise scale (or variance) in the privacy mechanism is upper bounded by a constant.

**Assumption E.7** (Upper bounded noise scale). Let $\beta > \frac{-\|X^T Y\| + \sqrt{\|X^T Y\|^2 + 4(1+d_x)\|X^T Y\| + 4d_x}}{2h}$. Then we assume that the noise scale $\sigma > 0$ we add for privacy in the fine-tuning process is upper-bounded by

$$\sigma^2 < \min\left\{\frac{h\beta + \|B_0 X^T Y\|^2}{2h}, \frac{h\beta - 1}{\sqrt{2}(1+d)}, \frac{1}{1 + \sqrt{2}(1+d)}\left[\frac{h\beta(h\beta + \|X^T Y\|^2)}{(1+d)\|X^T Y\| + d} - 1\right]\right\}. \tag{123}$$

Equation (25) upper monotonically decreases in time if Assumption E.7 also holds.

To understand the properties of a dynamics analysis problem, it can be useful to identify *invariants*, or functions whose output is conserved during optimization. Such conservation laws can be seen as a "weaker"

form of implicit bias, helping to elucidate which properties (e.g., sparsity, low-rank) are preferred by the optimization dynamics among a potentially infinite set of minimizers (Marcotte et al., 2023). To prove the convergence of our optimization, we study the *imbalance matrix*, an invariant for multi-layer linear networks that has previously been studied in the context of gradient flows (but not Langevin dynamics, to the best of our knowledge).

**Definition E.8** (Imbalance matrix)**.** For a two-layer linear network, we define the imbalance matrix as

$$D := vv^T - BB^T. \tag{124}$$

Prior work on gradient flows has found that the imbalance matrix remains invariant over the evolution of gradient flows modeling gradient descent (Arora et al., 2018; Du et al., 2018; Marcotte et al., 2023). This property can be used to derive tight convergence bounds (Min et al., 2021; 2023a). However, a similar analysis has not materialized for Langevin diffusion models of DP-GD.

We observe that prior work on Langevin diffusion to analyze private optimization has implicitly assumed that the sensitivity of each layer in a neural network is the same (Ganesh et al., 2023b; Ye et al., 2023b). Hence, they fix a uniform noise scale for every parameter of the network. Under these conditions, we show that, when we ignore the sensitivity of each layer and use a uniform noise scale $\sigma$, the imbalance matrix is *not* invariant in expectation, unlike in (noise-free) gradient flow (Arora et al., 2018; Du et al., 2018; Marcotte et al., 2023); that is, its derivative over time is nonzero. This complicates the use of the imbalance matrix for theoretical analysis (Ye & Du, 2021).

**Lemma E.9** (Imbalance matrix in fine-tuning)**.** *During fine-tuning (Equation* (122)*), the derivative of the imbalance matrix $D$ in Definition E.8 is*

$$\frac{\partial}{\partial t}\mathbb{E}[D] = (1 - d)\sigma^2 I_{h \times h}, \tag{125}$$

*where $d$ is the dimension of data inputs ($B \in \mathbb{R}^{h \times d}$).*

Our main observation is that by modeling differences in sensitivity of different layers, we can recover the invariance property of the imbalance matrix. The following proposition characterizes the sensitivity of the linear head and the feature extractor, and illustrates why they have differing sensitivities at initialization.

**Proposition E.10.** *We assume that the training dataset $\mathcal{D} = (X, Y)$ is normalized such that $X^T X = I_{d \times d}, \|Y\|_2 = 1$. We initialize the linear head by $v_0 \sim \mathcal{N}(0, \beta I_{h \times h})$ and $\beta = h/\sqrt{d}$. At the initialization of full fine-tuning, the linear head $v$ has a greater layer sensitivity (Béthune et al., 2024) than the feature extractor $B$:*

$$\Delta(\nabla_v \mathcal{L}(v_0, B_0)) = \Theta\left(\sqrt{d} \cdot \Delta(\nabla_B \mathcal{L}(v_0, B_0))\right) \tag{126}$$

Based on this observation, we propose a modified version of Langevin diffusion for full fine-tuning, which accounts for layer-wise sensitivity. With this modified definition, the imbalance matrix is again invariant in expectation.

**Definition E.11** (Modified Langevin diffusion for fine-tuning)**.** Let $Q_t$ be the standard $h$-dimensional brownian motion. Let $Q_t'$ be a $h \times d$ matrix whose entries are standard and independent brownian motions. Then we define the modified Langevin diffusion for fine-tuning a two-layer linear network as

$$\begin{aligned}
\partial v = & - \nabla_v \mathcal{L}(v, B) \partial t + \sqrt{2\sigma^2 d} \partial Q_t \\
= & - BX^T X (B^T v - X^T Y) \partial t + \sqrt{2\sigma^2 d} \partial Q_t \\
\partial B = & - \nabla_B \mathcal{L}(v, B) \partial t + \sqrt{2\sigma^2} \partial Q_t' \\
= & - v(XB^T v - Y)^T X \partial t + \sqrt{2\sigma^2} \partial Q_t'.
\end{aligned} \tag{127}$$

The only difference between this diffusion and Equation (122) is the additional factor of $\sqrt{d}$, shown in red, reflecting the fact that the linear head has greater function sensitivity than the feature extractor.

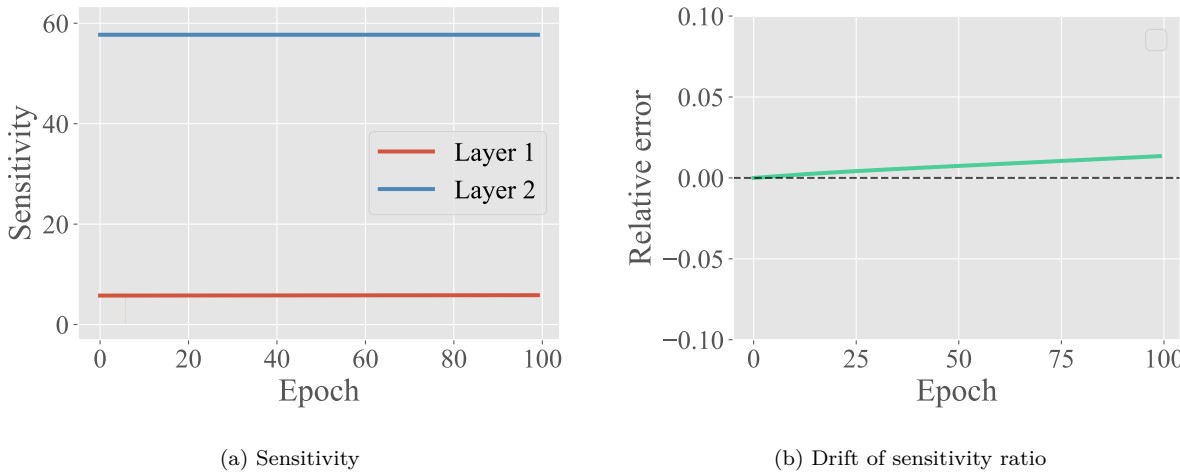

(a) Sensitivity

(b) Drift of sensitivity ratio

Figure 9: Evaluation of layer-wise sensitivity when running DP-GD on 2-layer linear networks and synthetic data (Béthune et al., 2024). We initialize the network parameter according to Proposition E.10. We take average on $10^4$ random seeds with standard error smaller than $10^{-3}$.

### E.3   Linear probing loss upper bound

The main idea of the proofs for convergence is to replace gradient terms with loss terms. By doing so, we obtain inequalities containing only loss terms and some other constants.

For the linear probing setting, we first show the strong convexity of the loss function. Then we can use the Lojasiewicz inequality to replace gradient terms with the loss terms.

**Lemma E.12** ((Strong) convexity of linear probing phase). *The empirical risk $\mathcal{L} = \frac{1}{2}\sum_{i=1}^{n} \ell(f(x_i), y_i)$ is 1-strongly convex.*

**Lemma E.13** (Initial loss before linear probing). *If we initialize the linear head by $v_{t=0} \sim \mathcal{N}(0, \beta I_{h \times h})$, then the expected empirical risk before linear probing is*

$$\mathbb{E}[\mathcal{L}_0] = \frac{1}{2}(h\beta + \|Y\|^2) \tag{128}$$

*Proof of Lemma E.13.* We initialize the linear head with a Gaussian distribution $\mathcal{N}(0, \beta I_{h \times h})$. So the expected initial loss is:

$$\mathbb{E}[\mathcal{L}_0] = \frac{1}{2}\mathbb{E}[\|XB_0^T v_0 - Y\|^2] \tag{129}$$

$$= \frac{1}{2}\mathbb{E}[v_0^T B_0 X^T X B_0^T v_0 + Y^T Y - 2Y^T X B_0^T v_0] \tag{130}$$

$$= \frac{1}{2}\mathbb{E}[v_0^T B_0 B_0^T v_0 + Y^T Y] \tag{131}$$

$$\text{//we assumed in section 3.1 that } B_0 \text{ has orthogonal rows} \tag{132}$$

$$= \frac{1}{2}\mathbb{E}[v_0^T v_0 + Y^T Y] \tag{133}$$

$$\text{//by } v_{t=0} \sim \mathcal{N}(0, I_{h \times h}) \tag{134}$$

$$= \frac{1}{2}(h\beta + \|Y\|^2) \tag{135}$$

$\square$

**Theorem E.14** (Expected loss upper bound of linear probing). *The expected empirical risk in linear probing is upper bounded by*

$$\mathbb{E}[\mathcal{L}_{\mathrm{lp}}(t)] \leq e^{-t}\mathbb{E}[\mathcal{L}_0] + (1 - e^{-t})(\gamma + h\sigma^2) \tag{136}$$

*Proof of Theorem 4.4.* By Lemma E.12, $\mathcal{L}$ is 1-strongly convex, we have the Lojasiewicz inequality. Here we abuse the notation $\mathcal{L}$ and consider it as a function of the linear head $v$ because we fix $B_0$ in the linear probing process.

$$\mathcal{L}(v) - \{\min_v \mathcal{L}\} \leq \frac{1}{2}\|\nabla_v \mathcal{L}(v)\|_2^2 \tag{137}$$

For simplicity, we denote $\mathbb{E}[\mathcal{L}] := \hat{\mathcal{L}}$. Consider the Langevin diffusion in Equation equation 121 when $\mathcal{L}(v) - \{\min_v \mathcal{L}\} - h\sigma^2 > 0$, by Corollary E.2:

$$\partial\mathcal{L}(v) = \langle \nabla_v \mathcal{L}(v), -\nabla_v \mathcal{L}(v)\partial t + \sqrt{2\sigma^2}\partial W_t \rangle + h\sigma^2\partial t \tag{138}$$

$$\partial\mathcal{L}(v) \leq -\|\nabla_v \mathcal{L}(v)\|_2^2 \partial t + \langle \nabla_v \mathcal{L}(v), \sqrt{2\sigma^2}\partial W_t \rangle + h\sigma^2\partial t \tag{139}$$

$$//\text{By Lojasiewicz inequality} \tag{140}$$

$$\partial\mathcal{L}(v) \leq (-\mathcal{L}(v) + \{\min_v \mathcal{L}\})\partial t + \langle \nabla_v \mathcal{L}(v), \sqrt{2\sigma^2}\partial W_t \rangle + h\sigma^2\partial t \tag{141}$$

$$\partial(\mathbb{E}[\mathcal{L}(v)] - \{\min_v \mathcal{L}\} - h\sigma^2) \leq -(\mathbb{E}[\mathcal{L}(v)] - \{\min_v \mathcal{L}\})\partial t + h\sigma^2\partial t \tag{142}$$

$$\partial(\hat{\mathcal{L}} - \{\min_v \mathcal{L}\} - h\sigma^2) \leq -(\hat{\mathcal{L}} - \{\min_v \mathcal{L}\} - h\sigma^2)\partial t \tag{143}$$

$$//\text{When } \hat{\mathcal{L}} - \{\min_v \mathcal{L}\} - h\sigma^2 > 0 \tag{144}$$

$$\partial \ln|\hat{\mathcal{L}} - \{\min_v \mathcal{L}\} - h\sigma^2| \leq -1\partial t \tag{145}$$

$$\ln|\hat{\mathcal{L}} - \{\min_v \mathcal{L}\} - h\sigma^2| \leq \ln|\widehat{\mathcal{L}(v_0)} - \{\min_v \mathcal{L}\} - h\sigma^2| - t \tag{146}$$

$$\hat{\mathcal{L}} - \{\min_v \mathcal{L}\} - h\sigma^2 \leq e^{-t}(\widehat{\mathcal{L}(v_0)} - \{\min_v \mathcal{L}\} - h\sigma^2) \tag{147}$$

$$\hat{\mathcal{L}} \leq e^{-t}(\widehat{\mathcal{L}(v_0)} - \{\min_v \mathcal{L}\} - h\sigma^2) + \{\min_v \mathcal{L}\} + h\sigma^2 \tag{148}$$

$$\hat{\mathcal{L}} \leq e^{-t}\widehat{\mathcal{L}(v_0)} + (1 - e^{-t})(\{\min_v \mathcal{L}\} + h\sigma^2) \tag{149}$$

$$\hat{\mathcal{L}} \leq e^{-t}\widehat{\mathcal{L}(v_0)} + (1 - e^{-t})(\gamma + h\sigma^2) \tag{150}$$

$$\square$$

When we substitute the initial loss $\mathcal{L}(v_0)$ with the hyper-parameters we use in the random initialization, we obtain the following corollary.

**Corollary E.15** (Expected loss upper bound of linear probing from random initialization). *If we initialize the linear head by $v_{t=0} \sim \mathcal{N}(0, I_{h \times h})$, then the expected loss is upper bounded by*

$$\mathbb{E}[\mathcal{L}_{\mathrm{lp}}(t)] \leq \frac{1}{2}(h\beta + \|Y\|^2)e^{-t} + (1 - e^{-t})(\gamma + h\sigma^2) \tag{151}$$

*Proof of Corollary E.15.* The result is immediate when we combine Lemma E.13 and Theorem 4.4. $\square$

### E.4 Imbalance matrix from linear probing

In the convergence analysis of fine-tuning, we eliminate variables and simplify the Langevin dynamics by the imbalance matrix. In this part, we characterize how the imbalance matrix changes in the linear probing phase. The following results will later help us analyze LP-FT.

**Lemma E.16** (Eigenvalues of imbalance matrix at the beginning of fine-tuning)**.** *During the linear probing phase (Equation equation 121), for the imbalance matrix defined in Definition E.8,*

1. *the minimum eigenvalue of the imbalance matrix is always $-1$;*

2. *other eigenvalues evolve in this way:*

$$\mathbb{E}[\lambda] = \mathbb{E}\left[\|v\|_2^2\right] - 1 \geq -1 \tag{152}$$

*Proof of Lemma E.16.* Consider any eigenpair $(\lambda, u)$ of matrix $D$, we have

$$Du = \lambda u \tag{153}$$
$$(vv^T - B_0 B_0^T)u = \lambda u \tag{154}$$
$$(vv^T - I_{h \times h})u = \lambda u \tag{155}$$
$$(v^T u)v = (\lambda + 1)u \tag{156}$$
$$\tag{157}$$

We can take any $u \perp v$ and $(u, -1)$ is an eigenpair of $D$. So $-1$ is always an eigenvalue of $D$. We need to discuss two different cases here:

1. If $\lambda = -1$, we only know that $u \perp v$.

2. If $\lambda \neq -1$, then $v$ and $u$ are parallel. Say $u = \alpha v$, then

$$u = \frac{v^T u}{\lambda + 1} v \tag{158}$$
$$\alpha v = \frac{\alpha \|v\|_2^2}{\lambda + 1} v \tag{159}$$
$$\implies \lambda = \|v\|_2^2 - 1 \geq -1 \tag{160}$$

$\square$

**Proposition E.17** (Expected eigenvalue of imbalance matrix at the beginning of fine-tuning)**.** *Say we run linear probing for time $t$. If we initialize the linear head by $v_{t=0} \sim \mathcal{N}(0, I_{h \times h})$, then for the imbalance matrix defined in Definition E.8, we have*

$$\mathbb{E}[\|v\|^2] = h\beta e^{-2t} + 2\|B_0 X^T Y\|^2 (e^{-t} - e^{-2t}) + (\|B_0 X^T Y\|^2 + h\sigma^2)(1 - e^{-2t}) \tag{161}$$

*throughout the linear probing process. Then by Lemma E.16, for those eigenvalues not equal to $-1$, we have*

$$\mathbb{E}[\lambda] = \mathbb{E}\left[\|v\|_2^2\right] - 1 = h\beta e^{-2t} + 2\|B_0 X^T Y\|^2 (e^{-t} - e^{-2t}) + (\|B_0 X^T Y\|^2 + h\sigma^2)(1 - e^{-2t}) - 1 \tag{162}$$

*at the beginning of fine-tuning after linear probing.*

*Proof of Proposition E.17.* By Equation equation 121, the Langevin diffusion of linear probing is:

$$\partial v = -B_0 X^T (X B_0^T v - Y)\partial t + \sqrt{2\sigma^2}\partial W_t = -v\partial t + B_0 X^T Y \partial t + \sqrt{2\sigma^2}\partial W_t \tag{163}$$

We consider the evolution of $v^T v$: by Itô's formula (Equation equation E.1)

$$\partial v^T v = 2v^T \partial v + (\partial v)^T I_h (\partial v) \tag{164}$$
$$\partial v^T v = -2v^T(v - B_0 X^T Y)\partial t + 2v^T \sqrt{2\sigma^2}\partial W_t + 2h\sigma^2 \partial t \tag{165}$$
$$\partial v^T v = (-2v^T v + 2v^T B_0 X^T Y)\partial t + 2v^T \sqrt{2\sigma^2}\partial W_t + 2h\sigma^2 \partial t \tag{166}$$

To solve the above equation, we need to solve the dynamics of $v^T B_0 X^T Y$:

$$\partial Y^T X B_0^T v = - Y^T X B_0^T (v - B_0 X^T Y) \partial t + \sqrt{2\sigma^2} \partial W_t \tag{167}$$

$$\partial \mathbb{E}[Y^T X B_0^T v] = - \mathbb{E}[Y^T X B_0^T v] dt + \|B_0 X^T Y\|^2 \partial t \tag{168}$$

$$\frac{\partial}{\partial t} \mathbb{E}[Y^T X B_0^T v - \|B_0 X^T Y\|^2] = - \mathbb{E}[Y^T X B_0^T v - \|B_0 X^T Y\|^2] \tag{169}$$

$$\frac{\partial}{\partial t} \ln |\mathbb{E}[Y^T X B_0^T v - \|B_0 X^T Y\|^2]| = - 1 \tag{170}$$

$$|\mathbb{E}[Y^T X B_0^T v_t - \|B_0 X^T Y\|^2]| = |\mathbb{E}[Y^T X B_0^T v_0 - \|B_0 X^T Y\|^2]| \cdot \exp(-t) \tag{171}$$

When we initialize the linear head by $v_{t=0} \sim \mathcal{N}(0, I_{h \times h})$, we have $\mathbb{E}[Y^T X B_0^T v_0] = 0$. Then

$$|\mathbb{E}[Y^T X B_0^T v_t - \|B_0 X^T Y\|^2]| = |\mathbb{E}[Y^T X B_0^T v_0 - \|B_0 X^T Y\|^2]| \cdot \exp(-t) \tag{172}$$

$$\mathbb{E}[\|B_0 X^T Y\|^2 - Y^T X B_0^T v_t] = \mathbb{E}[\|B_0 X^T Y\|^2 - Y^T X B_0^T v_0] \cdot \exp(-t) \tag{173}$$

So we can rewrite Equation equation 166 as:

$$\partial \mathbb{E}[\|v\|^2] = (-2\mathbb{E}[\|v\|^2] + 2\mathbb{E}[v^T B_0 X^T Y]) \partial t + 2h\sigma^2 \partial t \tag{174}$$

$$\partial \mathbb{E}[\|v\|^2] = (-2\mathbb{E}[\|v\|^2] + 2(\mathbb{E}[\|B_0 X^T Y\|^2 - Y^T X B_0^T v_0] \cdot \exp(-t) + \|B_0 X^T Y\|^2)) \partial t + 2h\sigma^2 \partial t \tag{175}$$

$$\frac{1}{2} \frac{\partial}{\partial t} \mathbb{E}[\|v\|^2] = - \mathbb{E}[\|v\|^2] + \mathbb{E}[\|B_0 X^T Y\|^2 - Y^T X B_0^T v_0] \cdot \exp(-t) + (\|B_0 X^T Y\|^2 + h\sigma^2) \tag{176}$$

Let $a_1 = \mathbb{E}[\|B_0 X^T Y\|^2 - Y^T X B_0^T v_0], a_2 = \|B_0 X^T Y\|^2 + h\sigma^2, f(t) = \mathbb{E}[\|v\|^2]$ and rewrite the above equation:

$$\frac{1}{2} f'(t) + f(t) = a_1 e^{-t} + a_2 \tag{177}$$

$$f'(t) + 2f(t) = 2a_1 e^{-t} + 2a_2 \tag{178}$$

$$e^{2t} f'(t) + 2e^{2t} f(t) = 2a_1 e^t + 2a_2 e^{2t} \tag{179}$$

$$e^{2t} f(t) \Big|_0^t = (2a_1 e^t + a_2 e^{2t}) \Big|_0^t \tag{180}$$

$$e^{2t} f(t) = f(0) + 2a_1 (e^t - 1) + a_2 (e^{2t} - 1) \tag{181}$$

$$f(t) = f(0) e^{-2t} + 2a_1 (e^{-t} - e^{-2t}) + a_2 (1 - e^{-2t}) \tag{182}$$

Since we initialize the linear head by $v_{t=0} \sim \mathcal{N}(0, I_{h \times h})$, we have $f(0) = h\beta$ and $a_1 = \|B_0 X^T Y\|^2$. $\qquad \square$

**Lemma E.18** (Imbalance matrix in fine-tuning). *During fine-tuning (Equation equation 122), the imbalance matrix D in Definition E.8 evolves as*

$$\frac{\partial}{\partial t} \mathbb{E}[D] = (1 - d) \sigma^2 I_{h \times h} \tag{183}$$

*where d is the dimension of data inputs ($B \in \mathbb{R}^{h \times d}$).*

*Proof of Lemma E.9.* We prove this lemma by analyzing the infinitesimal generator $A$ of imbalance matrix $D$ at any time:

$$A(D)_{ij} := \lim_{t \downarrow 0} \frac{\mathbb{E}^D[(D(t))_{ij}] - (D)_{ij}}{t} \tag{184}$$

$$= 0 + \sigma^2 \sum_{i' \in [h]} \sum_{j' \in [h]} \mathbf{1}[i' = j' = i = j] \tag{185}$$

$$- \sigma^2 \sum_{i' \in [h], j' \in [d]} \sum_{i'' \in [h], j'' \in [d]} \mathbf{1}[i' = i'' = i = j \text{ and } j' = j''] \tag{186}$$

the generator is zero for $i \neq j$. So we can just consider the case where $i = j$.

$$A(D)_{ii} = \sigma^2 \sum_{i' \in [h]} \sum_{j' \in [h]} \mathbf{1}[i' = j' = i] \tag{187}$$

$$- \sigma^2 \sum_{i' \in [h], j' \in [d]} \sum_{i'' \in [h], j'' \in [d]} \mathbf{1}[i' = i'' = i \text{ and } j' = j''] \tag{188}$$

$$= (1 - d)\sigma^2 \tag{189}$$

$\square$

**Lemma E.19** (Monotonic eigenvalue of imbalance matrix in fine-tuning). *Denote $D_{\mathrm{lp}}$ as the imbalance matrix right after linear probing phase. All eigenvalues of the imbalance matrix are decreasing in expectation during fine-tuning. Specifically,*

$$\mathbb{E}[\lambda(D)] = \mathbb{E}[\lambda(D_{\mathrm{lp}})] + (1 - d)\sigma^2 t \tag{190}$$

*where $t$ is the time-span of fine-tuning process.*

*Proof of Lemma E.19.* Pick any eigenpair $(\lambda, u)$ of imbalance matrix $D$ (Definition E.8) such that $\|u\|_2 = 1$. By Itô's lemma (Equation equation E.1):

$$\partial \lambda = u^T (\partial D) u + u^T (\partial D)(\lambda I - D)^\dagger (\partial D) u^T \tag{191}$$

$$= (1 - d)\sigma^2 \|u\|_2^2 \partial t + \partial M_t + (1 - d)^2 \sigma^4 u^T (\lambda I - D)^\dagger u^T \tag{192}$$

$$= (1 - d)\sigma^2 \partial t + \partial M_t + (1 - d)^2 \sigma^4 u^T (\lambda I - D)^\dagger u^T \tag{193}$$

where $M_t$ is the martingale induced by the Brownian noise and $(\cdot)^\dagger$ denotes the pseudo inverse of a certain matrix. Say the the singular value decomposition (SVD) of $D$ is

$$D = U \Sigma U^T = U \begin{bmatrix} \lambda_1 & & & \mathbf{0} \\ & \lambda_2 & & \\ & & \ddots & \\ \mathbf{0} & & & \end{bmatrix} U^T \tag{194}$$

where we have $\lambda \in \mathrm{diag}\Sigma$ and $u$ being a column vector in $U$. So we can write the SVD of $(\lambda I - D)$ as:

$$\lambda I - D = V \Sigma' V^T = V \begin{bmatrix} \lambda - \lambda_1 & & & \mathbf{0} \\ & \lambda - \lambda_2 & & \\ & & \ddots & \\ \mathbf{0} & & & \end{bmatrix} V^T \tag{195}$$

where we obtain $V$ by removing $u$ in the columns of $U$ and we obtain $\Sigma'$ by removing $\lambda$ in $\Sigma$. Then the pseudo inverse of $(\lambda I - D)$ is

$$(\lambda I - D)^\dagger = V \Sigma' V^T = V \begin{bmatrix} \frac{1}{\lambda - \lambda_1} & & & \mathbf{0} \\ & \frac{1}{\lambda - \lambda_2} & & \\ & & \ddots & \\ \mathbf{0} & & & \end{bmatrix} V^T \tag{196}$$

Since $U$ is orthogonal, we shall have $V^T u = \mathbf{0}$. Then we can rewrite the stochastic dynamics of $D$ as:

$$\frac{\partial}{\partial t} \mathbb{E}[\lambda] = (1 - d)\sigma^2 \tag{197}$$

$\square$

### E.5  Fine-tuning loss

**Lemma E.20** (Bounding the norm of linear head $\|v\|_2^2$). *During fine-tuning (Equation equation 122), we can bound the norm of $\|v\|_2^2$ with the imbalance matrix $D$ in Definition E.8 as*

$$\frac{\underline{\lambda} + \sqrt{\underline{\lambda}^2 + 4\|w\|^2}}{2} \le \|v\|_2^2 \le \frac{\bar{\lambda} + \sqrt{\bar{\lambda}^2 + 4\|w\|^2}}{2} \tag{198}$$

*where we denote $\underline{\lambda} = \lambda_{\min}(\hat{D}), \bar{\lambda} = \lambda_{\max}(\hat{D})$.*

*Proof of Lemma E.20.* Given the information of imbalance matrix, we can bound the linear head norm. Denote $\underline{\lambda} = \lambda_{\min}(D), \bar{\lambda} = \lambda_{\max}(D)$. Denote $w = B^T v$ and multiply $D$ with $v$ on both sides:

$$v^T D v = (v^T v)^2 - (v^T B)(B^T v) \tag{199}$$

$$v^T D v = \|v\|_2^4 - \|w\|_2^2 \tag{200}$$

We have a range for the Rayleigh quotient: $\frac{x^T D x}{x^T x} \in [\underline{\lambda}, \bar{\lambda}]$. So we obtain two inequalities:

$$\begin{cases} \|v\|_2^4 - \|w\|_2^2 \ge \underline{\lambda}\|v\|_2^2 \\ \|v\|_2^4 - \|w\|_2^2 \le \bar{\lambda}\|v\|_2^2 \end{cases} \tag{201}$$

$$= \begin{cases} \|v\|^4 - \underline{\lambda}\|v\|^2 - \|w\|^2 \ge 0 \\ \|v\|^4 - \bar{\lambda}\|v\|^2 - \|w\|^2 \le 0 \end{cases} \tag{202}$$

To get a lower bound of $v$, we can solve two quadratic inequalities. For the first quadratic equation, since the smaller root is non-positive, $\underline{\lambda} - \sqrt{\underline{\lambda}^2 + 4\|w\|^2} \le 0$, we just bound $\|v\|^2$ with the larger root:

$$\|v\|^2 \ge \frac{\underline{\lambda} + \sqrt{\underline{\lambda}^2 + 4\|w\|^2}}{2} \tag{203}$$

similarly, for the second quadratic equation, we obtain an upper bound for $\|v\|^2$ with the right-side zero point:

$$\|v\|^2 \le \frac{\bar{\lambda} + \sqrt{\bar{\lambda}^2 + 4\|w\|^2}}{2} \tag{204}$$

$$\square$$

**Lemma E.21** (Bounding eigenvalues of $B^T B$ (re-stated from Min et al. (2023b))). *During fine-tuning (Equation equation 122), we can bound any nonzero eigenvalue $\lambda_i$ of $B^T B$ as*

$$\lambda_i \in \left[ \frac{-\bar{\lambda} + \sqrt{\bar{\lambda}^2 + 4(z_i^T w)^2}}{2}, \frac{-\underline{\lambda} + \sqrt{\underline{\lambda}^2 + 4(z_i^T w)^2}}{2} \right] \tag{205}$$

*where we use the imbalance matrix $D$ in Definition E.8 and denote*

$$\begin{cases} \bar{\lambda} = \lambda_{\max}(D) \\ \underline{\lambda} = \lambda_{\min}(D) \end{cases} \tag{206}$$

*Proof of Lemma E.21.* The proof of this lemma follows the proof of Lemma 3 in Min et al. (2023b). $B^T B$ is symmetric and positive semidefinite ($x^T B^T B x = \|Bx\|_2^2 \ge 0$). So every eigenvalue of $B^T B$ is non-negative.

$D$ has at most one positive eigenvalue: if $D$ has more than one eigenvalues, then the subspace of $\mathbb{R}^h$ spanned by the all positive eigenvectors has dimension at least 2, which must have non-trivial intersection with

$\ker(v^T)$ as $\dim(\ker(v^T)) = h - 1$. Then there exists a nonzero vector $z \in \ker(v^T)$ such that $z^T D z > 0$, which would imply $-z^T B B^T z = z^T D z > 0$, a contradiction.

For any eigenvalue-eigenvector pair $(\lambda_i, z_i)$ of $B^T B$ where $\lambda_i \neq 0$ and $z_i \in \mathbb{S}^{d-1}$,

$$\lambda_i^2 = z_i^T (B^T B)^2 z_i \tag{207}$$

$$//\text{replace something with imbalance matrix} \tag{208}$$

$$\lambda_i^2 = (z_i^T w)^2 - z_i^T B^T D B z_i \tag{209}$$

$$\lambda_i^2 - (z_i^T w)^2 = -z_i^T B^T D B z_i \tag{210}$$

$$\lambda_i^2 - (z_i^T w)^2 \in (z_i^T (B^T B) z_i) \cdot [-\lambda_{\max}, -\lambda_{\min}] \tag{211}$$

$$\lambda_i^2 - (z_i^T w)^2 \in \lambda_i \cdot [-\lambda_{\max}, -\lambda_{\min}] \tag{212}$$

again, we can rewrite this as two quadratic inequalities

$$\begin{cases} \lambda_i^2 + \lambda_{\max} \lambda_i - (z_i^T w)^2 \geq 0 \\ \lambda_i^2 + \lambda_{\min} \lambda_i - (z_i^T w)^2 \leq 0 \end{cases} \tag{213}$$

from them we know that there are two possible intervals:

$$\begin{cases} \lambda_i \in \left[ -\infty, \frac{-\lambda_{\max} - \sqrt{\lambda_{\max}^2 + 4(z_i^T w)^2}}{2} \right] \cup \left[ \frac{-\lambda_{\max} + \sqrt{\lambda_{\max}^2 + 4(z_i^T w)^2}}{2}, +\infty \right] \\ \lambda_i \in \left[ \frac{-\lambda_{\min} - \sqrt{\lambda_{\min}^2 + 4(z_i^T w)^2}}{2}, \frac{-\lambda_{\min} + \sqrt{\lambda_{\min}^2 + 4(z_i^T w)^2}}{2} \right] \end{cases} \tag{214}$$

Note that we must have $\lambda_i \geq 0$ since $B^T B$ is positive semidefinite. So we can rewrite the bounds:

$$\lambda_i \in \left[ \frac{-\lambda_{\max} + \sqrt{\lambda_{\max}^2 + 4(z_i^T w)^2}}{2}, \frac{-\lambda_{\min} + \sqrt{\lambda_{\min}^2 + 4(z_i^T w)^2}}{2} \right] \tag{215}$$

since the function $f(x) = -x + \sqrt{x + c^2}$ is monotonically decreasing, we have $f(\lambda_{\max}) \leq f(\lambda_{\min})$, i.e. the lower bound is no greater than the upper bound, i.e. the above interval is always non-empty. $\qquad\square$

### E.6 Numerical conjecture on the eigenvalues

**Conjecture E.22** (Small relative error induced by Jensen gap (Equation 247))**.** We denote the minimum eigenvalue of the imbalance matrix $D$ as $\underline{\lambda}$. The relative error $\frac{\mathbb{E}[\max(0, -\underline{\lambda})^{1/2}]^2 - \mathbb{E}[\underline{\lambda}]}{\mathbb{E}[\max(0, -\underline{\lambda})^{1/2}]^2}$ increases slowly in time and is smaller than 1% under reasonable number of training epochs. Here we provide an empirical example with huge noise scale (much greater than the common noise scale in real-world applications). We observe that the relative approximation error is insignificant even with huge noise scale.

### E.7 Fine-tuning loss upper bound

**Lemma E.23** (Imbalance matrix in fine-tuning under layerwise noise)**.** *During fine-tuning (Equation* (127)*), the imbalance matrix $D$ in Definition E.8 evolves as*

$$\mathbb{E}\left[ \frac{dD}{dt} \right] = 0 \tag{216}$$

*Proof of Lemma E.23.* We prove this lemma by analyzing the infinitesimal generator $A$ of imbalance matrix $D$:

$$A(D_0(v, B))_{ij} := \lim_{t \downarrow 0} \frac{\mathbb{E}^{D_0}[D_{ij}] - (D_0)_{ij}}{t} \tag{217}$$

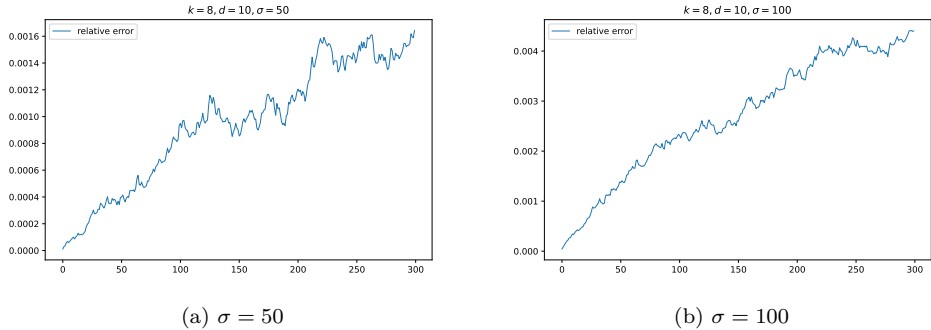

(a) $\sigma = 50$

(b) $\sigma = 100$

Figure 10: Growth of the relative error $\frac{\mathbb{E}[\max(0,-\lambda)^{1/2}]^2 - \mathbb{E}[\lambda]}{\mathbb{E}[\max(0,-\lambda)^{1/2}]^2}$ in the experiment setting: (1) we use a two-layer linear network with a linear head of size $h = 8$ and a feature extractor of size $h \times d = 8 \times 10$; (2) we train the linear network with DP-SGD; (3) we repeat the experiment with large noise multipliers $\sigma = 50$ and $\sigma = 100$.

$$=0 + \sigma^2 \sum_{i' \in [h]} \sum_{j' \in [h]} \mathbf{1}[i' = j' = i = j] \tag{218}$$

$$- \sigma^2 \sum_{i' \in [h], j' \in [d]} \sum_{i'' \in [h], j'' \in [d]} \mathbf{1}[i' = i'' = i = j \text{ and } j' = j''] \tag{219}$$

the generator is zero for $i \neq j$. So we can just consider the case where $i = j$.

$$A(D_0(v, B))_{ii} = \sigma^2 \sum_{i' \in [h]} \sum_{j' \in [h]} \mathbf{1}[i' = j' = i] \tag{220}$$

$$- \sigma^2 \sum_{i' \in [h], j' \in [d]} \sum_{i'' \in [h], j'' \in [d]} \mathbf{1}[i' = i'' = i \text{ and } j' = j''] \tag{221}$$

$$= (d - d)\sigma^2 \tag{222}$$

$$= 0 \tag{223}$$

$\square$

**Theorem E.24** (Loss upper bound of fine-tuning). *In fine-tuning under layerwise noise (Equation equation 127), we have*

$$\mathbb{E}[\mathcal{L}] \lesssim \mathbb{E}[\mathcal{L}]e^{(-\bar{\lambda} + \sqrt{2}\sigma^2(1+d))t} + L^{\square}(1 - e^{(-\bar{\lambda} + \sqrt{2}\sigma^2(1+d))t}) \tag{224}$$

*where* $L^{\square} = \sigma^2 \frac{(1+d)\|X^T Y\| - d\underline{\lambda}}{\bar{\lambda} - \sqrt{2}\sigma^2(1+d)}.$

*Proof of Theorem 4.5.* We first simplify the loss dynamics:

$$\partial \mathcal{L} = \partial \frac{1}{2} \|XB^T v - Y\|^2 \tag{225}$$

$$= \frac{1}{2} \left\langle \nabla_v \|XB^T v - Y\|^2, \partial v \right\rangle + \frac{1}{2} \left\langle \nabla_B \|XB^T v - Y\|^2, \mathrm{vec}(\partial B) \right\rangle \tag{226}$$

$$+ \frac{1}{4} (\partial v)^T H_{\|XB^T v - Y\|^2}(\partial v) + \frac{1}{4} [\mathrm{vec}(\partial B)]^T H_{\|XB^T v - Y\|^2} \mathrm{vec}(\partial B) \tag{227}$$

$$= (XB^T v - Y)^T X B^T \partial v + (XB^T v - Y)^T X (\partial B)^T v \tag{228}$$

$$+ \frac{1}{2} (\partial v)^T BB^T (\partial v) + \frac{1}{2} [\mathrm{vec}(\partial B)]^T H_{\|XB^T v - Y\|^2} \mathrm{vec}(\partial B) \tag{229}$$

$$= -(XB^T v - Y)^T X B^T B X^T (XB^T v - Y)\partial t + (XB^T v - Y)^T X B^T \sqrt{2\sigma^2 d}\partial W_t \tag{230}$$

$$- (XB^T v - Y)^T X X^T (XB^T v - Y)v^T v \partial t + (XB^T v - Y)^T X (\sqrt{2\sigma^2}\partial W_t')v \tag{231}$$

$$+ \sigma^2 \mathrm{trace}(BB^T)\partial t + \sigma^2 d\|v\|^2 \partial t \tag{232}$$

$$= -(B^T v - X^T Y)^T B^T B(B^T v - X^T Y)\partial t + (B^T v - X^T Y)^T B^T \sqrt{2\sigma^2}\partial W_t \tag{233}$$

$$- (B^T v - X^T Y)^T (B^T v - X^T Y) v^T v \partial t + (B^T v - X^T Y)^T (\sqrt{2\sigma^2} \partial W'_t) v \tag{234}$$

$$+ \sigma^2 \mathrm{trace}(B^T B) \partial t + \sigma^2 d \|v\|^2 \partial t \tag{235}$$

By Lemma E.20 and Lemma E.21, we have

$$\partial \mathbb{E}\mathcal{L} = - \mathbb{E}[(w - X^T Y)^T (B^T B + v^T v I_{d\times d})(w - X^T Y)] \partial t + \sigma^2 \mathbb{E}[\|B\|_F^2 + d\|v\|_2^2] \partial t \tag{236}$$

$$\leq \mathbb{E}\left\{ -\|w - X^T Y\|_2^2 \frac{\underline{\lambda} + \sqrt{\underline{\lambda}^2 + 4\|w\|^2}}{2} \partial t - \|w - X^T Y\|_2^2 \frac{-\bar{\lambda} + \sqrt{\bar{\lambda}^2 + 4(z_{\min}^T w)^2}}{2} \partial t \right\} \tag{237}$$

$$+ \mathbb{E}\left\{ \sigma^2 d \frac{-\underline{\lambda} + \sqrt{\underline{\lambda}^2 + 4(z_{\min}^T w)^2}}{2} \partial t + \sigma^2 d \frac{\bar{\lambda} + \sqrt{\bar{\lambda}^2 + 4\|w\|^2}}{2} \partial t \right\} \tag{238}$$

$$\leq - \frac{1}{2} \mathbb{E}[\|w - X^T Y\|_2^2 (\Lambda_{\min} + \Lambda_{\max})] \partial t + \frac{1}{2} \sigma^2 \mathbb{E}[d\Gamma_{\min} + \Gamma_{\max}] \partial t \tag{239}$$

where we define

$$\begin{cases} \Lambda_{\min} = \underline{\lambda} + \sqrt{\underline{\lambda}^2 + 4\|w\|^2} \geq \max(0, 2\underline{\lambda}) \\ \Lambda_{\max} = -\bar{\lambda} + \sqrt{\bar{\lambda}^2 + 4(z_{\min}^T w)^2} \geq \max(0, -2\bar{\lambda}) \\ \Gamma_{\min} = -\underline{\lambda} + \sqrt{\underline{\lambda}^2 + 4(z_{\min}^T w)^2} \leq \max(2\|w\|, 2\|w\| - 2\underline{\lambda}) = 2\|w\| + 2\max(0, -\underline{\lambda}) \\ \Gamma_{\max} = \bar{\lambda} + \sqrt{\bar{\lambda}^2 + 4\|w\|^2} \leq \max(2\|w\|, 2\|w\| + 2\bar{\lambda}) = 2\|w\| + 2\max(0, \bar{\lambda}) \end{cases} \tag{240}$$

Denote the probability measure of the state at time $t$ as $\nu_t$. Then by using Jensen's inequality, reverse Hölder's inequality, etc., we can bound the first term:

$$\mathbb{E}[\|w - w_*\|_2^2 (\Lambda_{\min} + \Lambda_{\max})] = \int \|w - w_*\|_2^2 (\Lambda_{\min} + \Lambda_{\max}) d\nu_t \tag{241}$$

$$\geq \left( \int \|w - w_*\|_2^{-1} d\nu_t \right)^{-2} \left( \int (\Lambda_{\min} + \Lambda_{\max})^{1/2} d\nu_t \right)^2 \tag{242}$$

$$= \mathbb{E}[\|w - w_*\|_2^{-1}]^{-2} \mathbb{E}[(\Lambda_{\min} + \Lambda_{\max})^{1/2}]^2 \tag{243}$$

$$\geq \mathbb{E}[\|w - w_*\|_2^2] \mathbb{E}[(\Lambda_{\min} + \Lambda_{\max})^{1/2}]^2 \tag{244}$$

according our empirical observation (Conjecture E.22) $\tag{245}$

we ignore the Jensen gap for the second multiplier $\tag{246}$

$$\gtrapprox - \frac{1}{2} \mathbb{E}[\|w - w_*\|_2^2] \mathbb{E}[\bar{\lambda}] \tag{247}$$

By Lemma E.19 $\tag{248}$

$$= \mathbb{E}[\|w - w_*\|_2^2](-\mathbb{E}[\bar{\lambda}(D_0)] + (d - 1)\sigma^2 t) \tag{249}$$

$$= 2(-\mathbb{E}[\bar{\lambda}(D_0)] + (d - 1)\sigma^2 t) \cdot \mathbb{E}[\mathcal{L}] \tag{250}$$

Then we rewrite the upper bound:

$$\partial \mathbb{E}[\mathcal{L}] \leq - \frac{1}{2} \mathbb{E}[\|w - X^T Y\|_2^2 (\Lambda_{\min} + \Lambda_{\max})] \partial t + \frac{1}{2} \sigma^2 \mathbb{E}[d\Gamma_{\min} + \Gamma_{\max}] \partial t \tag{251}$$

$$\partial \mathbb{E}[\mathcal{L}] \lessapprox - \bar{\lambda} \mathbb{E}[\mathcal{L}] \partial t + \sigma^2 (\sqrt{2}(1 + d) \mathbb{E}[\mathcal{L}]^{1/2} + (1 + d)\|X^T Y\| - d\underline{\lambda}) \partial t \tag{252}$$

$$\partial \mathbb{E}[\mathcal{L}] \lessapprox (-\bar{\lambda} + \sqrt{2}\sigma^2 (1 + d)) \mathbb{E}[\mathcal{L}] \partial t + \sigma^2 ((1 + d)\|X^T Y\| - d\underline{\lambda}) \partial t \tag{253}$$

$$\mathbb{E}[\mathcal{L}] \lessapprox \mathbb{E}[\mathcal{L}] e^{(-\bar{\lambda} + \sqrt{2}\sigma^2 (1+d))t} + L^{\square} (1 - e^{(-\bar{\lambda} + \sqrt{2}\sigma^2 (1+d))t}) \tag{254}$$

where $L^{\square} = \sigma^2 \frac{(1+d)\|X^T Y\| - d\underline{\lambda}}{\bar{\lambda} - \sqrt{2}\sigma^2 (1+d)}$. $\qquad\qquad\square$

## F  Theory with Clipping

In this section, we present the **first** theoretical investigation on Langevin diffusion **with clipping**. We believe that our contribution is significant for the Langevin diffusion and private optimization research community. We summarize our findings and contributions in the following list:

- A new definition for Langevin diffusion with clipping (Definition F.1).

- Zeroth order approximation error for the clipped Langevin diffusion (Theorem F.3).

- Privacy guarantee for the clipped Langevin diffusion (Theorem F.4).

- The exact "discrete vs. continuous" algebraic correspondence between the clipped Langevin diffusion and vanilla DP-SGD (Remark F.2).

- Feature distortion analysis for the clipped Langevin diffusion (Theorem F.5).

- The existence proof of a unique strong solution for the clipped Langevin diffusion (Corollary F.7).

**Definition F.1** (Clipped Langevin diffusion). Say we work on parameter $\theta \in \mathbb{R}^p$ to minimize a group of loss functions $\{\ell_i\}_{i \in [n]}$. The parameter evolve according to the following stochastic differential equation.

$$\partial\theta = -\sum_{i \in [n]} \mathrm{clip}_C(\nabla\ell_i(\theta))\partial t + \sigma\partial\xi_t \tag{255}$$

This equation is the clipped Langevin diffusion. $\xi_t$ is a vector containing $p$ independent 1-dimensional Brownian motion. The clipping function is defined by a constant $C > 0$ and

$$\mathrm{clip}_C(\nabla\ell_i(\theta)) := \min\left(1, \frac{C}{\|\nabla\ell_i(\theta)\|_2}\right)\nabla\ell_i(\theta).$$

This definition allows us to establish the first exact "discrete vs. continuous" algebraic correspondence between clipped Langevin diffusion and vanilla DP-SGD, creating a continuous analytical framework that closely mirrors real DP-SGD implementations.

*Remark* F.2 (Algebraic correspondence between the clipped Langevin diffusion and DP-SGD). The update rule of the vanilla DP-SGD with step-size $\eta > 0$ can be written as (Abadi et al., 2016):

$$\theta_{k+1} = \theta_k - \eta\frac{1}{|B|}\sum_{i \in \mathcal{B}_k}\left(\mathrm{clip}_C(\nabla\ell_i(\theta)) + \sigma\mathcal{N}(0, C^2\mathbf{I})\right) \tag{256}$$

where $B$ is the batch size and $\mathcal{B}_k$ is the batch of data points sampled at step $k$. We can rewrite the update rule by assuming full sampling, $\tilde{\eta} = \eta\frac{1}{|B|}$ and $\tilde{\sigma} = \sigma C$:

$$\theta_{k+1} = \theta_k - \tilde{\eta}\sum_{i \in [n]}\left(\mathrm{clip}_C(\nabla\ell_i(\theta)) + \tilde{\sigma}\mathcal{N}(0, \mathbf{I})\right) \tag{257}$$

One can compare this update rule with the clipped Langevin diffusion (Equation (255)):

$$\partial\theta = -\sum_{i \in [n]} \mathrm{clip}_C(\nabla\ell_i(\theta))\partial t + \sigma\partial\xi_t \tag{258}$$

It is easy to see the algebraic correspondence between the above two equations. We provide a rigorous derivation of DP-SGD update by discretizing the clipped Langevin diffusion with the Euler–Maruyama method.

Suppose that we want to solve the clipped Langevin diffusion on some interval of time $[0, T]$. Then the Euler–Maruyama approximation to the true solution $\theta$ is the Markov chain $\tilde{\theta}$ defined as follows:

- Partition the interval $[0, T]$ into $K$ equal subintervals of width $\tilde{\eta} > 0$:

$$0 = \tau_0 < \tau_1 < \cdots < \tau_K = T \text{ and } \tilde{\eta} = \frac{T}{K} \tag{259}$$

- Let $\tilde{\theta}_0 = \theta_0$ at the initialization.
- Iteratively compute $\tilde{\theta}_k$ for $1 \le k \le K$ by

$$\tilde{\theta}_k = \tilde{\theta}_{k-1} - \eta \sum_{i \in [n]} \left( \text{clip}_C(\nabla \ell_i(\tilde{\theta}_{k-1})) + \sigma \mathcal{N}(0, \mathbf{I}) \right) \tag{260}$$

In this way, we rediscover the update rules for DP-SGD by discretizing the clipped Langevin diffusion.

We give an approximation error bound following (Freidlin et al., 2012, Theorem 1.2, Chapter 2.1).

**Theorem F.3** (Zeroth order approximation error). *For all $t > 0, \delta > 0$, we have*

$$\mathbb{E}\left[ \left\| \theta_t - \theta_t^{(0)} \right\|^2 \right] \le \left( \sigma(2p)^{\frac{1}{2}} t^{\frac{1}{2}} + 2nCt \right)^2 \tag{261}$$

*Proof of Theorem F.3.*

$$\mathbb{E}[\partial \| \theta_t - \theta_t^{(0)} \|^2] = \mathbb{E}[\langle \theta_t - \theta_t^{(0)}, \partial \theta_t - \partial \theta_t^{(0)} \rangle + 2p\sigma^2 \partial t] \tag{262}$$

$$\partial \mathbb{E}[\| \theta_t - \theta_t^{(0)} \|^2] \le \mathbb{E}[4nC \| \theta_t - \theta_t^{(0)} \| \partial t + 2p\sigma^2 \partial t] \tag{263}$$

$$\mathbb{E}[\| \theta_t - \theta_t^{(0)} \|^2] \le \int_0^T (4nC \cdot \mathbb{E}[\| \theta_t - \theta_t^{(0)} \|] + 2p\sigma^2) \partial t \tag{264}$$

$$\mathbb{E}[\| \theta_t - \theta_t^{(0)} \|^2] \le \int_0^T (4nC \cdot \sqrt{\mathbb{E}[\| \theta_t - \theta_t^{(0)} \|^2]} + 2p\sigma^2) \partial t \tag{265}$$

$$\mathbb{E}[\| \theta_t - \theta_t^{(0)} \|^2] \le 2p\sigma^2 T + 4nC \int_0^T \cdot \sqrt{\mathbb{E}[\| \theta_t - \theta_t^{(0)} \|^2]} \partial t \tag{266}$$

By Lemma F.10, we have

$$\mathbb{E}[\| \theta_t - \theta_t^{(0)} \|^2] \le \left( \sigma(2p)^{\frac{1}{2}} t^{\frac{1}{2}} + 2nCt \right)^2 \tag{267}$$

$\square$

Note that this approximation error significantly improves upon the $O(\exp(T))$ error found under standard regularity assumptions (Freidlin et al., 2012, Theorem 1.2, Chapter 2.1).

We present a privacy guarantee for the clipped Langevin diffusion by deriving an upper bound on the KL divergence.

**Theorem F.4** (KL Divergence Bound for Clipped Langevin Diffusion). *Let $\theta_0, \theta_0'$ have the same distribution $\Theta_0, \Theta_0'$, $\theta_T$ be the solution to Equation (255) given initial condition $\theta_0$ and database $D$, $\theta_T'$ be the solution to Equation (255) given initial condition $\theta_0'$ and database $D'$, such that $D \sim D'$. Let $\Theta_{[0,T]}$ be the distribution of the trajectory $\theta_{t \in [0,T]}$. Then for any $T > 0$:*

$$\text{KL}(\Theta_{[0,T]} \| \Theta_{[0,T]}') \le \frac{2n^2 C^2}{\sigma^2} T \tag{268}$$

*Proof of Theorem F.4.* By Theorem B.1 & 3.1 of Ye et al. (2023a),

$$\text{KL}(\Theta_{[0,T]} \| \Theta_{[0,T]}') = \frac{1}{2\sigma^2} \int_0^T \mathbb{E}\left[ \left\| \sum_{i \in [n]} \text{clip}_C(\nabla \ell_i(\theta; D)) - \sum_{i \in [n]} \text{clip}_C(\nabla \ell_i(\theta; D')) \right\|_2^2 \right] dt$$

$$\leq \frac{1}{2\sigma^2} \int_0^T 4n^2C^2 dt$$
$$= \frac{2n^2C^2}{\sigma^2} T$$

$\square$

We demonstrate that our main result on feature distortion holds for clipped Langevin diffusion, reinforcing our paper's key insight. Here, our approximation technique is essential, as the stochastic analysis of Langevin diffusion with nonlinear & nonconvex coefficients would be extremely challenging without it.

**Theorem F.5** (Random initialization causes feature distortion). *If Assumption 3.1 and Assumption 3.2 hold, and the linear head is randomly initialized by $v_0 \sim \mathcal{N}(0, \beta I_{h \times h})$, then with probability $1 - 2^{-h}$, $\forall \beta > 0, \exists j \in [h], \Delta t > 0$ such that during the time interval $(0, \Delta t)$, DP-FFT distorts $w_j$ reducing its alignment with the data cluster. The cosine similarity between $w_j$ and the data cluster mean $\bar{x}_{c(j)}$ decreases monotonically:*

$$\frac{\partial}{\partial t} \cos\left(w_j, \bar{x}_{c(j)}\right) \Big|_t < 0, \quad \forall t \in (0, \Delta t) \tag{269}$$

*Proof of Theorem F.5.* The per-sample gradient for the $i$-th data point (before clipping) is

$$\nabla_{(v,W)}\ell_i = \begin{bmatrix} \nabla_v \ell_i \\ \mathrm{vec}(\nabla_W \ell_i) \end{bmatrix} = \begin{bmatrix} y_i \ell_i \mathrm{relu}(W^\top x_i) \\ y_i \ell_i v_1 \mathrm{relu}'(w_1^\top x_i) x_i \\ y_i \ell_i v_2 \mathrm{relu}'(w_2^\top x_i) x_i \\ \vdots \\ y_i \ell_i v_h \mathrm{relu}'(w_h^\top x_i) x_i \end{bmatrix} = y_i \ell_i \begin{bmatrix} \mathrm{relu}(W^\top x_i) \\ v_1 \mathrm{relu}'(w_1^\top x_i) x_i \\ v_2 \mathrm{relu}'(w_2^\top x_i) x_i \\ \vdots \\ v_h \mathrm{relu}'(w_h^\top x_i) x_i \end{bmatrix} \tag{270}$$

where the $\mathrm{vec}(\cdot)$ operator is defined as an operation that converts a tensor to a vector (Magnus & Neudecker, 1999, Chapter 2.4). We use $\mathrm{vec}(\cdot)$ to collect the gradients of $v$ and $W$ into one vector. Then we can write the clipped per-sample gradient for the $i$-th data point as:

$$\mathrm{clip}_C(\nabla_{(v,W)}\ell_i) = \min\left(1, \frac{C}{\|\nabla_{(v,W)}\ell_i\|_2}\right) \cdot y_i \ell_i \begin{bmatrix} \mathrm{relu}(W^\top x_i) \\ v_1 \mathrm{relu}'(w_1^\top x_i) x_i \\ v_2 \mathrm{relu}'(w_2^\top x_i) x_i \\ \vdots \\ v_h \mathrm{relu}'(w_h^\top x_i) x_i \end{bmatrix}. \tag{271}$$

Therefore, the dynamics of the parameter $w_j$ for any $j \in [h]$ under gradient clipping is,

$$\frac{\partial w_j}{\partial t} = \min\left(1, \frac{C}{\|\nabla_{(v,W)}\ell_i\|_2}\right) \cdot y_i \ell_i \cdot v_j \mathrm{relu}'(w_j^\top x_i) x_i \tag{272}$$

Note that the clipping operation only multiplies the gradient with a normalization term $\min\left(1, \frac{C}{\|\nabla_{(v,W)}\ell_i\|_2}\right)$. As a result, it does not change the signs of the gradient entries. Then we are ready to analyze the cosine similarity between $w_j$ and the mean data direction:

$$\frac{\partial}{\partial t} \cos(w_j, \bar{x}_{c(j)}) = \frac{2(w_j^\top \bar{x}_{c(j)})}{\|w_j\|_2^2} \left[\|w_j\|_2^2 \bar{x}_{c(j)}^\top \frac{\partial w_j}{\partial t} - \bar{x}_{c(j)}^\top w_j w_j^\top \frac{\partial w_j}{\partial t}\right] \tag{273}$$

$$= \frac{2(w_j^\top \bar{x}_{c(j)})}{\|w_j\|_2^2} \left[\|w_j\|_2^2 \bar{x}_{c(j)} - (\bar{x}_{c(j)}^\top w_j) w_j\right]^\top \frac{\partial w_j}{\partial t} \tag{274}$$

$$//\text{by Assumption 3.2} \tag{275}$$

$$\mathrm{sign}\left(\frac{\partial}{\partial t} \cos(w_j, \bar{x}_{c(j)})\right) = \mathrm{sign}\left(\left[\|w_j\|_2^2 \bar{x}_{c(j)} - (\bar{x}_{c(j)}^\top w_j) w_j\right]^\top \frac{\partial w_j}{\partial t}\right) \tag{276}$$

$$//\text{the clipping operation perserves the sign} \tag{277}$$

$$=\text{sign}\left(v_j(\|w_j\|_2^2 - (\bar{x}_{c(j)}^\top w_j)^2)\right) \tag{278}$$

$$=\text{sign}(v_j) \tag{279}$$

Since we initialize $v \sim \mathcal{N}(0, \beta I_{h \times h})$, with probability $1 - 2^{-h}$, there exists $j$ such that $v_j < 0$ at $t = 0 \implies \frac{\partial}{\partial t}\cos(w_j, \bar{x}_{c(j)}) < 0$ at $t = 0$. By the continuity of the approximated Langevin diffusion, there exists $\Delta t > 0$ such that for any $t \in (0, \Delta t)$,

$$\frac{\partial}{\partial t}\cos(w_j, \bar{x}_{c(j)}) < 0. \tag{280}$$

$\square$

We establish that a unique and strong solution exists for the clipped Langevin diffusion. This result is particularly noteworthy because it bypasses the standard regularity assumptions typically required in existence proofs for stochastic differential equations (Mao, 1997; Øksendal, 2014). Standard conditions demand that both the drift and diffusion coefficients exhibit linear growth in their parameters and are Lipschitz continuous. However, such assumptions are often impractical for the loss functions prevalent in modern machine learning. Additionally, deep learning architectures frequently introduce non-differentiability (as seen in the discontinuities of ReLU activation functions, for instance). In response, we propose relaxed regularity criteria to address these challenges.

**Theorem F.6** (Criteria of unique strong solution for SDE with irregular drift (Veretennikov, 1981, Theorem 1))**.** *Consider the following stochastic differential equation:*

$$dx_t = a(x_t, t)dt + b(x_t, t)dX_t \tag{281}$$

*where*

- *$X_t$ denotes the standard Wiener process.*

- *$a$ is a bounded, d-dimensional vector-valued, measurable function.*

- *$b$ is a bounded, matrix-valued, continuous measurable function of size $d \times d$. $b$ satisfies the following properties:*

  - *(Uniform elliptic condition): For any $x \in \mathbb{R}^d, v \in \mathbb{R}^d, t \geq 0$, there exists a constant $\lambda > 0$ such that*

    $$v^T b(x, t)b^T(x, t)v \geq \lambda v^T v \tag{282}$$

  - *(Fixed time uniform continuity): For every $T > 0$ and any $t \in [0, T]$, $b(\cdot, t)$ is uniformly continuous on any compact metric subspace $U \subset \mathbb{R}^d$.*

*Then a unique strong solution $X_t$ exists for the stochastic differential equation.*

**Corollary F.7.** *If the per-sample loss function $\ell$ has a discontinuity set with Lebesgue measure $0$, then the clipped Langevin diffusion (Equation $(255)$) has a unique strong solution.*

*Remark* F.8 (Toy-case example of Corollary F.7). Consider a 2-layer ReLU network $f$ parametrized by $v \in \mathbb{R}^h, W \in \mathbb{R}^{d \times h}$:

$$f(x) := v^\top \text{relu}\left(W^\top x\right), \tag{283}$$

a singleton training dataset $D := \{(x_0, y_0)\}$:

$$x_0 = \begin{bmatrix} 1 \\ 0 \\ \vdots \\ 0 \end{bmatrix}, \quad y_0 = 1 \tag{284}$$

and exponential loss $\ell(y, \hat{y}) := \exp(-y\hat{y})$. Then the drift coefficient (e.g. $a(x_t, t)$ in Theorem F.6) of the loss Langevin diffusion is

$$-\text{clip}_C\left(\nabla \ell_0(y_0, f(x_0))\right) = -\text{clip}_C\left(\nabla \ell_0(y_0, f(x_0))\right) \tag{285}$$

$$= -\min\left(1, \frac{C}{\|\nabla_{(v,W)} \ell_0\|_2}\right) \cdot y_i \ell_i \begin{bmatrix} \text{relu}(W^\top x_i) \\ v_1 \text{relu}'(w_1^\top x_i) x_i \\ v_2 \text{relu}'(w_2^\top x_i) x_i \\ \vdots \\ v_h \text{relu}'(w_h^\top x_i) x_i \end{bmatrix} \tag{286}$$

The set of all discontinuities of this drift coefficient has Lebesgue measure zero in the parameter space $\mathbb{R}^h \times \mathbb{R}^{d \times h}$. This drift coefficient is a measurable function. So we can apply Theorem F.6 in this example.

**Theorem F.9** (Exitence of stationary distribution (Cerrai, 2002, Theorem 2.2.1)). *Consider the following stochastic differential equation:*

$$dx_t = a(x_t)dt + b(x_t)dX_t \tag{287}$$

*where $X_t$ denotes the standard Wiener process, $a$ is $d$-dimensional vector-valued continuous function, and $b$ is a matrix-valued, continuous function of size $d \times d$. If the following conditions hold:*

- *There exists $k \geq 0$ such that*

$$\sup_{x \in \mathbb{R}^d} \frac{\|b(x)\|}{1 + |x|^k} < +\infty \tag{288}$$

- *The function $a$ is locally Lipschitz continuous and there exists $m \geq k$ such that*

$$\sup_{x \in \mathbb{R}^d} \frac{\|a(x)\|}{1 + |x|^{2m+1}} < +\infty \tag{289}$$

- *For any $p \geq 1$ there exists $c_p$ such that for each $x, y \in \mathbb{R}^d$*

$$\langle a(x) - a(y), x - y \rangle + p\|b(x) - b(y)\|_2^2 \leq c_p \|x - y\|_2^2 \tag{290}$$

- *There exist $\nu, \gamma > 0, c \in \mathbb{R}$ such that for any $x, h \in \mathbb{R}^d$*

$$\langle a(x + h) - a(x), h \rangle \leq -\kappa|h|^{2m+2} + c(|x|^\gamma + 1) \tag{291}$$

*Then there exists at least one stationary distribution for the stochastic differential equation.*

### F.1 Technical results

**Lemma F.10** (Gronwall type inequality IV). *Let $x : [a, b] \to \mathbb{R}_+$ be a continuous function that satisfies the inequality:*

$$x(t) \leq M + \int_a^t \Psi(s)\omega(x(s))ds, \quad t \in [a, b]$$

*where $M \geq 0, \Psi : [a, b] \to \mathbb{R}_+$ is continuous and $\omega : \mathbb{R}_+ \to \mathbb{R}_+$ is continuous and monotone-increasing. Then the estimation*

$$x(t) \leq \Phi^{-1}\left(\Phi(M) + \int_a^t \Psi(s)ds\right), \quad t \in [a, b]$$

*holds, where $\Phi : \mathbb{R} \to \mathbb{R}$ is give by*

$$\Phi(u) := \int_{u_0}^u \frac{1}{\omega(s)}ds, \quad u \in \mathbb{R}$$

*Proof of Lemma F.10.* This proof is done by Sever Silvestru Dragomir.

We just copy the proof here for completeness.

Denote $y(t)$ as

$$y(t) := \int_a^t \omega(x(s))\Psi(s)ds, \quad t \in [a, b]$$

we have $y(a) = 0$, and by the recursive integral condition of $x$, we obtain:

$$y'(t) = x(t)\Psi(t), \quad t \in [a, b]$$
$$y'(t) \leq \omega(M + y(t))\Psi(t)$$
$$\frac{1}{\omega(M + y(t))}\mathrm{d}(y(t)) \leq \Psi(t)\mathrm{d}t$$

By integration on $[a, t]$, we have

$$\left(\int_0^{y(t)} \frac{1}{\omega(M + s)}ds\right) - \Phi(M) \leq \int_a^t \Psi(s)ds$$

$$\int_0^{y(t)} \frac{1}{\omega(M + s)}ds \leq \int_a^t \Psi(s)ds + \Phi(M)$$

that is,

$$\Phi(y(t) + M) \leq \int_a^t \Psi(s)ds + \Phi(M)$$

By taking the inverse mapping of $\Phi$ on both sides, we finish the proof. $\qquad\square$

