# OpenReview forum: "Characterizing the Training Dynamics of Private Fine-tuning with Langevin diffusion"
_TMLR — Accepted by TMLR_

### Review · Reviewer_P6M5 · 2025-08-02

**Summary Of Contributions:**

One way to obtain a private neural model is to pre-train a representational or "backbone" model $w$ on public data then fine-tune with a linear head $v$ on private data. There are two variants of fine-tuning under consideration: full fine-tuning (FFT), where all weights are updated, and Linear-probing-first (LP-FFT), where the head is fine-tuned before a full fine-tuning pass. The authors extend the prior work by Kumar et al., who found that LP-FFT mitigates OOD errors caused by random initialization of the linear head, to the case where test data is in-distribution and where Gaussian noise to SGD is added for differential privacy (DPSGD).

Their theoretical results begin with a bound on the error of a zeroth order approximation of Langevin diffusion, which prior work by Ganesh et al. has used to model DPSGD. This is then used to explain why random initialization of $v$ is initially bad for $w$ when performing DP-FFT (Theorem 3.3), as well as why LP alleviates the distortion in representation(s) (Theorem 3.4). The zeroth order approximation is also used to derive bounds on convergence after running DP-LP (Theorem 4.2) and running DP-FFT (Theorem 4.3). Finally, the authors use the approximation to derive how much "privacy budget" to spend on the LP pass to avoid distortions (Theorem 5.1).

In Figure 4, the authors validate Theorems 3.3 & 3.4 by pre-training on Imagenet-1K, fine-tuning on CIFAR-10 & STL-10. In Table 1, they validate Theorem 5.1---"a greater proportion of the privacy budget should be allocated to DP-LP whenthe total privacy budget is smaller". In Table 2, they present evidence that such a pattern still holds when FFT is replaced with LoRA.

**Audience:**

Yes

**Audience Explanation:**

DP training and fine-tuning is a very active area of research. Additionally, the zeroth order approximation definitely would be appealing to those already aware of the Langevin diffusion analysis framework.

**Claims And Evidence:**

Yes

**Claims Explanation:**

The introduction gives a good exposition of the objectives of the paper. The theoretical results are generally presented in an easy-to-parse manner. And, in my opinion, the experimental results cover all the bases.

**Requested Changes:**

Clarify the following:
- Theorems 3.3 and 3.4 concern DP processes, but the text surrounding it makes claims about non-DP processes as well (see also 2. in "Main contributions"). Are the authors claiming to have contributed a result for both DP and non-DP fine tuning, or just DP?
- I am unsure what story is told by the paragraph beginning with the bold text "DP-LP-FFT outperforms other fine-tuning methods: Pre-training on ImageNet." Given that $\sigma$ is fixed to 0.3, doesn't this amount to a (extra) column of Table 1, where privacy budget is fixed to some quantity?

Minor (typographical) errors:
- Inconsistency between N in (5) and n in Theorem 2.2
- Assumption 3.1 should use i,j instead of 1,2

---

> ### Author Response · Authors · 2025-08-20
>
> We thank the reviewer for the constructive and detailed feedback. Below we respond point-by-point and indicate the manuscript changes we will make in the camera-ready version.
>
> ## Requested changes 1 (Scope of Theorem 3.3 & 3.4)
> > “Theorems 3.3 and 3.4 concern DP processes, but the text surrounding it makes claims about non-DP processes as well (see also 2. in "Main contributions"). Are the authors claiming to have contributed a result for both DP and non-DP fine tuning, or just DP?”
>
> - Response: **The result works for both DP and non-DP**. Letting the noise multiplier $\sigma \rightarrow 0$ removes the privacy perturbation but does not eliminate the perturbation introduced by the random initialisation of the linear head. In our Langevin-diffusion view, the privacy noise and the initialisation error simply add. Therefore when $\sigma=0$ the stochastic term drops out yet the residual error from the head initialisation remains, causing the same early-stage representation distortion predicted by Theorem 3.3. Consequently, Theorem 3.4 (effect of LP) and all subsequent convergence bounds continue to apply.
>
> - Revision: We will add an explicit corollary $(\sigma=0)$ right after the two theorems to highlight the non-DP interpretation.
>
> ## Requested changes 2 (pre-training on ImageNet)
> > “I am unsure what story is told by the paragraph beginning with the bold text "DP-LP-FFT outperforms other fine-tuning methods: Pre-training on ImageNet." Given that is fixed to 0.3, doesn't this amount to a (extra) column of Table 1, where privacy budget is fixed to some quantity?”
>
> - Response: Figure 5 zooms in on the LP-FT transition that Table 1 keeps fixed, revealing how accuracy changes when we vary the number of LP versus FT epochs at a constant noise level. Table 1 compares different privacy budgets by varying the noise multiplier $\sigma$ while fixing the training schedule (5 LP epochs followed by 5 FT epochs). In contrast, Figure 5 holds $\sigma$ constant and sweeps the number of LP epochs from 0 to 10 before finishing with FT, thereby probing exactly the LP-vs-FT trade-off that Table 1 does not visualise. This sweep traces the utility curve predicted by Corollary 5.3: moderate allocation to LP is optimal when σ is moderate, whereas too many or too few LP steps hurt performance. Thus Figure 5 complements Table 1 rather than replicating it.
>
> - Revision:
>    - We will move the sentence “$\sigma= 0,\; 0.3$” to the beginning of the paragraph and add: “Here we fix $\sigma$ and vary e_{LP} to trace the full utility curve predicted by Corollary 5.3; Table 1 instead varies $\sigma$ (hence $\epsilon$) at a fixed e_{LP}=3.”
>    - We will cross-reference Figure 5 to Corollary 5.3 explicitly.
>
> ## Requested changes 3 & 4 (minor typographical issues)
>    - N vs n in Eq. (5) and Theorem 2.2: We will unify the symbol n throughout.
>    - Assumption 3.1 indices: We will replace “1, 2” with “i, j”.

---

### Review · Reviewer_kFvb · 2025-08-04

**Summary Of Contributions:**

This is an interesting paper!

The authors study the dynamics of fine-tuning a model under differential privacy constraints. As I understand it, such models are first trained on public data and then they are appropriately fine-tuned on private data. To reduce recall of specific instances from the private data, techniques like DP-SGD are used, which includes clipping and noise injection. Such properties can make formal analysis challenging.

To study such dynamics, typically researchers resort to various approximations, e.g., Taylor expansions of models, perturbations, etc., which make analysis tractable, albeit at the cost of losing some of the information about the true behavior of the model.

The authors focus on the alignment dynamics between a randomly-initialized head and a pre-trained backbone under various fine-tuning schedules: full fine-tuning of both parts (DP-FFT), fine-tuning of only the head (DP-LP) or short fine-tuning of the head followed by longer fine-tuning of the backbone (DP-LP-FFT), specifically for the DP case.

# Contributions

## Theory

The authors construct a zeroth-order theory that captures the feature distortion when jointly fine-tuning a randomly-initialised linear probe with a pre-trained backbone during DP-SGD.
- To construct the zeroth-order theory, the authors introduce an ODE derived by setting the noise term of a Langevin diffusion model of DP-SGD to zero, getting $\partial \tilde{\theta}=-\nabla L(\tilde{\theta}|f)\partial t$. This ODE is used to analyze the behaviors of the various schemes.
- The authors provide an upper bound for the $L_2$ error of $\tilde{\theta}_t$ from the exact $\theta_t$.
- Theorems that demonstrate the initial distortion due to DP-FFT as well as the stability of features when using DP-LP-FFT, as well as more theoretical analysis between the zeroth-order approximation and the initial Langevin diffusion.
- A study of the privacy budget allocation for users of DP-LP-FFT, with an explicit formula for the zeroth-order approximation.

## Experiments

On ImageNet→STL-10/CIFAR-10 and LoRA-style parameter-efficient tuning, the results track the theory (except for the case of Figure 2b, also highlighted by the authors as a case of synthetic pre-training data).


# Strengths
- As you can see from the above notes, there is a lot of content in those 12 pages, but at no place did I feel like the writing was dense.
- The zeroth-order analysis is presented clearly, with a lot more content in the appendix.
- Experiments even look into parameter-efficient fine-tuning techniques like LoRA, and broadly agree with the theoretical results.
- There is clear experimental guidance provided, e.g., the privacy budget allocation for the DP-LP-FFT method.

# Weaknesses
- It would be interesting to see experiments in other modalities, e.g., text.


[1]: As in the work of Min, H., Mallada, E. and Vidal, R., 2023. Early neuron alignment in two-layer relu networks with small initialization. arXiv preprint arXiv:2307.12851.

**Audience:**

Yes

**Audience Explanation:**

TMLR readers who care about differential privacy, transfer learning, and the theory-practice gap in DP-SGD will find the paper worthwhile. The zeroth-order surrogate offers a novel analytic tool for understanding how DP noise perturbs feature representations during fine-tuning, and the DP-LP-FFT schedule provides actionable guidance on privacy-budget allocation.

**Broader Impact Concerns:**

No concerns.

**Claims And Evidence:**

Yes

**Claims Explanation:**

The claims are:

1. the explanation of the feature distortion effect under the zeroth-order theory, including the practical suggestions re: budget allocation and
2. the experimental results on vision examples.

Point 1. is backed by the theory, as developed in the main paper and expanded in the appendix. The assumptions are clear and the proofs seem to me to be solid.

For the experiments, sufficient detail is given to reproduce the experiments, which use standard datasets and models.

**Requested Changes:**

# Necessary changes

1. In "Assumption 3.1", there is a typo, $i,j$ are not used in the expression (Equation (8)).
2. "We follow the approximation scheme outlines in Equation  (4)to derive" -- equation label needs some space from "to".
3. Proposition 5.2 has some typos, e.g., "Assumption #.7 hold**s**".

# Changes that are not necessary, but may strengthen the work.

1. The sentence "From a contrastive learning viewpoint, it makes the representation of them semantically similar." could probably be tightened up a bit to be more specific, e.g., "it=DP-SGD" (?), etc.
2. Perhaps highlighting a bit more the example in Figure 2b -- it's always interesting to look into examples that don't follow the theory closely.

---

> ### Author Response · Authors · 2025-08-20
>
> We thank the reviewer for the thoughtful and encouraging feedback. Below we address each point in the "Weaknesses" and "Requested Changes" sections and outline the manuscript updates we will make in the camera-ready version.
>
> ## Additional Modality Experiments
> We appreciate the suggestion to test text models; however, given space and compute limits we were planning to defer a full NLP investigation to future work. The linear-head-plus-backbone abstraction we analyse is a faithful model for vision fine-tuning pipelines, but modern NLP practice often adopts parameter-efficient adapters and non-linear heads that lie outside our theoretical scope. However, if the reviewer thinks it would be necessary to include NLP experiments, we can try to put something together prior to the revision deadline.
>
> ## Clarifications & Stylistic Improvements
> 1. We will replace the unused symbols $i,j$ and ensure indices align with later notation.
> 2. We will insert space: "…scheme outlined in Equation (4) to derive…".
> 3. For proposition 5.2, we will correct the typos: "Assumption 3.7 holds", "satisfies $\delta$ ≤ …".
> 4. We will re-write the sentence as: "From a contrastive-learning viewpoint, DP-SGD noise pulls semantically similar examples closer in the embedding space by…".
>
> ## Highlighting Figure 2b
> 1. We will rewrite the sentence as: “From a contrastive-learning viewpoint, Assumption 3.2 makes the representation of them semantically similar”.
> 2. Figure 2b follows the Tang et al. (2023) protocol, which introduces EMA smoothing and gradient-averaging across augmentations before clipping. These two ingredients are absent from our theoretical setup, and these modifications dampen the representation-distortion predicted by Theorem 3.3. Our interpretation of Figure 2b is currently heuristic and is an early-stage conjecture rather than a formally proved result.
>
>    1. EMA: Tang et al. maintain an EMA copy of the network parameters and report accuracy with that averaged model. EMA acts as a low-pass filter on the parameter trajectory, effectively smoothing out the rapid weight adjustments induced by the large initial head-gradient. This could delay the transient distortion our theory attributes to the first few DP-FFT steps.
>
>    2. Gradient averaging over augmentations: Before per-example clipping, Tang et al. average the gradients of multiple augmentations of the same image. Averaging reduces variance and shrinks the expected norm of each per-example gradient, lowering the probability that the clipping threshold is hit. Consequently, the random-initialisation error injected by the head could have a smaller effective magnitude. This potentially mitigates the early distortion phase.
>
>    **Revision**: We will place the above explanation on Figure 2b in Appendix A.2. The manuscript change log will be updated accordingly.

---

> > ### Comment · Reviewer_kFvb · 2025-09-15
> > **Thank you for the updates**
> >
> > I'll be reviewing the answers to the other reviewers as well.
> >
> > >However, if the reviewer thinks it would be necessary to include NLP experiments, we can try to put something together prior to the revision deadline.
> >
> > No worries, I don't think that would be necessary for the story you are telling here.

---

### Review · Reviewer_UYLt · 2025-08-29

**Summary Of Contributions:**

**Summary**

This paper studies the training dynamics of DP-SGD for pretrained models. The authors model clipped DP-SGD as a Langevin diffusion and derive a zeroth-order approximation for analysis. Under this lens, they show that naïve full-parameter DP fine-tuning initially misaligns a randomly initialized head with pretrained features, causing a temporary drop in representation quality (feature distortion). They propose a hybrid strategy, DP linear probing followed by full fine-tuning (DP-LP-FFT), and provide guidance on splitting the privacy budget between LP and FFT. Experiments on CIFAR-10/STL-10 with ResNet/ViT backbones (plus LoRA variants) support the theory: small noise favors DP-FFT, moderate noise favors DP-LP-FFT, and large noise favors DP-LP.

**Strengths And Weaknesses**

Strengths:
1. The paper addresses an interesting problem and presents clear takeaways.
2. Splitting the privacy budget between LP and FFT is a promising idea.
3. The hybrid method is theoretically grounded and empirically well supported.

Weaknesses:
1. Assumption 3.2 (and Assumption 3.1 partially) seems quite strong for a binary classification setting. In practice, this assumption is likely violated, which could undermine the validity of the theory.
2. The theoretical analysis is conducted on simplified two-layer ReLU/linear networks, which differ substantially from the architectures used in practice. This gap limits the external validity and practical utility of the results.
3. More details regarding the hyperparameters of DP-SGD in the experiments should be presented, such as the clipping norm C, learning rate, etc.

**Audience:**

Yes

**Audience Explanation:**

Yes, I think some audience will be interested in the results of this paper.

**Claims And Evidence:**

Yes

**Claims Explanation:**

The claims are accurate and well supported by the theory or experiments.

**Requested Changes:**

1. Authors should better discuss the experimental designs in more details.

2. It is better to plot the privacy-utility curve (accuracy, epsilon) for DP-LP, DP-LP-FFT, and DP-FFT (Table 1) if possible.

---

> ### Author Response · Authors · 2025-09-08
>
> We thank the reviewer for the feedback.
> Below we comment on the weaknesses and requested additions.
>
> ## Weakness 1: strength of Assumption 3.1 and 3.2
> > “Assumption 3.2 (and Assumption 3.1 partially) seems quite strong for a binary classification setting. In practice, this assumption is likely violated, which could undermine the validity of the theory.”
>
> Assumptions 3.1-3.2 can be softened at the cost of looser constants. However, the more salient points are that (1) these assumptions are only used in the early phase of training (the “feature distortion” phase), and (2) our qualitative predictions (initial distortion under DP‑FFT, mitigation under DP‑LP-FFT) hold empirically even when the assumptions are not perfectly satisfied. The assumptions serve as analytical scaffolding to isolate the head–backbone alignment mechanism under DP-SGD, allowing us to make precise statements, which we later validate empirically. They are not intended as literal data‑generating truths.
>
> Revisions we will make: Clarify in section 3 that the assumptions are local/early‑phase and serve to make the distortion mechanism transparent; we will also move informal guidance to section 6 (Limitations) to emphasize that the assumptions can be softened at the cost of looser constants.
>
> We relax Assumption 3.1 by allowing non‑zero cross‑class correlation, controlled by a parameter $ \rho\in [0,1) $, and we relax Assumption 3.2 by allowing bounded activation leakage of a feature $w_j$ onto the opposite class, also controlled by $\rho$. (Setting $\rho=0$ recovers the original assumptions.
>
> Relaxed Assumption 3.1: There exists $\mu_{\mathrm{in}}$ and $\rho\in[0,1)$ such that for all $i\not=j$,
> $$
> \text{ (within class) } y_i​ =y_j ​ \Longrightarrow \frac{ \langle x_i,x_j\rangle }{ \|x_i\| \|x_j\| } \ge \mu_{\mathrm{in}},\\
> \text{ (across class) } y_i\not=y j ​ \Longrightarrow \frac{ \langle x_i,x_j\rangle }{ \|x_i\| \|x_j\| }​ \le \rho \mu_{\mathrm{in}}.
> $$
> Equivalently, the (label‑signed) pairwise cosine similarity has a positive gap
> $$
> \inf_{y_i=y_j}\cos(x_i,x_j)-\sup_{y_i\not=y_j}\cos(x_i,x_j) \ge (1-\rho)\mu_{\mathrm{in}} > 0
> $$
> This weakens Assumption 3.1, which enforced a sign separation (across‑class cosines ≤ -μ ≤-μ), to a gap separation that permits some positive cross‑class correlation. The original Assumption 3.1 and its cone construction appear around Eq. (6).
>
> Relaxed Assumption 3.2: Let $c(j)\in\{+1,-1\}$ be the class index associated with feature $w_j$ (same convention as the original Assumption 3.2). Define the “activated mass” at t=0 for $w_j$ under the exponential loss weights $\ell_i=\exp(-y_i f(x_i))$:
> $$
> A_j^{+}=\sum_{i:y_i=c(j)}\ell_i(0)\boldsymbol{1} (w_j(0)^{\top}x_i>0) ,\\
> A_j^{-}=\sum_{i:y_i\not=c(j)}\ell_i(0)\boldsymbol{1}(w_j(0)^{\top}x_i>0)
> $$
> Assume leakage is bounded by the same $\rho$ as above:
> $$
> \forall j,  A_j^{-} \le \rho A_j^{+}
> $$
> And the pre-trained features are not well aligned yet with the downstream data, i.e. we need to fine-tune the features. We describe this by an upper bound upon the alignment
>
> $$
> \cos ( \bar{x} , w_{j} ) < \mu_{ \mathrm{in} } (1-\rho^2).
> $$
>
> This is a quantitative relaxation of Assumption 3.2 (the old statement implied $A_j^{-}=0$).
>
> Reviewers can find more details and proofs in Appendix B.1 in our updated submission.
>
> ## Weakness 2: two‑layer analysis vs. deep networks
> > “The theoretical analysis is conducted on simplified two-layer ReLU/linear networks, which differ substantially from the architectures used in practice. This gap limits the external validity and practical utility of the results.”
>
> Our use of a two‑layer surrogate and a zeroth‑order ODE is a local approximation around the pre‑trained weights. In the short horizon that governs the distortion phase, it has been previously shown that deep networks behave approximately like their linearization [1,2,3]; the dominant term is the interaction between the head’s random initialization and the backbone’s Jacobian under DP‑SGD updates. This is precisely what our surrogate captures.
>
> Revisions we will make:
> 1. Add to section 3: an explicit discussion of the model choice and why the zeroth‑order ODE captures the envelope of early‑stage DP‑SGD dynamics.
> 2. Insert a pointer to section 5 noting that our main experiments intentionally follow a plain DP‑SGD recipe (no EMA, no cross‑augmentation gradient averaging), so the empirical trends align closely with theory
>
> [1] (ICLR 2022 Oral) Ananya Kumar, Aditi Raghunathan, Robbie Jones, Tengyu Ma, Percy Liang. Fine-Tuning can Distort Pretrained Features and Underperform Out-of-Distribution.
>
> [2] (NeurIPS 2018 Spotlight) Arthur Jacot, Franck Gabriel, Clement Hongler. Neural Tangent Kernel: Convergence and Generalization in Neural Networks.
>
> [3] (NeurIPS 2019). Jaehoon Lee et al. Wide Neural Networks of Any Depth Evolve as Linear Models Under Gradient Descent.

---

> ### Author Response · Authors · 2025-09-08
>
> ## Weakness 3 & requested changes 1: experiment details
> > “More details regarding the hyperparameters of DP-SGD in the experiments should be presented, such as the clipping norm C, learning rate, etc.”
>
> We consolidate the hyper-parameter settings in Appendix A. For experiments in table 1 and table 2, we use clipping thresholds C=0.1 and C=1, use batch size 1000 and sweep over learning rates {9, 5, 1, 0.5, 0.2, 0.15, 0.1, 0.05, 0.025}.
>
> For the vision benchmarks in our paper, these values are based on established empirical studies that explore optimal clipping thresholds for DP-SGD. In particular, Appendix B.1 of Unlocking High-Accuracy Differentially Private Image Classification through Scale by Soham De et al. (2022, DeepMind) provides an in-depth analysis of clipping norms, concluding with the choice of C=1 for their primary experiments.
>
> Our experimental settings also draw from the methodologies outlined in:
>
> [1] Unlocking High-Accuracy Differentially Private Image Classification through Scale by Soham De et al., 2022.
>
> [2] Differentially Private Image Classification by Learning Priors from Random Processes by Tang et al. (NeurIPS 2023).
>
> Revisions we will make: We will consolidate all DP‑SGD hyperparameters into a Main‑Text paragraph.
>
> ## Requested changes 2: privacy-utility curves
> > “It is better to plot the privacy-utility curve (accuracy, epsilon) for DP-LP, DP-LP-FFT, and DP-FFT (Table 1) if possible.”
>
> Thanks for the constructive suggestion. We add the plots to Appendix A.1 in the updated submission.
>
> As expected, accuracy increases with epsilon for every method and backbone, and the results generally (but not always) qualitatively match our theoretical predictions.
> 1. For Mini-DeiT-Ti ,the ViT‑style backbone is comparatively robust. DP‑LP‑FFT retains the lead in high epsilon regimes while DP-LP wins for small epsilons, as predicted by our theory.
> 2. For MobileNet-v3 and ResNet-18, the cross-over pattern is different from Mini-DeiT: even at moderate epsilon, DP-LP-FFT outperforms DP-LP, and under strong privacy DP‑LP is best. And DP‑FFT retains the lead over the high epsilon regime. This suggests that small conv‑nets are more prone to head-induced distortion, so the front-loading budget into LP pays off sooner.
> 3. With a deeper conv‑net, the trends predicted by our theory persist: DP-FFT wins at large epsilon, DP‑LP‑FFT at moderate epsilon, DP‑LP at small epsilon. The DP‑LP‑FFT curve sits close to DP‑FFT in the high‑epsilon regime (no downside when noise is small) yet clearly exceeds it as epsilon shrinks, which is exactly the “mitigate‑then‑fine‑tune” behavior predicted by Theorems 3.3-3.4.

---

### Author Response · Authors · 2025-09-08
**Manuscript change log**

We sincerely thank the reviewers for their valuable feedback. As mentioned in our rebuttal, we have committed to incorporating several revisions, and we confirm the following:
1. All updates promised in our rebuttal to be included in the "revised version" have been fully incorporated into the updated submission.

2. Other remaining points raised in the reviews have already been addressed in our rebuttal responses and will be implemented in the final camera-ready version, primarily for better formatting.

Below is a summary of the updates made in the current revised version:

1. Added an explicit corollary $(\sigma=0)$ right after the Theorem 3.3 and Theorem 3.4 to highlight the non-DP interpretation of the distortion results.
2. Added sentence to explain the relation between Figure 5 and Table 1. “Here we fix $\sigma$ and vary e_{LP} to trace the full utility curve predicted by Corollary 5.3; Table 1 instead varies $\sigma$ (hence $\epsilon$) at a fixed e_{LP}=5.”
3. Unified the symbol $N,n$ in Equation 5 and Theorem 2.2.
4. Fixed Assumption 3.1 indices. Replaced "1,2" with "i,j".
5. Added explanation on Figure 2b in Appendix B.2.
6. Clarified in section 3 that Assumption 3.1 and 3.2 are local/early‑phase and serve to make the distortion mechanism transparent.
7. Added relaxed versions of Assumption 3.1 and 3.2 in Appendix B.1.
8. Added relaxed results of Theorem 3.3 in Appendix B.1 based on relaxed assumptions.
9. Added to section 3: an explicit discussion of the model choice and why the zeroth‑order ODE captures the envelope of early‑stage DP‑SGD dynamics.
10. Consolidated all DP‑SGD hyperparameters into a Main‑Text paragraph.
11. Added four privacy-utility curve plots to Appendix A.1.

---

### Decision · Action_Editor_YsYd · 2025-10-07

**Recommendation:** Accept as is

**Additional Comments:**

The reviewers found that the problem studied is interesting and important and an active research area. All reviewers commented on the high quality of the presentation: the paper "presents clear take-aways" (Reviewer UYLt), the "analysis is presented clearly" (Reviewer kFvb), "theoretical results are generally presented in an easy-to-parse manner" (Reviewer P6M5).
Reviewers also found the presented method to be "theoretically grounded and empirically well-supported" (Reviewer UYLt), noting also that there is "clear experimental guidance provided" (Reviewer kFvb), and that "the experimental results cover all the bases." (Reviewer P6M5).

In the rebuttal phase, the authors made clarifications to address reviewer comments, and have revised the paper accordingly, which has improved the presentation of the results.

Unaddressed (or partially addressed) weaknesses identified by reviewers are that the theoretical results are limited to the setting of 2-layer ReLU networks, and that the experiments are limited to vision tasks. Extending the theoretical analysis to deeper and more general models, and extending the empirical analysis to consider language modeling tasks would both strengthen the paper.
However, beyond doubt, the existing contribution meets the bar for publication at TMLR.

**Audience:**

Yes

**Audience Explanation:**

Differentially-private finetuning of public pretrained models is an important area of study that is relevant for TMLR. This paper makes an important contribution in this area by showing that (and explaining theoretically why) the standard finetuning approach can distort the pretrained backbone features, an important issue in practice, while also showing how to mitigate this issue, and why the mitigation is effective.

**Claims And Evidence:**

Yes

**Claims Explanation:**

This paper studies differentially-private finetuning of (public) pretrained models. Drawing inspiration from prior work in transfer learning, they show both theoretically and empirically that starting the private finetuning phase with a randomly-initialized linear head can distort the features of the pretrained backbone. Intuitively, those features can get distorted by having to change suboptimally in order to accommodate the initially-bad linear head values.

The authors develop an approximation scheme, based on a zeroth order approximation of Langevin diffusion. Based on that, they show why feature distortion happens and why LP-FFT -- a two-stage strategy that first does linear probing with the backbone parameters fixed, and then finetunes the backbone parameters too -- mitigates this issue. The authors also leverage this approximation to derive how much "privacy budget" should be spent on the LP phase of LP-FFT to effectively avoid distortions.

The claims made by the authors are backed by sufficient evidence, both theoretical and empirical. The reviewers found that the theory is sound, proving the authors' claims about the interpretation of the feature distortion issue and its mitigation. Experiments on vision tasks further provide empirical evidence, even looking into LoRA parameter-efficient fine-tuning, broadly agreeing with the theoretical results.